# REVERSE DIFFUSION MONTE CARLO

**Xunpeng Huang**[*†]    **Hanze Dong**[*†§]    **Yifan Hao**[†]    **Yi-An Ma**[‡]    **Tong Zhang**[¶]

[†] The Hong Kong University of Science and Technology
[§] Salesforce AI Research
[‡] University of California, San Diego
[¶] University of Illinois Urbana-Champaign

## ABSTRACT

We propose a Monte Carlo sampler from the reverse diffusion process. Unlike the practice of diffusion models, where the intermediary updates—the score functions—are learned with a neural network, we transform the score matching problem into a mean estimation one. By estimating the means of the regularized posterior distributions, we derive a novel Monte Carlo sampling algorithm called reverse diffusion Monte Carlo (rdMC), which is distinct from the Markov chain Monte Carlo (MCMC) methods. We determine the sample size from the error tolerance and the properties of the posterior distribution to yield an algorithm that can approximately sample the target distribution with any desired accuracy. Additionally, we demonstrate and prove under suitable conditions that sampling with rdMC can be significantly faster than that with MCMC. For multi-modal target distributions such as those in Gaussian mixture models, rdMC greatly improves over the Langevin-style MCMC sampling methods both theoretically and in practice. The proposed rdMC method offers a new perspective and solution beyond classical MCMC algorithms for the challenging complex distributions.

## 1 INTRODUCTION

Recent success of diffusion models has shown great promise for the the reverse diffusion processes in generating samples from a complex distribution (Song et al., 2020; Rombach et al., 2022). In the existing line of works, one is given samples from the target distribution and aims to generate more samples from the same target. One would diffuse the target distribution into a standard normal one, and use score matching to learn the transitions between the consecutive intermediary distributions (Ho et al., 2020; Song et al., 2020). Reversing the learned diffusion process leads us back to the target distribution. The benefit of the reverse diffusion process lies in the efficiency of convergence from any complex distribution to a normal one (Chen et al., 2022a; Lee et al., 2023). For example, diffusing a target multi-modal distribution into a normal one is not harder than diffusing a single mode. Backtracking the process from the normal distribution directly yields the desired multi-modal target. If one instead adopts the forward diffusion process from a normal distribution to the multi-modal one, there is the classical challenge of mixing among the modes, as illustrated in Figure 1. This observation motivates us to ask:

*Can we create an efficient, general purpose Monte Carlo sampler from reverse diffusion processes?*

For Monte Carlo sampling, while we have access to an unnormalized density function $p_*(\boldsymbol{x}) \propto \exp(-f_*(\boldsymbol{x}))$, samples from the target distribution are unavailable (Neal, 1993; Jerrum & Sinclair, 1996; Robert et al., 1999). As a result, we need a different and yet efficient method of score estimation to perform the reverse SDE. This leads to the first contribution of this paper. We leverage the fact that the diffusion process from our target distribution $p_*$ towards a standard normal one $p_\infty$ is an Ornstein-Uhlenbeck (OU) process, which admits explicit solutions. We thereby transform the score matching problem into a non-parametric mean estimation one, *without* training a parameterized diffusion model. We name this new algorithm as *reverse diffusion Monte Carlo* (RDMC).

---

[*]Equal contribution. Random order.

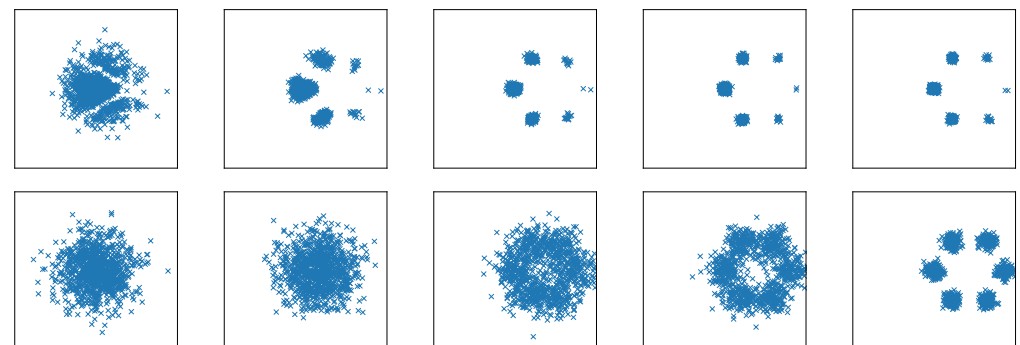

Figure 1: Langevin dynamics (first row) versus reverse SDE (second row). The first and second rows depict the intermediate states of the Langevin algorithm and the reverse SDE, respectively, illustrating the transition from a standard normal $p_0$ to a Gaussian mixture $p_*$. It can be observed that due to the local nature of the information contained in $\nabla \ln p_*$, the Langevin algorithm tends to get stuck in modes close to the initializations. In contrast, the reverse SDE excels at transporting particles to different modes proportional to the target densities.

To implement RDMC and solve the aforementioned mean estimation problem, we propose two approaches. One is to sample from a normal distribution $\rho_t$—determined at each time $t$ by the transition kernel of the OU process—then compute the mean estimates weighted by the target $p_*$. This approach translates all the computational challenge to the sample complexity in the importance sampling estimator. The iteration complexity required to achieve an overall $\epsilon$ TV accuracy is $O(\epsilon^{-2})$, under minimal assumptions. Another approach is to use Unadjusted Langevin Algorithm (ULA) to generate samples from the product distribution of the target density $p_*$ and the normal one $\rho_t$. This approach greatly reduces sample complexity in high dimensions, and yet is a better conditioned algorithm than ULA over $p_*$ due to the multiplication of the normal distribution. In our experiments, we find that a combination of the two approaches excels at distributions with multiple modes.

We then analyze the efficacy and efficiency of our RDMC method. We study the benefits of the reverse diffusion approach for both *multi-modal distributions* and high dimensional heavy-tail distributions that *breaks the isoperimetric properties* (Gross, 1975; Poincaré, 1890; Vempala & Wibisono, 2019). In multi-modal distributions, the Langevin algorithm based MCMC approaches suffer from an exponential cost for the need of barrier crossing, which makes mixing time extremely long. The RDMC approach can circumvent the hurdle and solve the problem. For high-dimensional heavy-tail distributions, our RDMC method circumvents the often-required isoperimetric properties of the target distributions, thereby avoiding the curse of dimensionality.

**Contributions.** We propose a non-parametric sampling algorithm that leverages the reverse SDE of the OU process. Our proposed approach involves estimating the score function by a mean estimation sub-problem for posteriors, enabling the efficient generation of samples through the reverse SDE. We focus on the complexity of the sub-problems and establish the convergence of our algorithm. We found that our approach effectively tackles sampling tasks with ill-conditioned log-Sobolev constants. For example, it excels in challenging scenarios characterized by multi-modal and long-tailed distributions. Our analysis sheds light on the varying complexity of the sub-problems at different time points, providing a fresh perspective for revisiting score estimation in diffusion-based models.

## 2 PRELIMINARIES

In this section, we begin by introducing related work from the perspectives of Markov Chain Monte Carlo (MCMC) and diffusion models, and we discuss the connection between these works and the present paper. Next, we provide a notation for the reverse process of diffusion models, which specifies the stochastic differential equation (SDE) that particles follow in RDMC.

We first introduce the related works as below.

**Langevin-based MCMC.** The mainstream gradient-based sampling algorithms are mainly based on the continuous Langevin dynamics (LD) for sampling from a target distribution $p_* \propto \exp(-f_*)$. The Unadjusted Langevin Algorithm (ULA) discretizes LD using the Euler-Maruyama scheme and

obtains a biased stationary distribution. Due to its simplicity, ULA is widely used in machine learning. Its convergence has been investigated for different criteria, including Total Variation (TV) distance, Wasserstein distances (Durmus & Moulines, 2019), and Kullback-Leibler (KL) divergence (Dalalyan, 2017; Cheng & Bartlett, 2018; Ma et al., 2019; Freund et al., 2022), in different settings, such as strongly log-concave, log-Sobolev inequality (LSI) (Vempala & Wibisono, 2019), and Poincaré inequality (PI) (Chewi et al., 2021). Several works have achieved acceleration convergence for ULA by decreasing the discretization error with higher-order SDE, e.g., underdamped Langevin dynamics (Cheng et al., 2018; Ma et al., 2021; Mou et al., 2021), and aligning the discretized stationary distribution to the target $p_*$, e.g., Metropolis-adjusted Langevin algorithm (Dwivedi et al., 2018) and proximal samplers (Lee et al., 2021b; Chen et al., 2022b;b; Altschuler & Chewi, 2023). Liu & Wang (2016); Dong et al. (2023) also attempt to perform sampling tasks with deterministic algorithms whose limiting ODE is derived from the Langevin dynamics.

Regarding the convergence guarantees, most of these works have landscape assumptions for the target distribution $p_*$, e.g., strong log-concavity, LSI, or PI. For more general distributions, Erdogdu & Hosseinzadeh (2021) and Mousavi-Hosseini et al. (2023) consider KL convergence in modified LSI and weak PI. These extensions allow for slower tail-growth of negative log-density $f_*$, compared to the quadratic or even linear case. Although these works extend ULA to more general distributions, the computational burden of ill-conditioned isoperimetry still exists, e.g., exponentially dependent on the dimension (Raginsky et al., 2017). In this paper, we introduce another SDE to guide the sampling algorithm, which is non-Markovian and time-inhomogeneous. Our algorithm discretizes such an SDE and can reduce the isoperimetric and dimension dependence wrt TV convergence.

**Diffusion Models and Stochastic Localization.** In recent years, diffusion models have gained significant attention due to their ability to generate high-quality samples (Ho et al., 2020; Rombach et al., 2022). The core idea of diffusion models is to parameterize the score, i.e., the gradient of the log-density, during the entire forward OU process from the target distribution $p_*$. In this condition, the reverse SDE is associated with the inference process in diffusion models to perform unnormalized sampling for intricate target distributions. Apart from conventional MCMC trajectories, the most desirable property of this process is that, if the score function can be well approximated, it can sample from a general distribution Chen et al. (2022a); Lee et al. (2022; 2023). This implies the reverse SDE has a lower dependency on the isoperimetric requirement for the target distribution, which inspires the designs of new sampling algorithms. On the other hand, stochastic localization framework (Eldan, 2013; 2020; 2022; El Alaoui et al., 2022; Montanari, 2023) formalizes the general reverse diffusion and discusses the relationship with current diffusion models. Along this line, Montanari & Wu (2023) consider the unnormalized sampling for the symmetric spiked model. However, for the properties of the general unnormalized sampling, the investigation is limited.

This work employs the backward path of diffusion models to design a sampling strategy which we can prove to be more effective than the forward path of Langevin algorithm under suitable conditions. Several other recent studies (Vargas et al., 2023a; Berner et al., 2022; Zhang et al., 2023; Vargas et al., 2023c;b) have also utilized the diffusion path in their posterior samplers. Instead of the closed form MC sampling approach, the above works learns an approximate score function via a parametrized model, following the standard practice of the diffusion generative models. The resulting approximate distribution following the backward diffusion path is akin to the ones obtained from variational inference, and contains errors that depend on the expressiveness of the parametric models for the score functions. In addition, the convergence of the sampling process using such an approach depends on the generalization error of learned score functions (Tzen & Raginsky, 2019) (See Appendix for more details). It remains open how to bound sample complexity of such approaches, due to the need for bounding the error of the learned score function along the entire trajectory.

In contrast, RDMC proposed in this paper can be directly analyzed. Specifically, our algorithm draws samples from the unnormalized distribution without training data and the algorithm is designed to avoid learning based score function estimation. We are interested in the comparisons between RDMC and conventional MCMC algorithms. Thus, we analyze the complexity to estimate the score by drawing samples from the posterior distribution and found that our proposed algorithm is much more efficient in certain cases when the log-Sobolev constant of the target distribution is small.

**Reverse SDE in Diffusion Models.** In this part, we introduce the notations and formulation of the reverse SDE associated with the inference process in diffusion models, which will also be commonly used in the following sections. We begin with an OU process starting from $p_*$ formulated as

$$\mathrm{d}\boldsymbol{x}_t = -\boldsymbol{x}_t \mathrm{d}t + \sqrt{2}\mathrm{d}B_t, \quad \boldsymbol{x}_0 \sim p_0 \propto e^{-f_*}, \tag{1}$$

where $(B_t)_{t\geq 0}$ is Brownian motion and $p_0$ is assigned as $p_*$. This is similar to the forward process of diffusion models (Song et al., 2020) whose stationary distribution is $\mathcal{N}(\mathbf{0}, \boldsymbol{I})$ and we can track the random variable $\boldsymbol{x}$ at time $t$ with density function $p_t$.

According to Cattiaux et al. (2021), under mild conditions in the following forward process $\mathrm{d}\boldsymbol{x}_t = b_t(\boldsymbol{x}_t)\mathrm{d}t + \sqrt{2}\mathrm{d}B_t$, the reverse process also admits an SDE description. If we fix the terminal time $T > 0$ and set $\tilde{\boldsymbol{x}}_t = \boldsymbol{x}_{T-t}$, for $t \in [0, T]$, the process $(\tilde{\boldsymbol{x}}_t)_{t\in[0,T]}$ satisfies the SDE $\mathrm{d}\tilde{\boldsymbol{x}}_t = \tilde{b}_t(\tilde{\boldsymbol{x}}_t)\mathrm{d}t + \sqrt{2}\mathrm{d}B_t$, where the reverse drift satisfies the relation $b_t + \tilde{b}_{T-t} = 2\nabla \ln p_t, \boldsymbol{x}_t \sim p_t$. In this condition, the reverse process of Eq. (1) is as $\mathrm{d}\tilde{\boldsymbol{x}}_t = (\tilde{\boldsymbol{x}}_t + 2\nabla \ln p_{T-t}(\tilde{\boldsymbol{x}}_t)) \mathrm{d}t + \sqrt{2}\mathrm{d}B_t, \tilde{\boldsymbol{x}}_t \sim \tilde{p}_t$. Thus, once the score function $\nabla \ln p_{T-t}$ is obtained, the reverse SDE induce a sampling algorithm.

**Discretization and realization of the reverse process.** To numerically solve the previous SDE, suppose $k := \lfloor t/\eta \rfloor$ for any $t \in [0, T]$, we approximate the score function $\nabla \ln p_{T-t}$ with $\boldsymbol{v}_k$ for $t \in [k\eta, (k+1)\eta]$. This modification results in a new SDE, given by

$$\mathrm{d}\tilde{\boldsymbol{x}}_t = (\tilde{\boldsymbol{x}}_t + \boldsymbol{v}_k(\tilde{\boldsymbol{x}}_{k\eta})) \mathrm{d}t + \sqrt{2}\mathrm{d}B_t, \quad t \in [k\eta, (k+1)\eta]. \tag{2}$$

Specifically, when $k = 0$, we set $\tilde{\boldsymbol{x}}_0$ to be sampled from $\tilde{p}_0$, which can approximate $p_T$. To find the closed form of the solution by setting an auxiliary random variable as $\mathbf{r}_i(\tilde{\boldsymbol{x}}_t, t) := \tilde{\boldsymbol{x}}_{t,i}e^{-t}$, we have

$$\mathrm{d}\mathbf{r}_i(\tilde{\boldsymbol{x}}_t, t) = \left( \frac{\partial \mathbf{r}_i}{\partial t} + \frac{\partial \mathbf{r}_i}{\partial \tilde{\boldsymbol{x}}_{t,i}} \cdot [\boldsymbol{v}_{k,i} + \tilde{\boldsymbol{x}}_{t,i}] + \mathrm{Tr}\left( \frac{\partial^2 \mathbf{r}_i}{\partial \tilde{\boldsymbol{x}}_{t,i}^2} \right) \right) \mathrm{d}t + \frac{\partial \mathbf{r}_i}{\partial \tilde{\boldsymbol{x}}_{t,i}} \cdot \sqrt{2}\mathrm{d}B_t$$

$$= \boldsymbol{v}_{k,i}e^{-t}\mathrm{d}t + \sqrt{2}e^{-t}\mathrm{d}B_t,$$

where the equalities are derived by the Itô's Lemma. Then, we set the initial value $\mathbf{r}_0(\tilde{\boldsymbol{x}}_t, 0) = \tilde{\boldsymbol{x}}_{t,i}$, and integral on both sides of the equation. We have

$$\tilde{\boldsymbol{x}}_{t+s} = e^s\tilde{\boldsymbol{x}}_t + (e^s - 1)\boldsymbol{v}_k + \mathcal{N}\left(0, \left(e^{2s} - 1\right)\boldsymbol{I}_d\right). \tag{3}$$

For the specific construction of $\boldsymbol{v}_k$ to approximate the score, we defer it to Section 3.

## 3 THE REVERSE DIFFUSION MONTE CARLO APPROACH

As shown in SDE (2), we introduce an estimator $\boldsymbol{v}_k \approx \nabla_{\boldsymbol{x}} \ln p_{T-t}(\boldsymbol{x})$ to implement the reverse diffusion. This section is dedicated to exploring viable estimators and the benefits of the reverse diffusion process. We found that we can reformulate the score as an expectation with the transition kernel of the forward OU process. We also derive the intuition that RDMC can reduce the isoperimetric dependence of the target distribution compared with conventional Langevin algorithm. Lastly, we introduce the detailed implementation of RDMC in real practice with different score estimators.

### 3.1 SCORE FUNCTION IS THE EXPECTATION OF THE POSTERIOR

We start with the formulation of $\nabla_{\boldsymbol{x}} \ln p_t(\boldsymbol{x})$. In general SDEs, the score functions $\nabla \ln p_t$ do not have an analytic form. However, our forward process is an OU process (SDE (1)) whose transition kernel is given as $p_{t|0}(\boldsymbol{x}|\boldsymbol{x}_0) = \left(2\pi\left(1 - e^{-2t}\right)\right)^{-d/2} \cdot \exp\left[\frac{-\|\boldsymbol{x} - e^{-t}\boldsymbol{x}_0\|^2}{2\left(1 - e^{-2t}\right)}\right]$. Such conditional density presents the probability of obtaining $\boldsymbol{x}_t = \boldsymbol{x}$ given $\boldsymbol{x}_0$ in SDE (1). Note that $p_t(\boldsymbol{x}) = \mathbb{E}_{p_0(\boldsymbol{x}_0)}p_{t|0}(\boldsymbol{x}|\boldsymbol{x}_0)$, we can use the property to derive other score formulations. Bayes' theorem demonstrates that the score can be reformulated as an expectation by the following Lemma.

**Lemma 1.** *Assume that Eq. (1) defines the forward process. The score function can be rewritten as*

$$\nabla_{\boldsymbol{x}} \ln p_{T-t}(\boldsymbol{x}) = \mathbb{E}_{\boldsymbol{x}_0 \sim q_{T-t}(\cdot|\boldsymbol{x})} \frac{e^{-(T-t)}\boldsymbol{x}_0 - \boldsymbol{x}}{(1 - e^{-2(T-t)})},$$

$$q_{T-t}(\boldsymbol{x}_0|\boldsymbol{x}) \propto \exp\left( -f_*(\boldsymbol{x}_0) - \frac{\left\|\boldsymbol{x} - e^{-(T-t)}\boldsymbol{x}_0\right\|^2}{2\left(1 - e^{-2(T-t)}\right)} \right). \tag{4}$$

---

**Algorithm 1** RDMC: reverse diffusion Monte Carlo

---

1: **Input:** Initial particle $\tilde{\boldsymbol{x}}_0$ sampled from $\tilde{p}_0$, Terminal time $T$, Step size $\eta, \eta'$, Sample size $n$.
2: **for** $k = 0$ to $\lfloor T/\eta \rfloor - 1$ **do**
3:     Set $\boldsymbol{v}_k = \boldsymbol{0}$;
4:     Create $n$ Monte Carlo samples to estimate

$$\boldsymbol{v}_k \approx \mathbb{E}_{\boldsymbol{x} \sim q_{T-t}} \left[ -\frac{\tilde{\boldsymbol{x}}_{k\eta} - e^{-(T-k\eta)}\boldsymbol{x}}{\left(1 - e^{-2(T-k\eta)}\right)} \right], \text{ where } q_{T-t}(\boldsymbol{x}|\tilde{\boldsymbol{x}}_{k\eta}) \propto \exp\left( -f_*(\boldsymbol{x}) - \frac{\left\| \tilde{\boldsymbol{x}}_{k\eta} - e^{-(T-k\eta)}\boldsymbol{x} \right\|^2}{2\left(1 - e^{-2(T-k\eta)}\right)} \right).$$

5:     $\tilde{\boldsymbol{x}}_{(k+1)\eta} = e^{\eta}\tilde{\boldsymbol{x}}_{k\eta} + (e^{\eta} - 1)\boldsymbol{v}_k + \xi$    where $\xi$ is sampled from $\mathcal{N}\left(0, \left(e^{2\eta} - 1\right)\boldsymbol{I}_d\right)$.
6: **end for**
7: **Return:** $\tilde{\boldsymbol{x}}_{\lfloor T/\eta \rfloor \eta}$ .

---

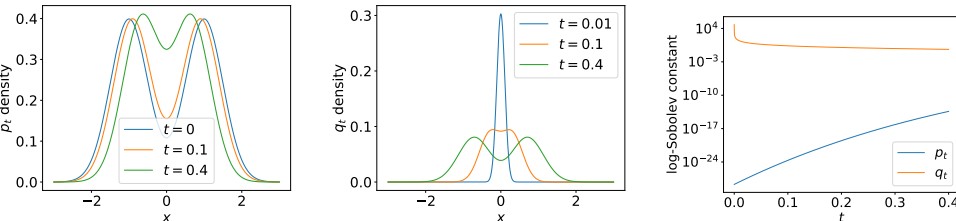

Figure 2: Illustrations of $p_t$, $q_t$, and their log-Sobolev (LSI) constants. The target distribution $p_*$ is a Gaussian mixture. We choose $q_t(\cdot|x = 0)$ for illustration. As $t$ increases, the modes of $p_t$ collapse to zero rapidly, corresponding to an improving LSI constant. For $q_t$, the barrier height of $q_t$ remains small, resulting in a relatively large LSI constant as long as $T = O(1)$. Thus initializing with $p_T$ and performing RDMC reduces computation complexity for multi-modal $p_*$.

The proof of Lemma 1 is presented in Appendix C.1. For any $t > 0$, we observe that $-\log q_{T-t}$ incorporates an additional quadratic term. In scenarios where $p_*$ adheres to the log-Sobolev inequality (LSI), this term enhances $q_{T-t}$'s log-Sobolev (LSI) constant, thereby accelerating convergence. Conversely, with heavy-tailed $p_*$ (where $f_*$'s growth is slower than a quadratic function), the extra term retains quadratic growth, yielding sub-Gaussian tails and log-Sobolev properties. Notably, as $t$ approaches $T$, the quadratic component becomes predominant, rendering $q_{T-t}$ strongly log-concave and facilitating sampling. In summary, every $q_{T-t}$ exhibits a larger LSI constant than $p_*$. As $t$ increases, this constant grows, ultimately leading $q_{T-t}$ towards strong convexity. Consequently, this provides a sequence of distributions with LSI constants surpassing those of $p_*$, enabling efficient score estimation for $\nabla \ln p_{T-t}$.

From Lemma 1, the expectation formula of $q_{T-t}(\cdot|\boldsymbol{x})$ can be obtained by empirical mean estimator with sufficient samples from $q_{T-t}(\cdot|\boldsymbol{x})$. Thus, the gradient complexity required in this sampling subproblem becomes the bottleneck of our algorithm. Suppose $\{\boldsymbol{x}_k^{(i)}\}$ is samples drawn from $q_{T-k\eta}(\cdot|\boldsymbol{x})$ when $\tilde{\boldsymbol{x}}_{k\eta} = \boldsymbol{x}$ for any $\boldsymbol{x} \in \mathbb{R}^d$, the construction of $\boldsymbol{v}_k(\boldsymbol{x})$ in Eq. (2) can be presented as

$$\boldsymbol{v}_k(\boldsymbol{x}) = \frac{1}{n_k} \sum_{i=1}^{n_k} \boldsymbol{v}_k^{(i)}(\boldsymbol{x}) \quad \text{where} \quad \boldsymbol{v}_k^{(i)}(\boldsymbol{x}) = 2\left(1 - e^{-2(T-k\eta)}\right)^{-1} \cdot \left(e^{-(T-k\eta)}\boldsymbol{x}_k^{(i)} - \boldsymbol{x}\right). \quad (5)$$

### 3.2 REVERSE SDE VS LANGEVIN DYNAMICS: INTUITION

From Figure 1, we observe that the RDMC method deviates significantly from the conventional gradient-based MCMC techniques, such as Langevin dynamics. It visualizes the paths from a Gaussian distribution to a mixture of Gaussian distributions. It can be observed that Langevin dynamics, being solely driven by the gradient information of the target density $p_*$, tends to get trapped in local regions, resulting in uneven sampling of the mixture of Gaussian distribution. More precisely, $p_*$ admits a small LSI constant due to the separation of the modes (Ma et al., 2019; Schlichting, 2019; Menz & Schlichting, 2014). Consequently, the convergence of conventional MCMC methods becomes notably challenging in such scenarios (Tosh & Dasgupta, 2014).

To better demonstrate the effect of our proposed SDE, we compute the LSI constant estimates for 1-$d$ case in Figure 2. Due to the shrinkage property of the forward process, the LSI constant of $p_t$

can be exponentially better than the original one. Meanwhile, for a $T = O(1)$, the LSI constant of $q_t$ is also well-behaved. In our algorithm, a well-conditioned LSI constant for both $p_T$ and $q_T$ reduces the computation overhead. Thus, we can choose a intermediate $T$ to connect those local modes and perform reverse SDE towards $p_*$. RDMC can distribute samples more evenly across different modes and enjoy faster convergence in those challenging cases. Moreover, if the growth rate of $-\log p_*$ is slower than the quadratic function (heavy tail), we can choose a large $T$ and use $p_\infty$ to estimate $p_T$. Then, all $-\log q_t$ have quadratic growth, which implies log-Sobolev property. These intuitions also explain why diffusion models excel in modeling complex distributions in high-dimensional spaces. We will provide the quantitative explanation in Section 4.

## 3.3 Algorithms for score estimation with $q_{T-t}(\cdot|x)$

According to the expectation form of scores shown in Lemma 1, a detailed reverse sampling algorithm can be naturally proposed in Algorithm 1. Specifically, it can be summarized as the following steps: (1) choose proper $T$ and $\tilde{p}_0$ such that $p_T \approx \tilde{p}_0$. This step can be done by either $\tilde{p}_0 = p_\infty$ for large $T$ or performing the ULA for $p_T$ (Algorithm 3 in Appendix); (2) sample from a distribution that approximate $q_{T-t}$ (Step 4 of Algorithm 1); (3) follow $\tilde{p}_t$ with the approximated $q_{T-t}$ samples (Step 5 of Algorithm 1); (4) repeat until $t \approx T$. After (4), we can also perform Langevin algorithm to fine-tune the steps when the gradient complexity limit is not reached.

The key of implementing Algorithm 1 is to estimate the scores $\nabla \ln p_{T-t}$ via Step 4 with samples from $q_{T-t}$. In what follows, we discuss the implementation that combines the importance weight sampling with the adjusted Langevin algorithm (ULA).

**Importance weighted score estimator.** We first consider importance sampling approach for estimating the score $\nabla \ln p_{T-t}$. From Eq. (4), we know that the key is to estimate:

$$\nabla_{\boldsymbol{x}} \ln p_{T-t}(\boldsymbol{x}) = \mathbb{E}_{\boldsymbol{x}_0 \sim q_{T-t}(\cdot|\boldsymbol{x})} \left[ \frac{e^{-(T-t)}\boldsymbol{x}_0 - \boldsymbol{x}}{(1 - e^{-2(T-t)})} \right] = \frac{1}{Z_*} \mathbb{E}_{\boldsymbol{x}_0 \sim \rho_{T-t}(\cdot|\boldsymbol{x})} \left[ \frac{e^{-(T-t)}\boldsymbol{x}_0 - \boldsymbol{x}}{(1 - e^{-2(T-t)})} \cdot e^{-f_*(\boldsymbol{x}_0)} \right],$$

and $Z_* = \mathbb{E}_{\boldsymbol{x}_0 \sim \rho_{T-t}(\cdot|\boldsymbol{x})} \left[ \exp(-f_*(\boldsymbol{x}_0)) \right]$, where $\rho_{T-t}(\cdot|\boldsymbol{x}) \propto \exp \left( \frac{-\|\boldsymbol{x} - e^{-(T-t)}\boldsymbol{x}_0\|^2}{2(1 - e^{-2(T-t)})} \right)$. Note that sampling from $\rho_{T-t}$ takes negligible computation resource. The main challenge is the sample complexity of estimating the two expectations.

Since $\rho_{T-t}(\boldsymbol{x}_0|\boldsymbol{x})$ is Gaussian with variance $\sigma_t^2 = \frac{1-e^{-2(T-t)}}{e^{-2(T-t)}}$, we know that as long as $-\frac{\boldsymbol{x} - e^{-(T-t)}\boldsymbol{x}_0}{(1 - e^{-2(T-t)})} \cdot \exp(-f_*(\boldsymbol{x}_0))$ is $G$-Lipschitz, the sample complexity scales as $N = \tilde{O}\left( \sigma_t^2 \frac{G^2}{\epsilon^2} \right)$ for the resulting errors of the two mean estimators to be less than $\epsilon$ (Wainwright, 2019). However, the sample size required of the importance sampling method to achieve an overall small error is known to scale exponentially with the KL divergence between $\rho_{T-t}$ and $q_{T-t}$ (Chatterjee & Diaconis, 2018), which can depend on the dimension. In our current formulation, this is due to the fact that the true denominator $Z_* = \mathbb{E}_{\boldsymbol{x}_0 \sim \rho_{T-t}(\cdot|\boldsymbol{x})}[e^{-f_*(\boldsymbol{x}_0)}]$ can be as little as $\exp(-d)$. As a result, to make the overall score estimation error small, the error tolerance and in turn the sample size required for estimating $Z_*$ can scale exponentially with the dimension.

**ULA score estimator.** An alternative score estimator considers that the mean of the underlying distribution $q'_{T-t}(\cdot|\boldsymbol{x})$ of these samples needs to sufficiently approach $q_{T-t}(\cdot|\boldsymbol{x})$, which can be achieved by closing the gap of KL divergence or Wasserstein distance between $q'_{T-t}(\cdot|\boldsymbol{x})$ and $q_{T-t}(\cdot|\boldsymbol{x})$. Since the additional quadratic term shown in Eq. (4) helps improve a quadratic tail growth for $q_{T-t}(\cdot|\boldsymbol{x})$, which implies the establishment of the isoperimetric condition. We expect the convergence can be achieved by performing the ULA on a sampling subproblem whose target distribution is $q_{T-t}(\cdot|\boldsymbol{x})$. We provide the detailed implementation in Algorithm 2.

**ULA score estimator with importance sampling initialization.** Inspired by the previous estimators, we can combine the importance sampling approach with the ULA. In particular, we first implement the importance sampling method to form a rough score estimator. We then perform ULA at the mean estimator and obtain a refined accurate score estimate. Via this combination, we are able to efficiently obtain accurate score estimation by virtue of the ULA algorithm when $t$ is close to $T$. When $t$ is close to 0, we are able to quickly obtain rough score estimates via the importance sampling approach. We discover empirically that this combination generally perform well when $t$ interpolates the two regimes (Figure 3 in Section 4).

---

**Algorithm 2** ULA inner-loop for the $q_t(\cdot|\boldsymbol{x})$ sampler (Step 4 of Algorithm 1)

1: **Input:** Condition $\boldsymbol{x}$ and time $t$, Sample size $n$, Initial particles $\{\boldsymbol{x}_0^i\}_{i=1}^n$, Iters $K$, Step size $\eta$.
2: **for** $k = 1$ to $K$ **do**
3:     **for** $i = 1$ to $n$ **do**
4:        $\boldsymbol{x}_k^i = \boldsymbol{x}_{k-1}^i - \eta\left(\nabla f_*(\boldsymbol{x}_{k-1}^i) + \frac{e^{-t}\left(e^{-t}\boldsymbol{x}_{k-1}^i - \boldsymbol{x}\right)}{1 - e^{-2t}}\right) + \sqrt{2\eta}\xi_k$, where $\xi_k \sim \mathcal{N}\left(0, \boldsymbol{I}_d\right)$
5:     **end for**
6: **end for**
7: **Return:** $\{\boldsymbol{x}_K^i\}_{i=1}^n$ .

---

# 4 ANALYSES AND EXAMPLES OF THE RDMC APPROACH

In this section, we analyze the overall complexity of the RDMC via ULA inner loop estimation. Since the complexity of the importance sampling estimate is discussed in Section 3.3, we only consider Algorithm 1 with direct ULA sampling of $q_{T-t}(\cdot|\boldsymbol{x})$ rather than the smart importance sampling initialization to make our analysis clear.

To provide the analysis of the convergence of RDMC, we make the following assumptions.

[$\mathbf{A_1}$] For all $t \geq 0$, the score $\nabla \ln p_t$ is $L$-Lipschitz.

[$\mathbf{A_2}$] The second moment of the target distribution $p_*$ is upper bounded, i.e., $\mathbb{E}_{p_*}\left[\|\cdot\|^2\right] = m_2^2$.

These assumptions are standard in diffusion analysis to guarantee the convergence to the target distribution (Chen et al., 2022a). Specifically, Assumption [$\mathbf{A_1}$] governs the smoothness characteristics of the forward process, which ensure the feasibility of numerical discretization methods used for approximating the solution of continuous SDE. In addition, Assumption [$\mathbf{A_2}$] introduces essential constraints on the moments of the target distribution. These constraints effectively prevent an excessive accumulation of probability mass in the tail region, thereby ensuring a more balanced and well-distributed target distribution.

## 4.1 OUTER LOOP COMPLEXITY

According to Algorithm 1, the overall gradient complexity depends on the number of outer loops $k$, as well as the complexity to achieve accurate score estimations (Line 5 in Algorithm 1). When the score is well-approximated and satisfies $\mathbb{E}_{p_{T-k\eta}}\left[\|\boldsymbol{v}_k - \nabla \ln p_{T-k\eta}\|^2\right] \leq \epsilon^2$, the overall error in TV distance, $D_{\mathrm{TV}}(p_*, \tilde{p}_T)$, can be made $\tilde{\mathcal{O}}(\epsilon)$ by choosing a small $\eta$. Under this condition, the number of outer loops satisfies $k = \Omega(L^2 d\epsilon^{-2})$, which shares the same complexity as that in diffusion analysis (Chen et al., 2022a; Lee et al., 2022; 2023). Such a complexity of the score estimation oracles is independent of the log-Sobolev (LSI) constant of the target density $p_*$, which means that the isoperimetric dependence of RDMC is dominated by the subproblem of sampling from $q_{T-t}$. Specifically, the following theorem demonstrates the conditions for achieving $D_{\mathrm{TV}}(p_*, \tilde{p}_T) = \tilde{\mathcal{O}}(\epsilon)$.

**Theorem 1.** *For any $\epsilon > 0$, assume that $D_{\mathrm{KL}}(p_T\|\tilde{p}_0) < \epsilon$ for some $T > 0$, $\hat{p}$ as suggested in Algorithm 1, $\eta = C_1(d + m_2^2)^{-1}\epsilon^2$. If the OU process induced by $p_*$ satisfies Assumption [$\mathbf{A_1}$], [$\mathbf{A_2}$], and $q_{T-k\eta}$ satisfies the log-Sobolev inequality (LSI) with constant $\mu_k$ ($k\eta \leq T$). We set*

$$n_k = 64Td\mu_k^{-1}\eta^{-3}\epsilon^{-2}\delta^{-1}, \quad \mathcal{E}_k = 2^{-13} \cdot T^{-4}d^{-2}\mu_k^2\eta^8\epsilon^4\delta^4 \tag{6}$$

*where $n_k$ is the number of samples to estimate the score, and $\mathcal{E}_k$ is the KL error tolerance for the inner loop. With probability at least $1 - \delta$, Algorithm 1 converges in TV distance with $\tilde{\mathcal{O}}(\epsilon)$ error.*

Therefore, the key points for the convergence of RDMC can be summarized as

- The LSI of target distributions of sampling subproblems, i.e., $q_{T-k\eta}$ is maintained.
- The initialization of the reverse process $\tilde{p}_0$ is sufficiently close to $p_T$.

## 4.2 OVERALL COMPUTATION COMPLEXITY

In this section, we consider the overall computation complexity of RDMC. Note that the LSI constants of $q_{T-k\eta}$ depend on the properties of $p_*$. As a result we consider more specific assumptions

that bound the LSI constants for $q_{T-k\eta}$. In particular, we demonstrate the benefits of using $q_{T-k\eta}$ for targets $p_*$ with infinite or exponentially large LSI constants. Compared with ULA, RDMC improves the gradient complexity, due to the improved LSI constant of $q_{T-k\eta}$ over that of $p_*$ in finite $T$. Even for heavy-tailed $p_*$ that does not satisfy the Poincaré inequality, target distributions of sampling subproblems, i.e., $q_{T-k\eta}$, can still preserve the LSI property, which helps RDMC to alleviate the exponential dimension dependence of gradient complexity for achieving TV distance convergence.

Specifically, Section 4.2.1 consider the case that the LSI constant of $p_*$ depend on the radius $R$ exponentially, which can usually be found in mixture models. Our proposed RDMC can reduce the exponent by a factor. Section 4.2.2 consider the case that $p_*$ is not LSI, but RDMC can create an LSI subproblem sequence which makes the dimension dependency polynomial.

We first provide an estimate for the LSI constant of $q_t$ under general Lipschitzness assumption [A$_1$].

**Lemma 2.** *Under [A$_1$], the LSI constant for $q_t$ in the forward OU process is $\frac{e^{-2t}}{2(1-e^{-2t})}$ when $0 \leq t \leq \frac{1}{2}\ln\left(1 + \frac{1}{2L}\right)$. This estimation indicate that when quadratic term dominate the log-density of $q_t$, the log-Sobolev property is well-guaranteed.*

### 4.2.1 IMPROVING THE LSI CONSTANT DEPENDENCE FOR MIXTURE MODELS

Apart from Assumption [A$_1$], [A$_2$], we study the case where $p_*$ satisfies the following assumption:

[A$_3$] There exists $R > 0$, such that $f_*(\boldsymbol{x})$ is $m$-strongly convex when $\|\boldsymbol{x}\| \geq R$.

In this case, the target density $p_*$ admits an LSI constant which scales exponentially with the radius of the region of nonconvexity $R$, i.e., $\frac{m}{2}\exp(-16LR^2)$, as shown in (Ma et al., 2019). This implies that if we draw samples from $p_*$ with ULA, the gradient complexity will have exponential dependency with respect to $R^2$. However, in RDMC with suitable choices of $T = O(\log R)$ and $\tilde{p}_0$, the exponential dependency on $R$ is removed, which is a bottleneck for mixture models.

In the Proposition 1 below, we select values of $T$ and $\tilde{p}_0$ to achieve the desired level of accuracy. Notably, Lemma 2 suggest that we can choose a $O(1)$ termination time to make the LSI constant of $q_t$ well-behehaved and $p_T$ is much simpler than $p_0$. That is to say, RDMC can exhibit lower isoperimetric dependency compared to conventional MCMC techniques. Thus, We find that the overall computation complexity of RDMC reduces the dependency on $R$.

**Proposition 1.** *Assume that the OU process induced by $p_*$ satisfies [A$_1$], [A$_2$], [A$_3$]. We estimate $p_T$ with $\tilde{p}_0$ by ULA, where the iterations wrt LSI constant is $\Omega\left(m^{-1}\exp(16LR^2e^{-2T} - 2T)\right)$. For any $\epsilon > 0$ and $T \leq \frac{1}{2}\ln\left(1 + \frac{1}{2L}\right)$, the Algorithm 1 from $\tilde{p}_0$ to target distribution convergence with a probability at least $1 - \delta$ and total gradient complexity $\Omega\left(\mathrm{poly}(\epsilon, \delta)\right)$ independent of $R$.*
*Example: Gaussian Mixture Model.* We consider an example that

$$p_*(x) \propto \exp\left(-\frac{1}{2}\|x\|^2\right) + \exp\left(-\frac{1}{2}\|x - y\|^2\right), \quad y \gg 1.$$

The LSI constant is $\Theta(e^{-C_0\|y\|^2})$ (Ma et al., 2019; Schlichting, 2019; Menz & Schlichting, 2014), corresponding to the complexity of the target distribution. Tosh & Dasgupta (2014) prove that the lower bound iteration complexity by MCMC-based algorithm to sample from the Gaussian mixture scales exponentially with the squared distance $\|y\|^2$ between the two modes: $\Omega(\exp(\|y\|^2/8))$. Note that RDMC is not a type of conventional MCMC. With importance sampling score estimator, the $\tilde{O}(\epsilon^{-2})$ iteration complexity and $\tilde{O}(\epsilon^{-2})$ samples at each step, the TV distance converges with $\tilde{O}(\epsilon)$ error (Appendix B).

The computation overhead of the outer-loop process does not depend on $y$. For the inner-loop score estimation we can choose $T = \frac{1}{2}\log\frac{3}{2}$ to make the LSI constant of $q_t$ to be $O(1)$. We can perform ULA to initialize $p_T$, which reduces the barrier between modes significantly. Specifically, the LSI constant of $p_T$ is $\Theta(e^{-C_0\|e^{-T}y\|^2})$, which improves the dependence on $\|y\|^2$ by a $e^{-2T} = \frac{2}{3}$ factor. Since this factor is on the exponent, the reduction is exponential.

Figure 3 is a demonstration for this example. We choose different $r$ to represent the change of $R$ in [A$_3$]. We compare the convergence of Langevin Monte Carlo (LMC), Underdamped LMC (ULMC) and RDMC in terms of gradient complexity. As $r$ increases, we find that LMC fails to converge within a limited time. ULMC, with the introduction of velocity, can alleviate this situation. Notably, our algorithm can still ensure fast mixing for large $r$. The inner loop is initialized with importance sampling mean estimator. Hyper-parameters and more results are provided in Appendix F.

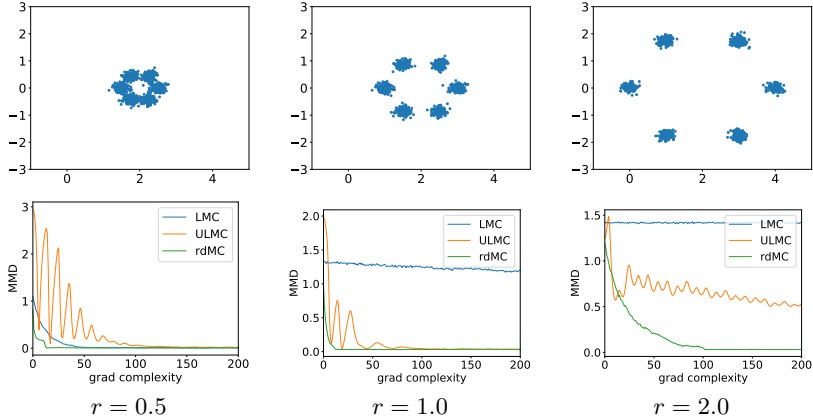

Figure 3: Maximum Mean Discrepancy (MMD) convergence of LMC, ULMC, RDMC. *First row* shows different target distributions, with increasing mode separation $r$ and barrier heights, leading to reduced log-Sobolev (LSI) constants. *Second row* displays the algorithms' convergence, revealing RDMC's pronounced advantage convergence compared to ULMC/LMC, especially for large $r$.

### 4.2.2 IMPROVING THE DIMENSION DEPENDENCE IN HEAVY-TAILED TARGET DISTRIBUTIONS

In this subsection, we study the case where $p_*$ satisfies Assumption [$\mathbf{A}_1$], [$\mathbf{A}_2$], and

[$\mathbf{A}_4$] For any $r > 0$, we can find some $R(r)$ satisfying $f_*(\boldsymbol{x}) + r\|\boldsymbol{x}\|^2$ is convex for $\|\boldsymbol{x}\| \geq R(r)$. Without loss of generality, we suppose $R(r) = c_R/r^n$ for some $n > 0, c_R > 0$.

Assumption [$\mathbf{A}_4$] can be considered as a soft version of [$\mathbf{A}_3$]. Specifically, it permits the tail growth of the negative log-density $f_*$ to be slower than quadratic functions. This encompasses certain target distributions that fall outside the constraints of LSI and PI. Furthermore, given that the additional quadratic term present in Eq. (4) dominates the tail, $q_{T-k\eta}$ satisfies LSI for all $t > 0$.

**Lemma 3.** *Under [$\mathbf{A}_1$], [$\mathbf{A}_4$], the LSI constant for $q_t$ in the forward OU process is $\frac{e^{-2t}}{6(1-e^{-2t})}$ · $e^{-16 \cdot 3L \cdot R^2 \left( \frac{e^{-2t}}{6(1-e^{-2t})} \right)}$ for any $t \geq 0$. The tail property guarantees a uniform LSI constant.*

The uniform bound on the LSI constant enables us to estimate the score for any $p_t$. We can consider cases that are free from the constraints on the properties of $p_T$ and let $T$ be sufficiently large. By setting $T$ at $\tilde{\mathcal{O}}(\ln 1/\epsilon)$ level, we can approximate $p_T$ with $p_\infty$ — the stationary distribution of the forward process. Furthermore, since $q_t$ are log-Sobolev, we can perform RDMC to sample from heavy-tailed $p^*$ in the absence of a log-Sobolev inequality. The explicit computational complexity of RDMC, needed to converge to any specified accuracy, is detailed in the subsequent proposition.

**Proposition 2.** *Assume that the target distribution $p_*$ satisfies Assumption [$\mathbf{A}_1$], [$\mathbf{A}_2$], and [$\mathbf{A}_4$]. We take $p_\infty$ to be $\tilde{p}_0$. For any $\epsilon > 0$, by performing Algorithm 1 with ULA inner loop and hyper-parameters in Theorem 1, with a probability at least $1 - \delta$, we have $D_{\text{TV}}(\tilde{p}_t, p_*) \leq \tilde{O}(\epsilon)$ with $\Omega\left(d^{18}\epsilon^{-16n-83}\delta^{-6}\exp\left(\epsilon^{-16n}\right)\right)$ gradient complexity.*

*Example: Potentials with Sub-Linear Tails.* We consider an example that

$$p_*(x) \propto \exp\left(-\left(\|x\|^2 + 1\right)^a\right) \quad \text{where} \quad a \in (0, 0.5).$$

Lemma 5 demonstrates that this $p_*$ satisfies Assumption [$\mathbf{A}_4$] with $n = (2 - 2a)^{-1} \leq 1$. Moreover, these target distributions with sub-linear tails also satisfy weak Poincare inequality (wPI) introduced in Mousavi-Hosseini et al. (2023), in which the dimension dependence of ULA to achieve the convergence in TV distance is proven to be $\tilde{\Omega}(d^{4a^{-1}+3})$. Compared with this result, the complexity of RDMC shown in Proposition 2 has a lower dimension dependence when $a \leq 4/15$.

**Conclusion.** This paper presents a novel sampling algorithm based on the reverse SDE of the OU process. The algorithm efficiently generates samples by utilizing the mean estimation of a sub-problem to estimate the score function. It demonstrates convergence in terms of total variation distance and proves efficacy in general sampling tasks with or without isoperimetric conditions. The algorithm exhibits lower isoperimetric dependency compared to conventional MCMC techniques, making it well-suited for multi-modal and high-dimensional challenging sampling. The analysis provides insights into the complexity of score estimation within the OU process, given the conditional posterior distribution.

ACKNOWLEDGEMENTS

The research is partially supported by the NSF awards: SCALE MoDL-2134209, CCF-2112665 (TILOS). It is also supported in part by the DARPA AIE program, the U.S. Department of Energy, Office of Science, the Facebook Research Award, as well as CDC-RFA-FT-23-0069 from the CDC's Center for Forecasting and Outbreak Analytics.

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

## A  PROOF SKETCH

For better understanding for our paper, we provide a proof sketch as below.

Lemma 1 is a direct result of Bayes theorem (Appendix C.1), which is the main motivation of our algorithm, that estimate the score with samples from $q$.

Our main contribution is to prove the convergence and evaluate the complexity of Algorithm 1, where the inner loop is performed by Algorithm 2.

Our analysis is based on the TV distance[1], where we use data-processing inequality, triangle inequality, and Pinsker's inequality (refer to Eq. (19) for details) to provide the upper bound as below

$$D_{\mathrm{TV}}\left(\tilde{p}_T, p_*\right) \le \underbrace{\sqrt{\frac{1}{2}D_{\mathrm{KL}}\left(\tilde{p}_0\|p_T\right)}}_{\text{Term 1}} + \underbrace{\sqrt{\frac{1}{2}D_{\mathrm{KL}}\left(\hat{P}_T\|\tilde{P}_T^{p_T}\right)}}_{\text{Term 2}},$$

where Term 1 is the error between $\tilde{p}_0$ and $p_T$ and Team 2 is the score estimation loss of the whole trajectory.

Theorem 1 considers the case that all $q_{T-k\eta}$ is log-Sobolev with constant $\mu_k$ $(k\eta \le T)$, where the log-Sobolev constants can further be estimated with additional assumptions on $p_*$.

By definition,

$$2(\text{Term 2})^2 = \sum_{k=0}^{N-1} \int_{k\eta}^{(k+1)\eta} \mathbb{E}_{\hat{P}_T}\left[\frac{1}{4} \cdot \|\mathbf{v}_k(\mathbf{x}_{k\eta}) - 2\nabla \ln p_{T-t}(\mathbf{x}_t)\|^2\right] \mathrm{d}t,$$

where Term 2 is defined by an integration.

To bound the error between integration and discretized algorithm, we have Lemma 7 that when $\eta = O(\epsilon^2)$

$$\frac{1}{4} \cdot \mathbb{E}_{\hat{P}_T}\left[\|\mathbf{v}_k(\mathbf{x}_{k\eta}) - 2\nabla \ln p_{T-t}(\mathbf{x}_t)\|^2\right] \le 4\epsilon^2 + \frac{1}{2} \cdot \underbrace{\mathbb{E}_{\hat{P}_T}\left[\|\mathbf{v}_k(\mathbf{x}_{k\eta}) - 2\nabla \ln p_{T-k\eta}(\mathbf{x}_{k\eta})\|^2\right]}_{\epsilon_{\text{score}}}$$

(7)

Note that the $\epsilon_{\text{score}}$ can be controlled by

$$2\underbrace{\left\|\mathbf{v}_k(\boldsymbol{x}) - 2\mathbb{E}_{\mathbf{x}_0'\sim q_{T-k\eta}'(\cdot|\boldsymbol{x})}\left[-\frac{\boldsymbol{x} - e^{-(T-k\eta)}\mathbf{x}_0'}{\left(1 - e^{-2(T-k\eta)}\right)}\right]\right\|^2}_{\text{Term 2.1}}$$

and

$$2\underbrace{\left\|\frac{2e^{-(T-k\eta)}}{1 - e^{-2(T-k\eta)}}\left[\mathbb{E}_{\mathbf{x}_0'\sim q_{T-k\eta}'(\cdot|\boldsymbol{x})}[\mathbf{x}_0'] - \mathbb{E}_{\mathbf{x}_0\sim q_{T-k\eta}(\cdot|\boldsymbol{x})}[\mathbf{x}_0]\right]\right\|^2}_{\text{Term 2.2}},$$

where $q'$ is the estimated inner loop distribution by ULA.

Lemma 8 provide the concentration for Term 2.1.

We also have

$$\text{Term 2.2} \le 8\eta^{-2}C_{\mathrm{LSI},k}^{-1}D_{\mathrm{KL}}\left(q_{T-k\eta}'(\cdot|\boldsymbol{x})\|q_{T-k\eta}(\cdot|\boldsymbol{x})\right),$$

where the KL divergence is controlled by the convergence of ULA (Lemma 9). Thus, the desired convergence can be obtained.

Note that Theorem 1 is based on the fact that all $q_{T-k\eta}$ are log-Sobolev. So we aim to estimate the log-Sobolev constants for every $q_{T-k\eta}$.

Lemma 2 and 3 provide two approaches to estimate the log-Sobolev constant for different time steps.

---

[1] Please refer to Table 1 for the notation definitions.

When $p_*$ is strongly convex outside a ball with radius $R$, we can derive that $p_T$ can reduce the radius so that improve the log-Sobolev constant. Thus, we can choose proper $T$ where $p_T$ has larger log-Sobolev constant than $p_0$ and $q_t$ are strongly convexity so that the log-Sobolev constants are independent of $R$, which makes the algorithm mix fast.

For more general distributions without log-Sobolev constant, Lemma 10 provide the log-Sobolev constant the whole trajectory, so that the computation complexity can be obtained.

# B  NOTATIONS AND DISCUSSIONS

## B.1  ALGORITHM

---

**Algorithm 3** Initialization of $\hat{p}$ if $\hat{p} \neq p_\infty$

---

1: **Input:** Initial particle $\boldsymbol{x}_0$, OU process terminate time $T$, Iters $T_0$, Step size $\eta_0$, Sample size $n$.
2: **for** $k = 1$ to $T_0$ **do**
3:     Create $n_k$ Monte Carlo samples to estimate

$$\mathbb{E}_{\boldsymbol{x} \sim q_t} \left[ -\frac{\boldsymbol{x}_{k-1} - e^{-T} \boldsymbol{x}}{(1 - e^{-2T})} \right], \text{ s.t. } q_t(\boldsymbol{x}|\boldsymbol{x}_{k-1}) \propto \exp \left( -f_*(\boldsymbol{x}) - \frac{\left\| \boldsymbol{x}_{k-1} - e^{-T} \boldsymbol{x} \right\|^2}{2 \left( 1 - e^{-2T} \right)} \right).$$

4:     Compute the corresponding estimator $\boldsymbol{v}_k$.
5:     $\boldsymbol{x}_k = \boldsymbol{x}_{k-1} + \eta_0 \boldsymbol{v}_k + \sqrt{2\eta_0} \xi$   where $\xi$ is sampled from $\mathcal{N}\left(0, \boldsymbol{I}_d\right)$.
6: **end for**
7: **Return:** $\boldsymbol{x}_{T_0}$.

---

## B.2  NOTATIONS

According to Cattiaux et al. (2021), under mild conditions in the following forward process $\mathrm{d}\mathbf{x}_t = b_t(\mathbf{x}_t)\mathrm{d}t + \sqrt{2}\mathrm{d}B_t$, the reverse process also admits an SDE description. If we fix the terminal time $T > 0$ and set $\hat{\mathbf{x}}_t = \mathbf{x}_{T-t}$,   for $t \in [0, T]$, the process $(\hat{\mathbf{x}}_t)_{t \in [0,T]}$ satisfies the SDE $\mathrm{d}\hat{\mathbf{x}}_t = \hat{b}_t(\hat{\mathbf{x}}_t)\mathrm{d}t + \sqrt{2}\mathrm{d}B_t$, where the reverse drift satisfies the relation $b_t + \hat{b}_{T-t} = 2\nabla \ln p_t, \mathbf{x}_t \sim p_t$. In this condition, the reverse process of SDE (1) is as follows

$$\mathrm{d}\hat{\mathbf{x}}_t = (\hat{\mathbf{x}}_t + 2\nabla \ln p_{T-t}(\hat{\mathbf{x}}_t)) \, \mathrm{d}t + \sqrt{2}\mathrm{d}B_t. \tag{8}$$

Thus, once the score function $\nabla \ln p_{T-t}$ is obtained, the reverse SDE induce a sampling algorithm. To obtain the particles along SDE (8), the first step is to initialize the particle with a tractable starting distribution. In real practice, it is usually hard to sample from the ideal initialization $p_T$ directly due to its unknown properties. Instead, we sample from an approximated distribution $\hat{p}$. For large $T$, $p_\infty$ is chosen for approximating $p_T$ as their gap can be controlled. For the iteration, we utilize the numerical discretization method, i.e., DDPM Ho et al. (2020), widely used in diffusion models' literature. Different from ULA, DDPM divides SDE (8) by different time segments, and consider the following SDE for each segment

$$\mathrm{d}\bar{\mathbf{x}}_t = (\bar{\mathbf{x}}_t + 2\nabla \ln p_{T-k\eta}(\bar{\mathbf{x}}_{k\eta})) \, \mathrm{d}t + \sqrt{2}\mathrm{d}B_t, \quad t \in [k\eta, (k+1)\eta], \quad \bar{\mathbf{x}}_0 \sim p_T \tag{9}$$

to discretize SDE (8). Suppose we obtain $\mathbf{v}_k$ to approximate $2\nabla \ln p_{T-k\eta}$ at each iteration. Then, we obtain the SDE 2 for practical updates of particles.

**Reiterate of Algorithm 1**   Once the score function can be estimated, the sampling algorithm can be presented. The detailed algorithm is described in Algorithm 1. By following these steps, our proposed algorithm efficiently addresses the given problem and demonstrates its effectiveness in practical applications. Specifically, we summarize our algorithm as below: (1) choose proper $T$ and proper $\hat{p}$ such that $p_T \approx \hat{p}$. This step can be done by either $\hat{p} = p_\infty$ for large $T$ or performing the Langevin Monte Carlo for $p_T$; (2) sample from $\hat{p}$; (3) sample from a distribution that approximate $q_{T-t}$; (4) update $\tilde{p}_t$ with the approximated $q_{T-t}$ samples. This step can be done by Langevin Monte Carlo inner loop, as $\nabla \log q_{T-t}$ is explicit; (5) repeat (3) and (4) until $t \approx T$. After (5), we can also

| Constant | Value | Constant | Value |
|---|---|---|---|
| $C_0$ | $D_{\mathrm{KL}}\left(p_0\|p_\infty\right)$ | $C_1$ | $2^{-16}\cdot L^{-2}$ |
| $C_2$ | $48L\cdot 6^{2n}\cdot 2^{-8n}\cdot C_0^{8n}$ | $C_3$ | $64\cdot C_0 C_1^{-3} C_6$ |
| $C_4$ | $2^{-13}\cdot C_0^{-4} C_1^8 C_6^{-2}$ | $C_5$ | $2^8\cdot 3^4\cdot 5^2 L^2\cdot C_2 C_4^{-1} C_6^2 \ln\left(2^{-4} L^2 C_0^4 C_4^{-1} C_6\right)$ |
| $C_6$ | $6\cdot 2^{-4}\cdot C_0^4$ | $C_5'$ | $2^{22}\cdot 3^4\cdot 5\cdot L^{-2} C_0^4 C_1^{-8} \ln\left(\frac{2^8}{L}\cdot C_0^8 C_1^{-8}\right)$ |
| $C_3'$ | $64 L^{-1}\cdot C_0 C_1^{-3}$ | | |

Table 2: Constant List

perform Langevin algorithm to fine-tune the steps when the gradient complexity limit is not reached. The main algorithm as Algorithm 1.

Here, we reiterate our notation in Table 1 and the definition of log-Sobolev inequality as follows.

**Definition 1.** *(Logarithmic Sobolev inequality) A distribution with density function $p$ satisfies the log-Sobolev inequality with a constant $\mu > 0$ if for all smooth function $g\colon \mathbb{R}^d \to \mathbb{R}$ with $\mathbb{E}_p[g^2] < \infty$,*

$$\mathbb{E}_{p_*}\left[g^2 \ln g^2\right] - \mathbb{E}_{p_*}\left[g^2\right] \ln \mathbb{E}_{p_*}\left[g^2\right] \le \frac{2}{\mu}\mathbb{E}_{p_*}\left[\|\nabla g\|^2\right]. \tag{10}$$

Table 1: Notation List

| Symbols | Description |
|---|---|
| $\varphi_{\sigma^2}$ | The density function of the centered Gaussian distribution, i.e., $\mathcal{N}\left(\mathbf{0}, \sigma^2 \boldsymbol{I}\right)$. |
| $p_*, p_0$ | The target density function (initial distribution of the forward process) |
| $(\mathbf{x}_t)_{t\in[0,T]}$ | The forward process, i.e., SDE (1) |
| $p_t$ | The density function of $\mathbf{x}_t$, i.e., $\mathbf{x}_t \sim p_t$ |
| $p_\infty$ | The density function of stationary distribution of the forward process. |
| $(\hat{\mathbf{x}}_t)_{t\in[0,T]}$ | The ideal reverse process, i.e., SDE (8) |
| $\hat{p}_t$ | The density function of $\hat{\mathbf{x}}_t$, i.e., $\hat{\mathbf{x}}_t \sim \hat{p}_t$ and $p_t = \hat{p}_{T-t}$ |
| $\hat{P}_T$ | The law of the ideal reverse process SDE (8) over the path space $\mathcal{C}\left([0,T];\mathbb{R}^d\right)$. |
| $(\bar{\mathbf{x}}_t)_{t\in[0,T]}$ | The reverse process following from SDE (9) |
| $\bar{p}_t$ | The density function of $\bar{\mathbf{x}}_t$, i.e., $\bar{\mathbf{x}}_t \sim \bar{p}_t$ |
| $(\tilde{\mathbf{x}}_t)_{t\in[0,T]}$ | The practical reverse process following from SDE (2) with initial distribution $q$ |
| $\tilde{p}_t$ | The density function of $\tilde{\mathbf{x}}_t$, i.e., $\tilde{\mathbf{x}}_t \sim \tilde{p}_t$ |
| $\tilde{P}_T$ | The law of the reverse process $(\tilde{\mathbf{x}}_t)_{t\in[0,T]}$ over the path space $\mathcal{C}\left([0,T];\mathbb{R}^d\right)$. |
| $(\tilde{\mathbf{x}}_t^{p_T})_{t\in[0,T]}$ | The reverse process following from SDE (2) with initial distribution $p_T$ |
| $\tilde{p}_t^{p_T}$ | The density function of $\tilde{\mathbf{x}}_t$, i.e., $\tilde{\mathbf{x}}_t^{p_T} \sim \tilde{p}_t^{p_T}$ |
| $\tilde{P}_T^{p_T}$ | The law of the reverse process $(\tilde{\mathbf{x}}_t^{p_T})_{t\in[0,T]}$ over the path space $\mathcal{C}\left([0,T];\mathbb{R}^d\right)$. |

Besides, there are many constants used in our proof. We provide notations here to prevent confusion.

### B.3 EXAMPLES

**Lemma 4.** *(Proof of the Gaussian Mixture example) The iteration and sample complexity of rdMC with importance sampling estimator is $O(\epsilon^{-2})$.*

*Proof.* Similar to Eq. (19) in Theorem 1, we have

$$D_{\text{TV}}\left(\tilde{p}_T, p_*\right) \leq \underbrace{\sqrt{\frac{1}{2}D_{\text{KL}}\left(\tilde{p}_0\|p_T\right)}}_{\text{Term 1}} + \underbrace{\sqrt{\frac{1}{2}D_{\text{KL}}\left(\hat{P}_T\|\tilde{P}_T^{p_T}\right)}}_{\text{Term 2}}. \tag{11}$$

by using data-processing inequality, triangle inequality, and Pinsker's inequality.

To ensure Term 1 controllable, we choose $T = 2\ln\frac{D_{\text{KL}}(p_*\|p_\infty)}{2\epsilon^2}$.

For Term 2,

$$D_{\text{KL}}\left(\hat{P}_T\|\tilde{P}_T^{p_T}\right) = \mathbb{E}_{\hat{P}_T}\left[\ln\frac{\mathrm{d}\hat{P}_T}{\mathrm{d}\tilde{P}_T^{p_T}}\right] = \frac{1}{4}\sum_{k=0}^{N-1}\mathbb{E}_{\hat{P}_T}\left[\int_{k\eta}^{(k+1)\eta}\|\mathbf{v}_k(\mathbf{x}_{k\eta}) - 2\nabla\ln p_{T-t}(\mathbf{x}_t)\|^2\,\mathrm{d}t\right]$$

$$= \sum_{k=0}^{N-1}\int_{k\eta}^{(k+1)\eta}\mathbb{E}_{\hat{P}_T}\left[\frac{1}{4}\cdot\|\mathbf{v}_k(\mathbf{x}_{k\eta}) - 2\nabla\ln p_{T-t}(\mathbf{x}_t)\|^2\right]\mathrm{d}t,$$

By Lemma 7,

$$\frac{1}{4}\cdot\mathbb{E}_{\hat{P}_T}\left[\|\mathbf{v}_k(\mathbf{x}_{k\eta}) - 2\nabla\ln p_{T-t}(\mathbf{x}_t)\|^2\right] \leq 4\epsilon^2 + \frac{1}{2}\cdot\underbrace{\mathbb{E}_{\hat{P}_T}\left[\|\mathbf{v}_k(\mathbf{x}_{k\eta}) - 2\nabla\ln p_{T-k\eta}(\mathbf{x}_{k\eta})\|^2\right]}_{\epsilon_{\text{score}}} \tag{12}$$

Thus, the iteration complexity of the RDS algorithm is $\tilde{O}(\epsilon^{-2})$ when the score estimator $L_2$ error is $\tilde{O}(\epsilon)$, which is controlled by concentration inequalities.

For time $t$, since we have $D_{\text{KL}}(\tilde{p}_{T-t}\|p_{T-t}) \leq \epsilon$ and $p_{T-t}$ is sub-Gaussian, the density $\tilde{p}_{T-t}$ is also sub-Gaussian with variance $\sigma_t'^2$.

We have

$$\mathbb{P}(|x - \mathbb{E}x| > \sqrt{2}\sigma_t'\ln(3/\delta)) \leq \frac{\delta}{3}$$

For Gaussian mixture model, $\exp(-f_*(x_0))$ is 2-Lipschitz and function $f_*$ is 1-Lipschitz smooth, $-\frac{x-e^{-(T-t)}x_0}{\left(1-e^{-2(T-t)}\right)}\cdot\exp(-f_*(x_0))$ is $G_{x,t}$-Lipschitz.

For time $t$, the variance is $\sigma_t^2 = \frac{1-e^{-2(T-t)}}{e^{-2(T-t)}}$.

Assume that the expectation and the estimator of $-\frac{x-e^{-(T-t)}x_0}{\left(1-e^{-2(T-t)}\right)}\cdot\exp(-f_*(x_0))$ is $\mu_X$ and $X$. The expectation and the estimator of $\exp(-f_*(x_0))$ is $\mu_Y$ and $Y$

$$\mathbb{P}(|X - \mu_X| > \epsilon) \leq \exp\left(-\frac{n\epsilon^2}{2\sigma_t^2 G_{x,t}^2}\right)$$

$$\mathbb{P}(|Y - \mu_Y| > \epsilon) \leq \exp\left(-\frac{n\epsilon^2}{8\sigma_t^2}\right)$$

We choose $x_+ = \mathbb{E}x + \sqrt{2}\sigma_t'\ln(3/\delta)$, $G = G_{x_+,t}$.

If $n = \max(G^2, 4)\sigma_t^2\epsilon^{-2}\ln(3\delta^{-1})$, with probability $1-\delta$, the error of the estimator is at most $\epsilon$.

Moreover, if we choose the inner loop iteration to estimate the posterior. We can start from some $T > 0$, and estimate $p_T$ initially.

Specifically, for the Gaussian mixture, we can get a tighter log-Sobolev bound. For time $t$, we have $p_t \propto e^{-\frac{1}{2}\|x\|^2} + e^{-\frac{1}{2}\|x-e^{-t}y\|^2}$, which indicate the log-Sobolev constant, $C_{\text{LSI},y,t}^{-1} \leq 1 + \frac{1}{4}(e^{\|e^{-t}y\|^2} + 1)$. Considering the smoothness, we have

$$\left|\frac{\mathrm{d}^2}{\mathrm{d}x^2}\log p(x)\right| = \left|-1 + \left(-\frac{e^{x^2}}{(e^{x^2/2} + e^{1/2(x-y)^2})^2} + \frac{e^{x^2/2}}{(e^{x^2/2} + e^{1/2(x-y)^2})}\right)y^2\right| \leq 1.$$

When we choose $T = -\frac{1}{2}\log\frac{2L}{2L+1} = \frac{1}{2}\log\frac{3}{2}$, The estimation of $p_T$ needs $O(e^{\frac{2}{3}y^2})$ iterations, which improves the original $O(e^{y^2})$. $\qquad\square$

**Lemma 5.** *Suppose the negative log density of target distribution $f_* = -\ln p_*$ satisfies*

$$f_*(x) = \left(\|x\|^2 + 1\right)^a$$

*where $a \in [0, 0.5]$.*

*Proof.* Consider the Hessian of $f_*$, we have

$$\nabla^2 f_*(\boldsymbol{x}) = 2a \cdot (x^2 + 1)^{a-2} \cdot \left((2a-1)x^2 + 1\right).$$

If we require $|x| \geq r^{-1/(2-2a)}$, it has

$$|x| \geq r^{-1/(2-2a)} \quad \Rightarrow |x|^{2-2a} \geq r^{-1} \quad \Leftrightarrow \quad r \geq \frac{|x|^2}{|x|^{4-2a}}$$

$$\Rightarrow \quad \frac{r}{a} \geq \frac{(1-2a)|x|^2 - 1}{(|x|^2 + 1)^{2-a}} \Rightarrow \quad 2a \cdot (x^2 + 1)^{a-2} \cdot \left((2a-1)x^2 + 1\right) + 2r = \nabla^2(f_*(\boldsymbol{x}) + r|x|^2) \geq 0.$$

It means if we choose $C_R = 1$ and $n = 1/(2 - 2a)$ $\qquad\square$

### B.4 More discussion about previous works

Recent studies have underscored the potential of diffusion models, exploring their integration across various domains, including approximate Bayesian computation methods. One line of research (Vargas et al., 2023a; Berner et al., 2022; Zhang et al., 2023; Vargas et al., 2023c;b) involves applying reverse diffusion in the VI framework to create posterior samplers by neural networks. These studies have examined the conditional expected form of the score function, similar to Lemma 1. Such score-based VI algorithms have shown to offer improvements over traditional VI. However, upon adopting a neural network estimator, VI-based algorithms are subject to an inherent unknown error.

Other research (Tzen & Raginsky, 2019; Chen et al., 2022a) has also delved into the characteristics of parameterized diffusion-like processes under assumed error conditions. Yet, the comparative advantages of diffusion models against MCMC methods and the computational complexity involved in learning the score function are not well-investigated. This gap hinders a straightforward comparison with Langevin-based dynamics.

Another related work is the Schrödinger-Föllmer Sampler (SFS) (Huang et al., 2021), which also tend to estimate the drift term with non-parametric estimators. The main difference of the proposed algorithm is that SFS leverages Schrödinger-Föllmer process. In this process, the target distribution $p_*$ is transformed into a Dirac delta distribution. This transformation often results in the gradient $\nabla \log p_t$ becoming problematic when $p_t$ closely resembles a delta distribution, posing challenges for maintaining the Lipschitz continuity of the drift term. Huang et al. (2021); Tzen & Raginsky (2019) note that the assumption holds when both $p_* \exp(\|x\|^2/2)$ and its gradient are Lipschitz continuous, and the former is bounded below by a positive value. However, this condition may not be met when the variance of $p_*$ exceeds 1, limiting its general applicability in the Schrödinger-Föllmer process. The comparison of SFS with traditional MCMC methods under general conditions remains an open question. However, given that the $p_\infty$ of the OU process represents a smooth distribution – a standard Gaussian, the requirement for the Lipschitz continuity of $\nabla p_t$ is much weaker, as diffusion analysis suggested (Chen et al., 2022a; 2023). Additionally, Lee et al. (2021a) indicated that $L$-smoothness in log-concave $p_*$ implies the smoothness in $p_t$. Moreover, the SFS algorithm considers the importance sampling estimator and the error analysis is mainly based on the Gaussian mixture model. As we mentioned in our Section 3.3, importance sampling estimator would suffer from curse of dimensionality in real practice. Our ULA-based analysis can be adapted to more general distributions for both ill-behaved LSI and non-LSI distributions. In a word, our proposed RDMC provide the complexity for general distributions and SFS is more task specific. It remains open to further investigate the complexity for SFS-like algorithm, which can be interesting future work. Moreover, it is also possible to adapt the ULA estimator idea to SFS.

Our approach, stands as an asymptotically exact sampling algorithm, akin to traditional MCMC methods. It allows us to determine an overall convergence rate that diminishes with increasing

computational complexity, enabling a direct comparison of complexity with MCMC approaches. The main technique of our algorithm is to analyze the complexity of the score estimation with non-parametric algorithm and we found the merits of the proposed one compared with MCMC. Our theory can also support the diffusion-based VI against Langevin-based ones.

## C  MAIN PROOFS

### C.1  PROOF OF LEMMA 1

*Proof.* When the OU process, i.e., Eq. 1, is selected as our forward path, the transition kernel of $(\mathbf{x}_t)_{t\geq 0}$ has a closed form, i.e.,

$$p(\boldsymbol{x}, t|\boldsymbol{x}_0, 0) = \left(2\pi\left(1 - e^{-2t}\right)\right)^{-d/2} \cdot \exp\left[\frac{-\left\|\boldsymbol{x} - e^{-t}\boldsymbol{x}_0\right\|^2}{2\left(1 - e^{-2t}\right)}\right], \quad \forall\, 0 \leq t \leq T.$$

In this condition, we have

$$p_{T-t}(\boldsymbol{x}) = \int_{\mathbb{R}^d} p_0(\boldsymbol{x}_0) \cdot p_{T-t|0}(\boldsymbol{x}|\boldsymbol{x}_0)\mathrm{d}\boldsymbol{x}_0$$

$$= \int_{\mathbb{R}^d} p_0(\boldsymbol{x}_0) \cdot \left(2\pi\left(1 - e^{-2(T-t)}\right)\right)^{-d/2} \cdot \exp\left[\frac{-\left\|\boldsymbol{x} - e^{-(T-t)}\boldsymbol{x}_0\right\|^2}{2\left(1 - e^{-2(T-t)}\right)}\right]\mathrm{d}\boldsymbol{x}_0$$

Plugging this formulation into the following equation

$$\nabla_{\boldsymbol{x}} \ln p_{T-t}(\boldsymbol{x}) = \frac{\nabla p_{T-t}(\boldsymbol{x})}{p_{T-t}(\boldsymbol{x})},$$

we have

$$\nabla_{\boldsymbol{x}} \ln p_{T-t}(\boldsymbol{x}) = \frac{\nabla \int_{\mathbb{R}^d} p_0(\boldsymbol{x}_0) \cdot \left(2\pi\left(1 - e^{-2(T-t)}\right)\right)^{-d/2} \cdot \exp\left[\frac{-\left\|\boldsymbol{x} - e^{-(T-t)}\boldsymbol{x}_0\right\|^2}{2\left(1 - e^{-2(T-t)}\right)}\right]\mathrm{d}\boldsymbol{x}_0}{\int_{\mathbb{R}^d} p_0(\boldsymbol{x}_0) \cdot \left(2\pi\left(1 - e^{-2(T-t)}\right)\right)^{-d/2} \cdot \exp\left[\frac{-\left\|\boldsymbol{x} - e^{-(T-t)}\boldsymbol{x}_0\right\|^2}{2\left(1 - e^{-2(T-t)}\right)}\right]\mathrm{d}\boldsymbol{x}_0}$$

$$= \frac{\int_{\mathbb{R}^d} p_0(\boldsymbol{x}_0) \cdot \exp\left(\frac{-\left\|\boldsymbol{x} - e^{T-t}\boldsymbol{x}_0\right\|^2}{2\left(1 - e^{-2(T-t)}\right)}\right) \cdot \left(-\frac{\boldsymbol{x} - e^{-(T-t)}\boldsymbol{x}_0}{\left(1 - e^{-2(T-t)}\right)}\right)\mathrm{d}\boldsymbol{x}_0}{\int_{\mathbb{R}^d} p_0(\boldsymbol{x}_0) \cdot \exp\left(\frac{-\left\|\boldsymbol{x} - e^{-(T-t)}\boldsymbol{x}_0\right\|^2}{2\left(1 - e^{-2(T-t)}\right)}\right)\mathrm{d}\boldsymbol{x}_0} \tag{13}$$

$$= \mathbb{E}_{\mathbf{x}_0 \sim q_{T-t}(\cdot|\boldsymbol{x})}\left[-\frac{\boldsymbol{x} - e^{-(T-t)}\mathbf{x}_0}{\left(1 - e^{-2(T-t)}\right)}\right]$$

where the density function $q_{T-t}(\cdot|\boldsymbol{x})$ is defined as

$$q_{T-t}(\boldsymbol{x}_0|\boldsymbol{x}) = \frac{p_0(\boldsymbol{x}_0) \cdot \exp\left(\frac{-\left\|\boldsymbol{x} - e^{T-t}\boldsymbol{x}_0\right\|^2}{2\left(1 - e^{-2(T-t)}\right)}\right)}{\int_{\mathbb{R}^d} p_0(\boldsymbol{x}_0) \cdot \exp\left(\frac{-\left\|\boldsymbol{x} - e^{T-t}\boldsymbol{x}_0\right\|^2}{2\left(1 - e^{-2(T-t)}\right)}\right)\mathrm{d}\boldsymbol{x}_0} \propto \exp\left(-f_*(\boldsymbol{x}_0) - \frac{\left\|\boldsymbol{x} - e^{-(T-t)}\boldsymbol{x}_0\right\|^2}{2\left(1 - e^{-2(T-t)}\right)}\right).$$

Hence, the proof is completed. $\qquad\qquad\square$

### C.2  PROOF OF LEMMA 2 AND 3

**Lemma 6.** *(Proposition 1 in Ma et al. (2019)) For $p_* \propto e^{-U}$, where $U$ is $m$-strongly convex outside of a region of radius $R$ and $L$-Lipschitz smooth, the log-Sobolev constant of $p_*$*

$$\rho_U \geq \frac{m}{2}e^{-16LR^2}.$$

*Proof.* By Lemma 6, for any $t = T - k\eta$, we have the LSI constant of $q_{T-k\eta}$ satisfies

$$C_{\mathrm{LSI},k} \geq \frac{e^{-2(T-k\eta)}}{6(1 - e^{-2(T-k\eta)})}\exp\left(-16 \cdot 3L \cdot R^2\left(\frac{e^{-2(T-k\eta)}}{6(1 - e^{-2(T-k\eta)})}\right)\right).$$

When $\frac{2L}{1+2L} \leq e^{-2(T-k\eta)} \leq 1$. We have

$$-L + \frac{e^{-2(T-k\eta)}}{2\left(1 - e^{-2(T-k\eta)}\right)} \geq 0,$$

which implies

$$\Sigma_{k,\max}\boldsymbol{I} := \frac{3e^{-2(T-k\eta)}}{2\left(1-e^{-2(T-k\eta)}\right)} \cdot \boldsymbol{I} \succeq \nabla^2 g_{T-k\eta}(\boldsymbol{x}) \succeq \frac{e^{-2(T-k\eta)}}{2\left(1-e^{-2(T-k\eta)}\right)} \cdot \boldsymbol{I} := \Sigma_{k,\min}\boldsymbol{I}.$$

(14)

Due to the fact that $g_{T-k\eta}$ is $\Sigma_{k,\min}$-strongly convex, $\Sigma_{k,\max}$-smooth and Lemma 20, we have $C_{\mathrm{LSI},k} \geq \Sigma_{k,\min}$.

□

### C.3 PROOF OF MAIN THEOREM

*Proof.* We have

$$
\begin{aligned}
D_{\mathrm{TV}}\left(\tilde{p}_T, p_*\right) \leq & D_{\mathrm{TV}}\left(\tilde{P}_T, \hat{P}_T\right) \leq D_{\mathrm{TV}}\left(\tilde{P}_T, \tilde{P}_T^{p_T}\right) + D_{\mathrm{TV}}\left(\tilde{P}_T^{p_T}, \hat{P}_T\right) \\
\leq & D_{\mathrm{TV}}\left(\tilde{p}_0, p_T\right) + D_{\mathrm{TV}}\left(\tilde{P}_T^{p_T}, \hat{P}_T\right) \leq \underbrace{\sqrt{\frac{1}{2}D_{\mathrm{KL}}\left(\tilde{p}_0\|p_T\right)}}_{\text{Term 1}} + \underbrace{\sqrt{\frac{1}{2}D_{\mathrm{KL}}\left(\hat{P}_T\|\tilde{P}_T^{p_T}\right)}}_{\text{Term 2}},
\end{aligned}
$$

(15)

where the first and the third inequalities follow from data-processing inequality, the second inequality follows from the triangle inequality, and the last inequality follows from Pinsker's inequality.

For Term 2,

$$
\begin{aligned}
D_{\mathrm{KL}}\left(\hat{P}_T\|\tilde{P}_T^{p_T}\right) = \mathbb{E}_{\hat{P}_T}\left[\ln\frac{\mathrm{d}\hat{P}_T}{\mathrm{d}\tilde{P}_T^{p_T}}\right] = & \frac{1}{4}\sum_{k=0}^{N-1}\mathbb{E}_{\hat{P}_T}\left[\int_{k\eta}^{(k+1)\eta}\|\mathbf{v}_k(\mathbf{x}_{k\eta}) - 2\nabla\ln p_{T-t}(\mathbf{x}_t)\|^2\,\mathrm{d}t\right] \\
= & \sum_{k=0}^{N-1}\int_{k\eta}^{(k+1)\eta}\mathbb{E}_{\hat{P}_T}\left[\frac{1}{4}\cdot\|\mathbf{v}_k(\mathbf{x}_{k\eta}) - 2\nabla\ln p_{T-t}(\mathbf{x}_t)\|^2\right]\,\mathrm{d}t,
\end{aligned}
$$

By Lemma 7,

$$\frac{1}{4}\cdot\mathbb{E}_{\hat{P}_T}\left[\|\mathbf{v}_k(\mathbf{x}_{k\eta}) - 2\nabla\ln p_{T-t}(\mathbf{x}_t)\|^2\right] \leq 4\epsilon^2 + \frac{1}{2}\cdot\underbrace{\mathbb{E}_{\hat{P}_T}\left[\|\mathbf{v}_k(\mathbf{x}_{k\eta}) - 2\nabla\ln p_{T-k\eta}(\mathbf{x}_{k\eta})\|^2\right]}_{\epsilon_{\text{score}}}$$

(16)

According to Eq. 4, for each term of the summation, we have

$$
\begin{aligned}
& \mathbb{E}_{\hat{P}_T}\left[\|\mathbf{v}_k(\mathbf{x}_{k\eta}) - 2\nabla\ln p_{T-k\eta}(\mathbf{x}_{k\eta})\|^2\right] \\
= & \mathbb{E}_{\hat{P}_T}\left[\left\|\mathbf{v}_k(\mathbf{x}_{k\eta}) - 2\mathbb{E}_{\mathbf{x}_0\sim q_{T-k\eta}(\cdot|\mathbf{x}_{k\eta})}\left[-\frac{\mathbf{x}_{k\eta} - e^{-(T-k\eta)}\mathbf{x}_0}{\left(1-e^{-2(T-k\eta)}\right)}\right]\right\|^2\right].
\end{aligned}
$$

For each $\mathbf{x}_{k\eta} = \boldsymbol{x}$, we have

$$
\begin{aligned}
&\left\| \mathbf{v}_k(\boldsymbol{x}) - 2\mathbb{E}_{\mathbf{x}_0 \sim q_{T-k\eta}(\cdot|\boldsymbol{x})} \left[ -\frac{\boldsymbol{x} - e^{-(T-k\eta)}\mathbf{x}_0}{\left(1 - e^{-2(T-k\eta)}\right)} \right] \right\|^2 \\
=&\left\| \mathbf{v}_k(\boldsymbol{x}) - 2\mathbb{E}_{\mathbf{x}'_0 \sim q'_{T-k\eta}(\cdot|\boldsymbol{x})} \left[ -\frac{\boldsymbol{x} - e^{-(T-k\eta)}\mathbf{x}'_0}{\left(1 - e^{-2(T-k\eta)}\right)} \right] \right. \\
&\left. + \frac{2e^{-(T-k\eta)}}{1 - e^{-2(T-k\eta)}} \left[ \mathbb{E}_{\mathbf{x}'_0 \sim q'_{T-k\eta}(\cdot|\boldsymbol{x})}[\mathbf{x}'_0] - \mathbb{E}_{\mathbf{x}_0 \sim q_{T-k\eta}(\cdot|\boldsymbol{x})}[\mathbf{x}_0] \right] \right\|^2 \\
\leq& 2\underbrace{\left\| \mathbf{v}_k(\boldsymbol{x}) - 2\mathbb{E}_{\mathbf{x}'_0 \sim q'_{T-k\eta}(\cdot|\boldsymbol{x})} \left[ -\frac{\boldsymbol{x} - e^{-(T-k\eta)}\mathbf{x}'_0}{\left(1 - e^{-2(T-k\eta)}\right)} \right] \right\|^2}_{\text{Term 2.1}} \\
&+ 2\underbrace{\left\| \frac{2e^{-(T-k\eta)}}{1 - e^{-2(T-k\eta)}} \left[ \mathbb{E}_{\mathbf{x}'_0 \sim q'_{T-k\eta}(\cdot|\boldsymbol{x})}[\mathbf{x}'_0] - \mathbb{E}_{\mathbf{x}_0 \sim q_{T-k\eta}(\cdot|\boldsymbol{x})}[\mathbf{x}_0] \right] \right\|^2}_{\text{Term 2.2}},
\end{aligned}
\tag{17}
$$

where we denote $q'_{T-k\eta}(\cdot|\boldsymbol{x})$ denote the underlying distribution of output particles of the auxiliary sampling task.

For Term 2.2, we denote an optimal coupling between $q_{T-k\eta}(\cdot|\boldsymbol{x})$ and $q'_{T-k\eta}(\cdot|\boldsymbol{x})$ to be

$$
\gamma \in \Gamma_{\text{opt}}(q_{T-k\eta}(\cdot|\boldsymbol{x}), q'_{T-k\eta}(\cdot|\boldsymbol{x})).
$$

Hence, we have

$$
\begin{aligned}
&\left\| \frac{2e^{-(T-k\eta)}}{1 - e^{-2(T-k\eta)}} \left[ \mathbb{E}_{\mathbf{x}'_0 \sim q'_{T-k\eta}(\cdot|\boldsymbol{x})}[\mathbf{x}'_0] - \mathbb{E}_{\mathbf{x}_0 \sim q_{T-k\eta}(\cdot|\boldsymbol{x})}[\mathbf{x}_0] \right] \right\|^2 \\
=&\left\| \mathbb{E}_{(\mathbf{x}_0, \mathbf{x}'_0) \sim \gamma} \left[ \frac{2e^{-(T-k\eta)}}{1 - e^{-2(T-k\eta)}} \cdot (\mathbf{x}_0 - \mathbf{x}'_0) \right] \right\|^2 \leq \frac{4e^{-2(T-k\eta)}}{(1 - e^{-2(T-k\eta)})^2} \cdot \mathbb{E}_{(\mathbf{x}_0, \mathbf{x}'_0) \sim \gamma} \left[ \|\mathbf{x}_0 - \mathbf{x}'_0\|^2 \right] \\
=&\frac{4e^{-2(T-k\eta)}}{(1 - e^{-2(T-k\eta)})^2} W_2^2 \left( q_{T-k\eta}(\cdot|\boldsymbol{x}), q'_{T-k\eta}(\cdot|\boldsymbol{x}) \right) \\
\leq&\frac{4e^{-2(T-k\eta)}}{\left(1 - e^{-2(T-k\eta)}\right)^2} \cdot \frac{2}{C_{\text{LSI},k}} D_{\text{KL}} \left( q'_{T-k\eta}(\cdot|\boldsymbol{x}) \| q_{T-k\eta}(\cdot|\boldsymbol{x}) \right) \\
\leq& 8\eta^{-2} C_{\text{LSI},k}^{-1} D_{\text{KL}} \left( q'_{T-k\eta}(\cdot|\boldsymbol{x}) \| q_{T-k\eta}(\cdot|\boldsymbol{x}) \right),
\end{aligned}
\tag{18}
$$

where the first inequality follows from Jensen's inequality, the second inequality follows from the Talagrand inequality and the last inequality follows from

$$
e^{-2(T-k\eta)} \leq e^{-2\eta} \leq 1 - \eta \;\Rightarrow\; \frac{e^{-2(T-k\eta)}}{(1 - e^{-2(T-k\eta)})^2} \leq \eta^{-2},
$$

when $\eta \leq 1/2$.

By Lemma 8 and 9, the desired convergence can be obtained.

$\square$

## C.4 PROOF OF THE MAIN PROPOSITIONS

*Proof.* By Eq. (19), we have the upper bound for

$$\underbrace{\sqrt{\frac{1}{2}D_{\mathrm{KL}}\left(\tilde{p}_0\|p_T\right)}}_{\text{Term 1}} + \underbrace{\sqrt{\frac{1}{2}D_{\mathrm{KL}}\left(\hat{P}_T\|\tilde{P}_T^{p_T}\right)}}_{\text{Term 2}}. \tag{19}$$

We aim to upper bound these two terms in our analysis.

**Errors from the forward process**  For Term 1 of Eq. 19, we can either choose $D_{\mathrm{KL}}\left(\hat{p}\|p_T\right)$ or choose large $T$ with $\hat{p} = p_\infty$.

If we choose $\hat{p} = p_\infty$, we have

$$D_{\mathrm{KL}}\left(\tilde{p}_0\|p_\infty\right) = D_{\mathrm{KL}}\left(p_T\|p_\infty\right) \leq C_0 \exp\left(-\frac{T}{2}\right),$$

where the inequality follows from Lemma 21. By requiring

$$C_0 \cdot \exp\left(-\frac{T}{2}\right) \leq 2\epsilon^2 \iff T \geq 2\ln\frac{C_0}{2\epsilon^2},$$

we have Term $1 \leq \epsilon$. To simplify the proof, we choose $T$ as its lower bound.

If we choose $\hat{p}$, then the iteration complexity depend on the log-Sobolev constant of $p_T$. In (Ma et al., 2019), it is demonstrated that any distribution satisfying Assumptions [$\mathbf{A}_1$] and [$\mathbf{A}_3$] has a log-Sobolev constant of $\frac{m}{2}\exp(-16LR^2)$, which scales exponentially with the radius $R$.

Considering the $p_T$, we have
$$X_T = e^{-T}X_0 + \sqrt{1 - e^{-2T}}\varepsilon.$$

Assume that the density of $e^{-T}X_0$ is $h$,

$$\log h(e^{-T}x) + \log|e^{-T}I| = \log p_0(x)$$

$$\log h(e^{-T}x) = \log p_0(x) + dT.$$

Assume that $y = e^{-T}x$ and Assumption [$\mathbf{A}_3$] holds,

$$-\nabla^2 h(y) = -e^{2T}\nabla^2 p_0(e^T y) \geq e^{2T}mI.$$

We have outside a ball with $e^{-T}R$, the negative log-density is $e^{2T}m$ strongly convex. By Lemma 16, the final log-Sobolev constant is

$$\frac{1}{\frac{2}{m\exp(-16LR^2 e^{-T}+2T)} + \frac{1}{1-e^{-2T}}} = O(m\exp(-16LR^2 e^{-T} + 2T)).$$

**Errors from the backward process**  Without loss of generality, we consider the Assumption [$\mathbf{A}_4$] case, where $t$ has been split to two intervals. The Assumption [$\mathbf{A}_3$] case can be recognized as the first interval of Assumption [$\mathbf{A}_4$]. For Term 2, we first consider the proof when Novikov's condition holds for simplification. A more rigorous analysis without Novikov's condition can be easily extended with the tricks shown in (Chen et al., 2022a). Considering Corollary 2, we have

$$
\begin{aligned}
D_{\mathrm{KL}}\left(\hat{P}_T\|\tilde{P}_T^{p_T}\right) = \mathbb{E}_{\hat{P}_T}\left[\ln\frac{\mathrm{d}\hat{P}_T}{\mathrm{d}\tilde{P}_T^{p_T}}\right] &= \frac{1}{4}\sum_{k=0}^{N-1}\mathbb{E}_{\hat{P}_T}\left[\int_{k\eta}^{(k+1)\eta}\|\mathbf{v}_k(\mathbf{x}_{k\eta}) - 2\nabla\ln p_{T-t}(\mathbf{x}_t)\|^2\,\mathrm{d}t\right] \\
&= \sum_{k=0}^{N-1}\int_{k\eta}^{(k+1)\eta}\mathbb{E}_{\hat{P}_T}\left[\frac{1}{4}\cdot\|\mathbf{v}_k(\mathbf{x}_{k\eta}) - 2\nabla\ln p_{T-t}(\mathbf{x}_t)\|^2\right]\mathrm{d}t,
\end{aligned}
\tag{20}
$$

where $N = \lfloor T/\eta \rfloor$.

We also have

$$\frac{1}{4} \cdot \mathbb{E}_{\hat{P}_T} \left[ \|\mathbf{v}_k(\mathbf{x}_{k\eta}) - 2\nabla \ln p_{T-t}(\mathbf{x}_t)\|^2 \right] \leq 4\epsilon^2 + \frac{1}{2} \cdot \underbrace{\mathbb{E}_{\hat{P}_T} \left[ \|\mathbf{v}_k(\mathbf{x}_{k\eta}) - 2\nabla \ln p_{T-k\eta}(\mathbf{x}_{k\eta})\|^2 \right]}_{\epsilon_{\text{score}}} \tag{21}$$

following Lemma 7 by choosing the step size of the backward path satisfying

$$\eta \leq C_1 \left(d + m_2^2\right)^{-1} \epsilon^2. \tag{22}$$

To simplify the proof, we choose $\eta$ as its upper bound. Plugging Eq. 32 into Eq. 20, we have

$$D_{\text{KL}} \left( \hat{P}_T \| \tilde{P}_T^{p_T} \right) \leq 8\epsilon^2 \ln \frac{C_0}{2\epsilon^2} + \frac{1}{2} \cdot \sum_{k=0}^{N-1} \eta \cdot \mathbb{E}_{\hat{P}_T} \left[ \|\mathbf{v}_k(\mathbf{x}_{k\eta}) - 2\nabla \ln p_{T-k\eta}(\mathbf{x}_{k\eta})\|^2 \right].$$

Besides, due to Lemma 10, we have

$$\sum_{k=0}^{N-1} \eta \cdot \mathbb{E}_{\hat{P}_T} \left[ \|\mathbf{v}_k(\mathbf{x}_{k\eta}) - 2\nabla \ln p_{T-k\eta}(\mathbf{x}_{k\eta})\|^2 \right] \leq 20\epsilon^2 \ln \frac{C_0}{2\epsilon^2}$$

with a probability at least $1 - \epsilon$ by requiring an

$$\mathcal{O} \left( \max \left(C_3 C_5, C_3' C_5'\right) \cdot C_1^{-1} C_0 \cdot (d + m_2^2)^{18} \epsilon^{-16n-88} \exp \left(5C_2 \epsilon^{-16n}\right) \right)$$

gradient complexity. Hence, we have

$$\text{Term 2} \leq \sqrt{\frac{1}{2} \cdot (4\epsilon^2 + 5\epsilon^2) \cdot T} \leq 3\epsilon \sqrt{\ln \left( \frac{C}{2\epsilon^2} \right)},$$

which implies

$$D_{\text{TV}} \left( \tilde{p}_t, p_* \right) \leq \epsilon + 3\epsilon \sqrt{\ln \left( \frac{C}{2\epsilon^2} \right)} = \tilde{O}(\epsilon). \tag{23}$$

Hence, the proof is completed.

$\square$

## D  Important Lemmas

**Lemma 7.** *(Errors from the discretization) With Algorithm 1 and notation list 1, if we choose the step size of outer loop satisfying*

$$\eta \leq C_1 \left(d + m_2^2\right)^{-1} \epsilon^2,$$

*then for $t \in [k\eta, (k+1)\eta]$ we have*

$$\mathbb{E}_{\hat{P}_T} \left[ \left\| \nabla \ln \frac{p_{T-k\eta}(\mathbf{x}_{k\eta})}{p_{T-t}(\mathbf{x}_{k\eta})} \right\|^2 \right] + L^2 \cdot \mathbb{E}_{\hat{P}_T} \left[ \|\mathbf{x}_{k\eta} - \mathbf{x}_t\|^2 \right] \leq \epsilon^2.$$

$$\frac{1}{4} \cdot \mathbb{E}_{\hat{P}_T} \left[ \|\mathbf{v}_k(\mathbf{x}_{k\eta}) - 2\nabla \ln p_{T-t}(\mathbf{x}_t)\|^2 \right] \leq 4\epsilon^2 + \frac{1}{2} \cdot \underbrace{\mathbb{E}_{\hat{P}_T} \left[ \|\mathbf{v}_k(\mathbf{x}_{k\eta}) - 2\nabla \ln p_{T-k\eta}(\mathbf{x}_{k\eta})\|^2 \right]}_{\epsilon_{\text{score}}}.$$

*Proof.* According to the choice of $t$, we have $T - k\eta \geq T - t$. With the transition kernel of the forward process, we have the following connection

$$p_{T-k\eta}(\mathbf{x}) = \int p_{T-t}(\mathbf{y}) \cdot \mathbb{P} \left( \mathbf{x}, T - k\eta | \mathbf{y}, T - t \right) \mathrm{d}\mathbf{y}$$

$$= \int p_{T-t}(\mathbf{y}) \left[ 2\pi \left( 1 - e^{-2(t-k\eta)} \right) \right]^{-\frac{d}{2}} \cdot \exp \left[ \frac{-\left\| \mathbf{x} - e^{-(t-k\eta)} \mathbf{y} \right\|^2}{2 \left( 1 - e^{-2(t-k\eta)} \right)} \right] \mathrm{d}\mathbf{y}$$

$$= \int e^{(t-k\eta)d} p_{T-t} \left( e^{(t-k\eta)} \mathbf{z} \right) \left[ 2\pi \left( 1 - e^{-2(t-k\eta)} \right) \right]^{-\frac{d}{2}} \cdot \exp \left[ -\frac{\|\mathbf{x} - \mathbf{z}\|^2}{2 \left( 1 - e^{-2(t-k\eta)} \right)} \right] \mathrm{d}\mathbf{z}, \tag{24}$$

where the last equation follows from setting $\boldsymbol{z} = e^{-(t-k\eta)}\boldsymbol{y}$. We should note that

$$p'_{T-t}(\boldsymbol{z}) := e^{(t-k\eta)d}p_{T-t}(e^{(t-k\eta)}\boldsymbol{z})$$

is also a density function. For each element $\mathbf{x}_{k\eta} = \boldsymbol{x}$, we have

$$\left\|\nabla\ln\frac{p_{T-t}(\boldsymbol{x})}{p_{T-k\eta}(\boldsymbol{x})}\right\|^2 = \left\|\nabla\ln\frac{p_{T-t}(\boldsymbol{x})}{e^{(t-k\eta)d}p_{T-t}\left(e^{(t-k\eta)}\boldsymbol{x}\right)} + \nabla\ln\frac{e^{(t-k\eta)d}p_{T-t}\left(e^{(t-k\eta)}\boldsymbol{x}\right)}{p_{T-k\eta}(\boldsymbol{x})}\right\|^2$$

$$\leq 2\left\|\nabla\ln\frac{p_{T-t}(\boldsymbol{x})}{e^{(t-k\eta)d}p_{T-t}\left(e^{(t-k\eta)}\boldsymbol{x}\right)}\right\|^2 + 2\left\|\nabla\ln\frac{e^{(t-k\eta)d}p_{T-t}\left(e^{(t-k\eta)}\boldsymbol{x}\right)}{\left(p'_{T-t}*\varphi_{\left(1-e^{-2(t-k\eta)}\right)}\right)(\boldsymbol{x})}\right\|^2,$$

where the inequality follows from the triangle inequality and Eq. 24. For the first term, we have

$$\left\|\nabla\ln\frac{p_{T-t}(\boldsymbol{x})}{e^{(t-k\eta)d}p_{T-t}\left(e^{(t-k\eta)}\boldsymbol{x}\right)}\right\| = \left\|\nabla\ln p_{T-t}(\boldsymbol{x}) - e^{(t-k\eta)}\cdot\nabla\ln p_{T-t}\left(e^{(t-k\eta)}\boldsymbol{x}\right)\right\|$$

$$\leq \left\|\nabla\ln p_{T-t}(\boldsymbol{x}) - e^{(t-k\eta)}\cdot\nabla\ln p_{T-t}(\boldsymbol{x})\right\| + e^{(t-k\eta)}\cdot\left\|\nabla\ln p_{T-t}(\boldsymbol{x}) - \nabla\ln p_{T-t}\left(e^{(t-k\eta)}\boldsymbol{x}\right)\right\|$$

$$\leq \left(e^{(t-k\eta)} - 1\right)\|\nabla\ln p_{T-t}(\boldsymbol{x})\| + e^{(t-k\eta)}\cdot\left(e^{(t-k\eta)} - 1\right)L\|\boldsymbol{x}\|.$$

$$(25)$$

For the second term, the score $-\nabla\ln p'_{T-t}$ is $\left(e^{2(t-k\eta)}L\right)$-smooth. Therefore, with Lemma 13 and the requirement

$$2\cdot e^{2(t-k\eta)}\cdot\left(1 - e^{-2(t-k\eta)}\right) \leq \frac{1}{L} \Leftarrow \begin{cases} 4(t-k\eta) \leq \dfrac{1}{2L} \\ t-k\eta \leq \dfrac{1}{2} \end{cases} \Leftarrow \eta \leq \min\left\{\frac{1}{8L}, \frac{1}{2}\right\},$$

we have

$$\left\|\nabla\ln p'_{T-t}(\boldsymbol{x}) - \nabla\ln\left(p'_{T-t}*\varphi_{\left(1-e^{-2(t-k\eta)}\right)}\right)(\boldsymbol{x})\right\|$$

$$\leq 6e^{2(t-k\eta)}L\sqrt{\left(1 - e^{-2(t-k\eta)}\right)}d^{1/2} + 2e^{3(t-k\eta)}L\left(1 - e^{-2(t-k\eta)}\right)\left\|\nabla\ln p_{T-t}\left(e^{(t-k\eta)}\boldsymbol{x}\right)\right\|$$

$$\leq 6e^{2(t-k\eta)}L\sqrt{\left(1 - e^{-2(t-k\eta)}\right)}d^{1/2}$$

$$+ 2Le^{(t-k\eta)}\cdot\left(e^{2(t-k\eta)} - 1\right)\left\|\nabla\ln p_{T-t}\left(e^{(t-k\eta)}\boldsymbol{x}\right) - \nabla\ln p_{T-t}(\boldsymbol{x}) + \nabla\ln p_{T-t}(\boldsymbol{x})\right\|$$

$$\leq 6e^{2(t-k\eta)}L\sqrt{\left(1 - e^{-2(t-k\eta)}\right)}d^{1/2} + 2L^2e^{(t-k\eta)}\cdot\left(e^{2(t-k\eta)} - 1\right)\left(e^{(t-k\eta)} - 1\right)\|\boldsymbol{x}\|$$

$$+ 2Le^{(t-k\eta)}\cdot\left(e^{2(t-k\eta)} - 1\right)\|\nabla\ln p_{T-t}(\boldsymbol{x})\|.$$

$$(26)$$

Due to the range $\eta \leq 1/2$, we have the following inequalities

$$e^{2(t-k\eta)} \leq e^{2\eta} \leq 1 + 4\eta, \quad 1 - e^{-2(t-k\eta)} \leq 2(t-k\eta) \leq 2\eta \quad \text{and} \quad e^{(t-k\eta)} \leq e^{\eta} \leq 1 + 2\eta.$$

Thus, Eq. 25 and Eq. 26 can be reformulated as

$$\left\|\nabla\ln\frac{p_{T-t}(\boldsymbol{x})}{e^{(t-k\eta)d}p_{T-t}\left(e^{(t-k\eta)}\boldsymbol{x}\right)}\right\| \leq 2\eta\|\nabla\ln p_{T-t}(\boldsymbol{x})\| + 4\eta L\|\boldsymbol{x}\|$$

$$\Rightarrow \left\|\nabla\ln\frac{p_{T-t}(\boldsymbol{x})}{e^{(t-k\eta)d}p_{T-t}\left(e^{(t-k\eta)}\boldsymbol{x}\right)}\right\|^2 \leq 8\eta^2\|\nabla\ln p_{T-t}(\boldsymbol{x})\|^2 + 32\eta^2L^2\|\boldsymbol{x}\|^2$$

$$(27)$$

and

$$\left\|\nabla\ln p'_{T-t}(\boldsymbol{x}) - \nabla\ln\left(p'_{T-t}*\varphi_{\left(1-e^{-2(t-k\eta)}\right)}\right)(\boldsymbol{x})\right\|$$

$$\leq 6\left(4\eta + 1\right)L\sqrt{2\eta d} + 2L^2\cdot(2\eta + 1)\cdot 4\eta\cdot 2\eta\cdot\|\boldsymbol{x}\| + 2L\cdot(2\eta + 1)\cdot 4\eta\cdot\|\nabla\ln p_{T-t}(\boldsymbol{x})\|$$

$$\leq 18L\sqrt{2\eta d} + 32L^2\eta^2\cdot\|\boldsymbol{x}\| + 16L\eta\cdot\|\nabla\ln p_{T-t}(\boldsymbol{x})\|,$$

which is equivalent to

$$
\left\| \nabla \ln p'_{T-t}(\boldsymbol{x}) - \nabla \ln \left( p'_{T-t} * \varphi_{\left(1-e^{-2(t-k\eta)}\right)} \right)(\boldsymbol{x}) \right\|^2
$$
$$
\leq 3 \cdot \left( 2^{11} \cdot L^2 \eta d + 2^{10} \cdot L^4 \eta^4 \|\boldsymbol{x}\|^2 + 2^8 \cdot L^2 \eta^2 \|\nabla \ln p_{T-t}(\boldsymbol{x})\|^2 \right) \tag{28}
$$
$$
\leq 2^{13} \cdot L^2 \eta d + 2^{12} \cdot L^4 \eta^4 \|\boldsymbol{x}\|^2 + 2^{10} \cdot L^2 \eta^2 \|\nabla \ln p_{T-t}(\boldsymbol{x})\|^2 .
$$

Without loss of generality, we suppose $L \geq 1$, combining Eq. 27 and Eq. 28, we have the following bound

$$
\mathbb{E}_{\hat{P}_T} \left[ \left\| \nabla \ln \frac{p_{T-t}(\mathbf{x}_{k\eta})}{p_{T-k\eta}(\mathbf{x}_{k\eta})} \right\|^2 \right] \leq 2^{14} \cdot L\eta d + 2^8 \cdot L^2 \eta^2 \mathbb{E}_{\hat{P}} \left[ \|\mathbf{x}_{k\eta}\|^2 \right] + 2^{12} \cdot L^2 \eta^2 \mathbb{E}_{\hat{P}} \left[ \|\nabla \ln p_{T-t}(\mathbf{x}_{k\eta})\|^2 \right]
$$
$$
\leq 2^{14} \cdot L\eta d + 2^8 \cdot L^2 \eta^2 \mathbb{E}_{\hat{P}} \left[ \|\mathbf{x}_{k\eta}\|^2 \right] + 2^{13} \cdot L^2 \eta^2 \mathbb{E}_{\hat{P}} \left[ \|\nabla \ln p_{T-t}(\mathbf{x}_t)\|^2 \right]
$$
$$
+ 2^{13} \cdot L^4 \eta^2 \mathbb{E}_{\hat{P}} \left[ \|\mathbf{x}_{k\eta} - \mathbf{x}_t\|^2 \right] . \tag{29}
$$

Besides, we have

$$
4 \left[ \mathbb{E}_{\hat{P}_T} \left[ \left\| \nabla \ln \frac{p_{T-k\eta}(\mathbf{x}_{k\eta})}{p_{T-t}(\mathbf{x}_{k\eta})} \right\|^2 \right] + L^2 \mathbb{E}_{\hat{P}_T} \left[ \|\mathbf{x}_{k\eta} - \mathbf{x}_t\|^2 \right] \right]
$$
$$
\leq 4 \left[ 2^{14} \cdot L\eta d + 2^8 \cdot L^2 \eta^2 \mathbb{E}_{\hat{P}_T} \left[ \|\mathbf{x}_{k\eta}\|^2 \right] + 2^{13} \cdot L^2 \eta^2 \mathbb{E}_{\hat{P}_T} \left[ \|\nabla \ln p_{T-t}(\mathbf{x}_t)\|^2 \right] \right.
$$
$$
\left. + \left( 2^{13} \cdot L^2 \eta^2 + 1 \right) L^2 \mathbb{E}_{\hat{P}_T} \left[ \|\mathbf{x}_{k\eta} - \mathbf{x}_t\|^2 \right] \right] \tag{30}
$$
$$
\leq 2^{16} \cdot L\eta d + 2^{10} \cdot L^2 \eta^2 (d + m_2^2) + 2^{15} \cdot L^3 \eta^2 d + 2^8 \cdot L^2 \left( 2(m_2^2 + d)\eta^2 + 4d\eta \right),
$$

where the last inequality with Lemma 14 and Lemma 15. To diminish the discretization error, we require the step size of backward sampling, i.e., $\eta$ satisfies

$$
\begin{cases} 2^{16} \cdot L\eta d \leq \epsilon^2 \\ 2^{10} \cdot L^2 \eta^2 (d + m_2^2) \leq \epsilon^2 \\ 2^{15} \cdot L^3 \eta^2 d \leq \epsilon^2 \\ 2^8 \cdot L^2 \left( 2(m_2^2 + d)\eta^2 + 4d\eta \right) \leq \epsilon^2 \end{cases} \Longleftarrow \begin{cases} \eta \leq 2^{-16} \cdot L^{-1} d^{-1} \epsilon^2 \\ \eta \leq 2^{-5} \cdot L^{-1} \left( d + m_2^2 \right)^{-0.5} \epsilon \\ \eta \leq 2^{-7.5} \cdot L^{-1.5} d^{-0.5} \epsilon \\ \eta \leq 2^{-5} L^{-0.5} \left( d + m_2^2 \right)^{-0.5} \epsilon \\ \eta \leq 2^{-10} L^{-2} d^{-1} \epsilon^2 \end{cases}
$$

Specifically, if we choose

$$
\eta \leq 2^{-16} \cdot L^{-2} \left( d + m_2^2 \right)^{-1} \epsilon^2 = C_1 (d + m_2^2)^{-1} \epsilon^2,
$$

we have

$$
\mathbb{E}_{\hat{P}_T} \left[ \left\| \nabla \ln \frac{p_{T-k\eta}(\mathbf{x}_{k\eta})}{p_{T-t}(\mathbf{x}_{k\eta})} \right\|^2 \right] + L^2 \mathbb{E}_{\hat{P}_T} \left[ \|\mathbf{x}_{k\eta} - \mathbf{x}_t\|^2 \right] \leq \epsilon^2. \tag{31}
$$

Hence, the proof is completed.

Thus, for $t \in [k\eta, (k+1)\eta]$, it has

$$
\frac{1}{4} \cdot \mathbb{E}_{\hat{P}_T} \left[ \|\mathbf{v}_k(\mathbf{x}_{k\eta}) - 2\nabla \ln p_{T-t}(\mathbf{x}_t)\|^2 \right]
$$
$$
\leq 2 \mathbb{E}_{\hat{P}_T} \left[ \|\nabla \ln p_{T-k\eta}(\mathbf{x}_{k\eta}) - \nabla \ln p_{T-t}(\mathbf{x}_t)\|^2 \right] + \frac{1}{2} \cdot \mathbb{E}_{\hat{P}_T} \left[ \|\mathbf{v}_k(\mathbf{x}_{k\eta}) - 2\nabla \ln p_{T-k\eta}(\mathbf{x}_{k\eta})\|^2 \right]
$$
$$
\leq 4 \mathbb{E}_{\hat{P}_T} \left[ \left\| \nabla \ln \frac{p_{T-k\eta}(\mathbf{x}_{k\eta})}{p_{T-t}(\mathbf{x}_{k\eta})} \right\|^2 \right] + 4L^2 \cdot \mathbb{E}_{\hat{P}_T} \left[ \|\mathbf{x}_{k\eta} - \mathbf{x}_t\|^2 \right]
$$
$$
+ \frac{1}{2} \cdot \mathbb{E}_{\hat{P}_T} \left[ \|\mathbf{v}_k(\mathbf{x}_{k\eta}) - 2\nabla \ln p_{T-k\eta}(\mathbf{x}_{k\eta})\|^2 \right]
$$
$$
\leq 4\epsilon^2 + \frac{1}{2} \cdot \underbrace{\mathbb{E}_{\hat{P}_T} \left[ \|\mathbf{v}_k(\mathbf{x}_{k\eta}) - 2\nabla \ln p_{T-k\eta}(\mathbf{x}_{k\eta})\|^2 \right]}_{\epsilon_{\text{score}}}.
$$
$$
\tag{32}
$$

where the second inequality follows from Assumption [$\mathbf{A}_1$].

$\square$

**Lemma 8.** *For each inner loop, we denote $q_z(\cdot|\boldsymbol{x})$ and $q(\cdot|\boldsymbol{x})$ to be the underlying distribution of output particles and the target distribution, respectively, where $q$ satisfies LSI with constant $\mu$. When we set the step size of outer loops to be $\eta$, by requiring*

$$n \geq 64 T d \mu^{-1} \eta^{-3} \epsilon^{-2} \delta^{-1} \quad \text{and} \quad D_{\mathrm{KL}}\left(q_z \| q\right) \leq 2^{-13} \cdot T^{-4} d^{-2} \mu^2 \eta^8 \epsilon^4 \delta^4,$$

*we have*

$$\mathbb{P}_{\left\{\mathbf{x}_0^{(i)}\right\}_{i=1}^n \sim q_z^{(n)}(\cdot|\boldsymbol{x})}\left[\left\|\frac{1}{n}\sum_{i=1}^n \mathbf{v}_i(\boldsymbol{x}) - \frac{1}{n}\mathbb{E}\left[\sum_{i=1}^n \mathbf{v}_i(\boldsymbol{x})\right]\right\| \geq 2\epsilon\right] \leq \exp\left(-\frac{1}{\delta/(2\lfloor T/\eta\rfloor)}\right) + \frac{\delta}{2\lfloor T/\eta\rfloor},$$

*where*

$$\mathbf{v}_i(\boldsymbol{x}) := -2 \cdot \frac{\boldsymbol{x} - e^{-(T-k\eta)}\mathbf{x}_0^{(i)}}{1 - e^{-2(T-k\eta)}} \quad i \in \{1, \ldots, n\}.$$

*Proof.* For each inner loop, we abbreviate the target distribution as $\tilde{q}$, the initial distribution as $q_0$. Then the iteration of the inner loop is presented as

$$\mathbf{x}_{z+1} = \mathbf{x}_z + \tau \nabla \ln q(\mathbf{x}_z|\boldsymbol{x}) + \sqrt{2\tau}\mathcal{N}(\mathbf{0}, \boldsymbol{I}).$$

We suppose the underlying distribution of the $z$-th iteration to be $q_z(\cdot|\boldsymbol{x})$. Hence, we expect the following inequality

$$\mathbb{P}\left[\left\|\mathbf{v}(\boldsymbol{x}) - \int q_z(\boldsymbol{x}_0)(-2) \cdot \frac{\boldsymbol{x} - e^{T-k\eta}\boldsymbol{x}_0}{1 - e^{-2(T-k\eta)}}\right\|^2 \leq \epsilon^2\right] \geq 1 - \delta$$

is established, where $\mathbf{v}(\boldsymbol{x}) = 1/n \sum_{i=1}^n \mathbf{v}_i(\boldsymbol{x})$. In this condition, we have

$$\mathbb{P}_{\left\{\mathbf{x}_0^{(i)}\right\}_{i=1}^n \sim q_z^{(n)}(\cdot|\boldsymbol{x})}\left[\left\|\frac{1}{n}\sum_{i=1}^n \mathbf{v}_i(\boldsymbol{x}) - \frac{1}{n}\mathbb{E}\left[\sum_{i=1}^n \mathbf{v}_i(\boldsymbol{x})\right]\right\| \geq \mathbb{E}\left\|\frac{1}{n}\sum_{i=1}^n \mathbf{v}_i(\boldsymbol{x}) - \mathbb{E}\mathbf{v}_1(\boldsymbol{x})\right\| + \epsilon\right]$$

$$= \mathbb{P}_{\left\{\mathbf{x}_0^{(i)}\right\}_{i=1}^n \sim q_z^{(n)}(\cdot|\boldsymbol{x})}\left[\left\|\sum_{i=1}^n \mathbf{x}_0^{(i)} - \mathbb{E}\left[\sum_{i=1}^n \mathbf{x}_0^{(i)}\right]\right\| \geq \mathbb{E}\left\|\sum_{i=1}^n \mathbf{x}_i - \mathbb{E}\left[\sum_{i=1}^n \mathbf{x}_0^{(i)}\right]\right\| + \frac{1 - e^{-2(T-k\eta)}}{2e^{-(T-k\eta)}} \cdot n\epsilon\right].$$

$$\tag{33}$$

To simplify the notations, we set

$$\boldsymbol{b}_z := \mathbb{E}_{q_z(\cdot|\boldsymbol{x})}[\mathbf{x}_0], \quad v_z := \mathbb{E}_{\left\{\mathbf{x}_0^{(i)}\right\}_{i=1}^n \sim q_z^n(\cdot|\boldsymbol{x})}\left\|\sum_{i=1}^n \mathbf{x}_i - \mathbb{E}\left[\sum_{i=1}^n \mathbf{x}_0^{(i)}\right]\right\|,$$

$$\boldsymbol{b} := \mathbb{E}_{q(\cdot|\boldsymbol{x})}[\mathbf{x}_0], \quad \text{and} \quad v := \mathbb{E}_{\left\{\mathbf{x}_0^{(i)}\right\}_{i=1}^n \sim q^n(\cdot|\boldsymbol{x})}\left\|\sum_{i=1}^n \mathbf{x}_i - \mathbb{E}\left[\sum_{i=1}^n \mathbf{x}_0^{(i)}\right]\right\|.$$

Then, we have

$$\mathbb{P}_{\left\{\mathbf{x}_0^{(i)}\right\}_{i=1}^n \sim q_z^{(n)}(\cdot|\boldsymbol{x})}\left[\left\|\sum_{i=1}^n \mathbf{x}_0^{(i)} - n\boldsymbol{b}_z\right\| \geq v_z + \frac{1 - e^{-2(T-k\eta)}}{2e^{-(T-k\eta)}} \cdot n\epsilon\right]$$

$$\leq \mathbb{P}_{\left\{\mathbf{x}_0^{(i)}\right\}_{i=1}^n \sim q^{(n)}(\cdot|\boldsymbol{x})}\left[\left\|\sum_{i=1}^n \mathbf{x}_0^{(i)} - n\boldsymbol{b}_z\right\| \geq v_z + \frac{1 - e^{-2(T-k\eta)}}{2e^{-(T-k\eta)}} \cdot n\epsilon\right] + D_{\mathrm{TV}}(q^{(n)}(\cdot|\boldsymbol{x}), q_z^{(n)}(\cdot|\boldsymbol{x}))$$

$$\leq \mathbb{P}_{\left\{\mathbf{x}_0^{(i)}\right\}_{i=1}^{N_k} \sim q^{(n)}(\cdot|\boldsymbol{x})}\left[\left\|\sum_{i=1}^n \mathbf{x}_0^{(i)} - n\boldsymbol{b}_z\right\| \geq v_z + \frac{1 - e^{-2(T-k\eta)}}{2e^{-(T-k\eta)}} \cdot n\epsilon\right] + n \cdot D_{\mathrm{TV}}(q(\cdot|\boldsymbol{x}), q_z(\cdot|\boldsymbol{x})).$$

$$\tag{34}$$

Consider the first term, we have the following relation

$$
\left\| \sum_{i=1}^{n} \mathbf{x}_0^{(i)} - n \boldsymbol{b}_z \right\| \geq v_z + \frac{1 - e^{-(T-k\eta)}}{2e^{-2(T-k\eta)}} \cdot n\epsilon
$$

$$
\Rightarrow \left\| \sum_{i=1}^{n} \mathbf{x}_0^{(i)} - n\boldsymbol{b} \right\| \geq \left\| \sum_{i=1}^{n} \mathbf{x}_0^{(i)} - n\boldsymbol{b}_z \right\| - n \left\| \boldsymbol{b} - \boldsymbol{b}_z \right\|
$$

$$
\geq v_z + \frac{1 - e^{-2(T-k\eta)}}{2e^{-(T-k\eta)}} \cdot n\epsilon - n \left\| \boldsymbol{b} - \boldsymbol{b}_z \right\| \tag{35}
$$

$$
= v + \frac{1 - e^{-2(T-k\eta)}}{2e^{-(T-k\eta)}} \cdot n\epsilon - n \left\| \boldsymbol{b} - \boldsymbol{b}_z \right\| + (v_z - v)
$$

$$
\geq v + \frac{1 - e^{-2(T-k\eta)}}{2e^{-(T-k\eta)}} \cdot n\epsilon - n \cdot W_2(q(\cdot|\boldsymbol{x}), q_z(\cdot|\boldsymbol{x})) - \sqrt{nd\mu^{-1}},
$$

where the last inequality follows from

$$
\| \boldsymbol{b} - \boldsymbol{b}_z \| = \left\| \int (q(\boldsymbol{x}_0|\boldsymbol{x}) - q_z(\boldsymbol{x}_0|\boldsymbol{x})) \, \boldsymbol{x}_0 \mathrm{d}\boldsymbol{x}_0 \right\| = \left\| \int (\boldsymbol{x}_0 - \boldsymbol{x}_z) \, \gamma(\boldsymbol{x}_0, \boldsymbol{x}_z) \mathrm{d}(\boldsymbol{x}_0, \boldsymbol{x}_z) \right\|
$$

$$
\leq \left( \int \gamma(\boldsymbol{x}_0, \boldsymbol{x}_z) \mathrm{d}(\boldsymbol{x}_0, \boldsymbol{x}_z) \right)^{1/2} \cdot \left( \int \gamma(\boldsymbol{x}_0, \boldsymbol{x}_z) \cdot \| \boldsymbol{x}_0 - \boldsymbol{x}_z \|^2 \, \mathrm{d}(\boldsymbol{x}_0, \boldsymbol{x}_z) \right)^{1/2} \tag{36}
$$

$$
\leq W_2(q(\cdot|\boldsymbol{x}), q_z(\cdot|\boldsymbol{x})),
$$

and

$$
v = n \cdot \mathbb{E}_{\left\{ \mathbf{x}_0^{(i)} \right\}_{i=1}^{n} \sim q^{(n)}(\cdot|\boldsymbol{x})} \left\| \frac{1}{n} \sum_{i=1}^{n} \mathbf{x}_0^{(i)} - \boldsymbol{b} \right\| \leq n \cdot \sqrt{\mathrm{var}\left( \frac{1}{n} \sum_{i=1}^{n} \mathbf{x}_0^{(i)} \right)}
$$

$$
= \sqrt{n \mathrm{var}\left( \mathbf{x}_0^{(1)} \right)} \leq \sqrt{nd\mu^{-1}}
$$

deduced by Lemma 23. By requiring

$$
W_2(q(\cdot|\boldsymbol{x}), q_z(\cdot|\boldsymbol{x})) \leq \frac{1 - e^{-2(T-k\eta)}}{8e^{-(T-k\eta)}} \cdot \epsilon \quad \text{and} \quad n \geq \frac{64 e^{-2(T-k\eta)} d}{\left( 1 - e^{-2(T-k\eta)} \right)^2 \mu\epsilon^2} \tag{37}
$$

in Eq. 35, we have

$$
\mathbb{P}_{\left\{ \mathbf{x}_0^{(i)} \right\}_{i=1}^{n} \sim q^{(n)}(\cdot|\boldsymbol{x})} \left[ \left\| \sum_{i=1}^{n} \mathbf{x}_0^{(i)} - n\boldsymbol{b}_z \right\| \geq v_z + \frac{1 - e^{-2(T-k\eta)}}{2e^{-(T-k\eta)}} \cdot n\epsilon \right]
$$

$$
\leq \mathbb{P}_{\left\{ \mathbf{x}_0^{(i)} \right\}_{i=1}^{n} \sim q^{(n)}(\cdot|\boldsymbol{x})} \left[ \left\| \sum_{i=1}^{n} \mathbf{x}_0^{(i)} - n\boldsymbol{b} \right\| \geq v + \frac{1 - e^{-2(T-k\eta)}}{4e^{-(T-k\eta)}} \cdot n\epsilon \right] \tag{38}
$$

According to Lemma 16, the LSI constant of

$$
\sum_{i=1}^{n} \mathbf{x}_0^{(i)} \sim \underbrace{q * q \cdots * q}_{n}
$$

is $\mu/n$. Besides, considering the function $F(\boldsymbol{x}) = \|\boldsymbol{x}\| : \mathbb{R}^d \to \mathbb{R}$ is 1-Lipschitz because

$$
\|F\|_{\mathrm{Lip}} = \sup_{\boldsymbol{x} \neq \boldsymbol{y}} \frac{|F(\boldsymbol{x}) - F(\boldsymbol{y})|}{\|\boldsymbol{x} - \boldsymbol{y}\|} = \sup_{\boldsymbol{x} \neq \boldsymbol{y}} \frac{|\|\boldsymbol{x}\| - \|\boldsymbol{y}\||}{\|(\boldsymbol{x} - \boldsymbol{y})\|} = 1,
$$

we have

$$
\mathbb{P}_{\left\{ \mathbf{x}_0^{(i)} \right\}_{i=1}^{n} \sim q^{(n)}(\cdot|\boldsymbol{x})} \left[ \left\| \sum_{i=1}^{n} \mathbf{x}_0^{(i)} - n\boldsymbol{b} \right\| \geq v + \frac{1 - e^{-2(T-k\eta)}}{4e^{-(T-k\eta)}} \cdot n\epsilon \right]
$$

$$
\leq \exp \left\{ -\frac{(1 - e^{-(T-k\eta)})^2 n\epsilon^2 \mu}{32 e^{-2(T-k\eta)}} \right\} \tag{39}
$$

due to Lemma 18. Plugging Eq. 39 and Eq. 38 into Eq. 34 and Eq. 33, we have

$$
\mathbb{P}_{\left\{\mathbf{x}_0^{(i)}\right\}_{i=1}^n \sim q_z^{(n)}(\cdot|\boldsymbol{x})}\left[\left\|\frac{1}{n}\sum_{i=1}^n \mathbf{v}_i(\boldsymbol{x}) - \frac{1}{n}\mathbb{E}\left[\sum_{i=1}^n \mathbf{v}_i(\boldsymbol{x})\right]\right\| \geq \mathbb{E}\left\|\frac{1}{n}\sum_{i=1}^n \mathbf{v}_i(\boldsymbol{x}) - \mathbb{E}\mathbf{v}_1(\boldsymbol{x})\right\| + \epsilon\right]
$$

$$
\leq \exp\left\{-\frac{(1-e^{-(T-k\eta)})^2 n\epsilon^2\mu}{32e^{-2(T-k\eta)}}\right\} + n\cdot D_{\mathrm{TV}}(q(\cdot|\boldsymbol{x}), q_z(\cdot|\boldsymbol{x})).
$$
(40)

Besides, we have

$$
\mathbb{E}_{\left\{\mathbf{x}_0^{(i)}\right\}_{i=1}^n \sim q_z^n(\cdot|\boldsymbol{x})}\left\|\frac{1}{n}\sum_{i=1}^n \mathbf{v}_i(\boldsymbol{x}) - \mathbb{E}\mathbf{v}_1(\boldsymbol{x})\right\| \leq \sqrt{\mathrm{var}\left(\frac{1}{n}\sum_{i=1}^n \mathbf{v}_i\right)}
$$

$$
=\sqrt{\frac{\mathrm{var}(\mathbf{v}_1)}{n}} = \frac{2e^{-(T-k\eta)}}{1-e^{-2(T-k\eta)}}\sqrt{\frac{\mathrm{var}(\mathbf{x}_0)}{n}}.
$$

Suppose the optimal coupling of $q_z(\cdot|\boldsymbol{x})$ and $q(\cdot|\boldsymbol{x})$ is $\gamma_z \in \Gamma_{\mathrm{opt}}\left(q_z(\cdot|\boldsymbol{x}), q(\cdot|\boldsymbol{x})\right)$, then we have

$$
\mathrm{var}_{q_z(\cdot|\boldsymbol{x})}(\mathbf{x}_0) = \int q_z(\boldsymbol{x}_z|\boldsymbol{x})\left\|\boldsymbol{x}_z - \boldsymbol{b}_z\right\|^2 \mathrm{d}\boldsymbol{x}_z = \int \left\|\boldsymbol{x}_z - \boldsymbol{b}_z\right\|^2 \mathrm{d}\gamma(\boldsymbol{x}_z, \boldsymbol{x}_0)
$$

$$
= \int \left\|\boldsymbol{x}_z - \boldsymbol{x}_0 + \boldsymbol{x}_0 - \boldsymbol{b} + \boldsymbol{b} - \boldsymbol{b}_z\right\|^2 \mathrm{d}\gamma(\boldsymbol{x}_z, \boldsymbol{x}_0)
$$

$$
\leq 3\int \left\|\boldsymbol{x}_z - \boldsymbol{x}_0\right\|^2 \mathrm{d}\gamma(\boldsymbol{x}_z, \boldsymbol{x}_0) + 3\int \left\|\boldsymbol{x}_0 - \boldsymbol{b}\right\|^2 \mathrm{d}\gamma(\boldsymbol{x}_z, \boldsymbol{x}_0) + 3\left\|\boldsymbol{b} - \boldsymbol{b}_z\right\|^2
$$

$$
\leq 6W_2^2(\tilde{q}, q_z) + 3d\mu^{-1}
$$

where the last inequality follows from Eq. 36 and Lemma 23. By requiring $W_2^2(\tilde{q}, q_z) \leq \frac{d}{6\mu}$, we have

$$
\frac{2e^{-(T-k\eta)}}{1-e^{-2(T-k\eta)}}\sqrt{\frac{\mathrm{var}_{q_z}(\mathbf{x}_0)}{n}} \leq \frac{4}{\sqrt{n}}\cdot\frac{e^{-(T-k\eta)}}{1-e^{-2(T-k\eta)}}\cdot\sqrt{\frac{d}{\mu}}.
$$

Combining this result with Eq. 40, we have

$$
\mathbb{P}_{\left\{\mathbf{x}_0^{(i)}\right\}_{i=1}^n \sim q_z^n(\cdot|\boldsymbol{x})}\left[\left\|\frac{1}{n}\sum_{i=1}^n \mathbf{v}_i(\boldsymbol{x}) - \frac{1}{n}\mathbb{E}\left[\sum_{i=1}^{N_k} \mathbf{v}_i(\boldsymbol{x})\right]\right\| \geq \frac{4}{\sqrt{n}}\cdot\frac{e^{-(T-k\eta)}}{1-e^{-2(T-k\eta)}}\cdot\sqrt{\frac{d}{\mu}} + \epsilon\right]
$$

$$
\leq \exp\left\{-\frac{(1-e^{-(T-k\eta)})^2 n\epsilon^2\mu}{32e^{-2(T-k\eta)}}\right\} + n\cdot D_{\mathrm{TV}}(q(\cdot|\boldsymbol{x}), q_z(\cdot|\boldsymbol{x})).
$$

By requiring

$$
\frac{4}{\sqrt{n}}\cdot\frac{e^{-(T-k\eta)}}{1-e^{-2(T-k\eta)}}\cdot\sqrt{\frac{d}{\mu}} \leq \epsilon \quad \Rightarrow \quad n \geq \frac{16e^{-2(T-k\eta)}d}{\left(1-e^{-2(T-k\eta)}\right)^2\mu\epsilon^2},
$$

$$
-\frac{(1-e^{-(T-k\eta)})^2 n\epsilon^2\mu}{32e^{-2(T-k\eta)}} \leq -\lfloor T/\eta\rfloor\cdot\frac{2}{\delta} \quad \Rightarrow \quad n \geq \lfloor T/\eta\rfloor\cdot\frac{64e^{-2(T-k\eta)}}{(1-e^{-(T-k\eta)})^2\mu\epsilon^2\delta} \quad (41)
$$

$$
\text{and} \quad D_{\mathrm{TV}}(\tilde{q}, q_z) \leq \frac{\delta}{2n\cdot\lfloor T/\eta\rfloor}.
$$

we have

$$
\mathbb{P}_{\left\{\mathbf{x}_0^{(i)}\right\}_{i=1}^n \sim q_z^{(n)}(\cdot|\boldsymbol{x})}\left[\left\|\frac{1}{n}\sum_{i=1}^n \mathbf{v}_i(\boldsymbol{x}) - \frac{1}{n}\mathbb{E}\left[\sum_{i=1}^n \mathbf{v}_i(\boldsymbol{x})\right]\right\| \geq 2\epsilon\right] \leq \exp\left(-\frac{1}{\delta/(2\lfloor T/\eta\rfloor)}\right) + \frac{\delta}{2\lfloor T/\eta\rfloor}.
$$

Combining the choice of $n$ and the gap between $q(\cdot|\boldsymbol{x})$ and $q_z(\cdot|\boldsymbol{x})$ in Eq. 37 and Eq. 41, we have

$$
\begin{cases}
n \geq \dfrac{64 e^{-2(T-k\eta)} d}{\left(1 - e^{-2(T-k\eta)}\right)^2 \mu \epsilon^2} \\[2ex]
n \geq \lfloor T/\eta \rfloor \cdot \dfrac{64 e^{-2(T-k\eta)}}{(1 - e^{-(T-k\eta)})^2 \mu \epsilon^2 \delta} \\[2ex]
n \geq \dfrac{16 e^{-2(T-k\eta)} d}{\left(1 - e^{-2(T-k\eta)}\right)^2 \mu \epsilon^2}
\end{cases}
\quad \text{and} \quad
\begin{cases}
W_2(q(\cdot|\boldsymbol{x}), q_z(\cdot|\boldsymbol{x})) \leq \dfrac{1 - e^{-2(T-k\eta)}}{8 e^{-(T-k\eta)}} \cdot \epsilon \\[2ex]
W_2^2(q(\cdot|\boldsymbol{x}), q_z(\cdot|\boldsymbol{x})) \leq d/(6\mu) \\[2ex]
D_{\mathrm{TV}}(q(\cdot|\boldsymbol{x}), q_z(\cdot|\boldsymbol{x})) \leq \dfrac{\delta}{2n \cdot \lfloor T/\eta \rfloor}
\end{cases}.
$$

$$(42)$$

Without loss of generality, we suppose $\eta \leq 1/2$, due to the range of $e^{-2(T-k\eta)}$ as follows

$$
e^{-2(T-k\eta)} \leq e^{-2\eta} \leq 1 - \eta \;\Rightarrow\; \frac{e^{-2(T-k\eta)}}{(1 - e^{-2(T-k\eta)})^2} \leq \eta^{-2},
$$

we obtain the sufficient condition for achieving Eq. 42 is

$$
n \geq 64 T d \mu^{-1} \eta^{-3} \epsilon^{-2} \delta^{-1} \geq \max\left\{ 64 d \mu^{-1} \eta^{-2} \epsilon^{-2},\, 64 T \eta^{-3} \mu^{-1} \epsilon^{-2} \delta^{-1},\, 16 d \mu^{-1} \eta^{-2} \epsilon^{-2} \right\}
$$

and $\quad D_{\mathrm{TV}}(\tilde{q}, q_z) \leq \dfrac{1}{2} \cdot \delta \eta n^{-1} T^{-1} \Leftarrow D_{\mathrm{KL}}(q_z \| \tilde{q}) \leq \dfrac{1}{2} \cdot \delta^2 \eta^2 n^{-2} T^{-2} \leq 2^{-13} \cdot T^{-4} d^{-2} \mu^2 \eta^8 \epsilon^4 \delta^4.$

Hence, the proof is completed. $\qquad \square$

**Lemma 9.** *With Algorithm 1 and notation list 1, if we choose the initial distribution of the k-th inner loop to be*

$$
q'_{T-k\eta}(\boldsymbol{x}_0|\boldsymbol{x}) \propto \exp\left( -\frac{\left\| \boldsymbol{x} - e^{-(T-k\eta)} \boldsymbol{x}_0 \right\|^2}{2\left(1 - e^{-2(T-k\eta)}\right)} \right),
$$

*then suppose the the LSI constant of $q_{T-k\eta}$ is $C_{\mathrm{LSI},k}$, their KL divergence can be upper bounded as*

$$
D_{\mathrm{KL}}\left(q'_{T-k\eta}(\cdot|\boldsymbol{x}) \| q_{T-k\eta}(\cdot|\boldsymbol{x})\right) \leq \frac{L^2}{2 C_{\mathrm{LSI},k}} \cdot e^{2(T-k\eta)} \left(d + \|\boldsymbol{x}\|^2\right).
$$

*Proof.* According to the fact that the LSI constant of $q_{T-k\eta}$ is $C_{\mathrm{LSI},k}$, then we have

$$
D_{\mathrm{KL}}\left(q'_{T-k\eta}(\cdot|\boldsymbol{x}) \| q_{T-k\eta}(\cdot|\boldsymbol{x})\right) \leq (2 C_{\mathrm{LSI},k})^{-1} \cdot \int q'_{T-k\eta}(\boldsymbol{x}_0|\boldsymbol{x}) \left\| \nabla f(\boldsymbol{x}_0) \right\|^2 \, \mathrm{d}\boldsymbol{x}_0
$$

$$
\leq L^2 (2 C_{\mathrm{LSI},k})^{-1} \cdot \mathbb{E}_{\mathbf{x}_0 \sim q'_{T-k\eta}(\cdot|\boldsymbol{x})}\left[\|\mathbf{x}_0\|^2\right] = L^2 (2 C_{\mathrm{LSI},k})^{-1} \cdot \left(\mathrm{var}(\mathbf{x}_0) + \|\mathbb{E}[\mathbf{x}_0]\|^2\right).
$$

Because $q'_{T-k\eta}(\cdot|\boldsymbol{x})$ is a high-dimensional Gaussian, its mean value satisfies $\mathbb{E}[\mathbf{x}_0] = e^{(T-k\eta)} \boldsymbol{x}$. Besides, we have

$$
-\nabla^2 \ln q'_{T-k\eta}(\boldsymbol{x}_0) = \frac{e^{-2(T-k\eta)}}{1 - e^{-2(T-k\eta)}} \cdot \boldsymbol{I}.
$$

According to Lemma 20, $q'_{T-k\eta}(\cdot|\boldsymbol{x})$ satisfies the log-Sobolev inequality (and the Poincaré inequality). Follows from Lemma 23, we have

$$
\mathrm{var}_{\mathbf{x}_0 \sim q'_{T-k\eta}(\cdot|\boldsymbol{x})}[\mathbf{x}_0] \leq d \cdot (1 - e^{-2(T-k\eta)}) \cdot e^{2(T-k\eta)} \leq d e^{2(T-k\eta)}.
$$

Hence, we have

$$
D_{\mathrm{KL}}\left(q'_{T-k\eta}(\cdot|\boldsymbol{x}) \| q_{T-k\eta}(\cdot|\boldsymbol{x})\right) \leq \frac{L^2}{2 C_{\mathrm{LSI},k}} \cdot e^{2(T-k\eta)} \left(d + \|\boldsymbol{x}\|^2\right)
$$

and the proof is completed. $\qquad \square$

**Lemma 10.** *(Errors from the inner loop sampling task) Suppose Assumption [$A_1$],[$A_2$],[$A_4$] hold. With Algorithm 1 notation list 1 and suitable $\eta = C_1 \left(d + m_2^2\right)^{-1} \epsilon^2$, there is*

$$
\sum_{k=0}^{N-1} \eta \cdot \mathbb{E}_{\hat{P}_T}\left[\left\| \mathbf{v}_k(\mathbf{x}_{k\eta}) - 2 \nabla \ln p_{T-k\eta}(\mathbf{x}_{k\eta}) \right\|^2\right] \leq 20 \epsilon^2 \ln \frac{C_0}{2\epsilon^2},
$$

*with a probability at least $1 - \delta$. The gradient complexity of achieving this result is*

$$\max\left(C_3 C_5, C_3' C_5'\right) \cdot C_1^{-1} C_0 \cdot (d + m_2^2)^{18} \epsilon^{-16n-83} \exp\left(5C_2 \epsilon^{-16n}\right) \delta^{-6}$$

*where constants $C_i$ and $C_i'$ are independent with $d$ and $\epsilon$.*

*Proof.* To upper bound it more precisely, we divide the backward process into two stages.

**Stage 1: when $e^{-2(T-k\eta)} \leq 2L/(1+2L)$.** It implies the iteration $k$ satisfies

$$k \leq \frac{1}{2\eta}\left(2T - \ln\frac{1+2L}{2L}\right) := N_1. \tag{43}$$

In this condition, we set

$$q_{T-k\eta}(\boldsymbol{x}_0|\boldsymbol{x}) \propto \exp(-g_{T-k\eta}(\boldsymbol{x}_0|\boldsymbol{x})) := \exp\left(-f_*(\boldsymbol{x}_0) - \frac{\left\|\boldsymbol{x} - e^{-(T-k\eta)}\boldsymbol{x}_0\right\|^2}{2\left(1 - e^{-2(T-k\eta)}\right)}\right). \tag{44}$$

Hence, we can reformulate $g_{T-k\eta}(\boldsymbol{x}_0|\boldsymbol{x})$ as

$$g_{T-k\eta}(\boldsymbol{x}_0|\boldsymbol{x}) = \underbrace{f_*(\boldsymbol{x}_0) + \frac{e^{-2(T-k\eta)}}{3(1 - e^{-2(T-k\eta)})}\left\|\boldsymbol{x}_0\right\|^2}_{\text{part 1}}$$

$$+ \underbrace{\frac{e^{-2(T-k\eta)}}{6(1 - e^{-2(T-k\eta)})}\left\|\boldsymbol{x}_0\right\|^2 - \frac{e^{-(T-k\eta)}}{(1 - e^{-2(T-k\eta)})}\boldsymbol{x}_0^\top \boldsymbol{x} + \frac{\left\|\boldsymbol{x}\right\|^2}{2(1 - e^{-2(T-k\eta)})}}_{\text{part 2}}.$$

According to Assumption [$\mathbf{A}_4$], we know part 1 and part 2 are both strongly convex outside the ball with radius $R(e^{-2(T-k\eta)}/(6(1 - e^{-2(T-k\eta)})))$. With Lemma 22, the function $g_{T-k\eta}(\cdot|\boldsymbol{x})$ is $e^{-2(T-k\eta)}/(3(1 - e^{-2(T-k\eta)}))$-strongly convex outside the ball. Besides, the gradient Lipschitz constant of $g_{T-k\eta}(\cdot|\boldsymbol{x})$ can be upper bounded as

$$\nabla^2 g_{T-k\eta}(\boldsymbol{x}_0|\boldsymbol{x}) \preceq \nabla^2 f_*(\boldsymbol{x}_0) + \frac{e^{-2(T-k\eta)}}{1 - e^{-2(T-k\eta)}} \cdot \boldsymbol{I} \preceq \left(L + \frac{e^{-2(T-k\eta)}}{1 - e^{-2(T-k\eta)}}\right) \cdot \boldsymbol{I} \preceq 3L\boldsymbol{I}$$

where the last inequality follows from the choice of $e^{-2(T-k\eta)}$ in the stage.

When the total time satisfies $\exp(-T/2) = 2\epsilon^2/C_0$, we have

$$C_{\mathrm{LSI},k}^{-1} \leq 6(1 - e^{-2(T-k\eta)})e^{2(T-k\eta)} \cdot \exp\left(48L \cdot R^2\left(\frac{e^{-2(T-k\eta)}}{6(1 - e^{-2(T-k\eta)})}\right)\right)$$

$$\leq 6\exp\left(2(T - k\eta) + 48L \cdot \left(6(1 - e^{-2(T-k\eta)})e^{2(T-k\eta)}\right)^{2n}\right)$$

$$\leq 6\exp\left(2(T - k\eta) + 48L \cdot 6^{2n} \cdot e^{4n(T-k\eta)}\right).$$

The second inequality follows from Assumption [$\mathbf{A}_4$], and the last inequality follows from the setting $n, L \geq 1$ without loss of generality.

$$C_{\mathrm{LSI},k}^{-1} \leq 6 \cdot 2^{-4} \cdot C_0^4 \epsilon^{-8} \exp\left(48L \cdot 6^{2n} \cdot 2^{-8n} \cdot C_0^{8n} \cdot \epsilon^{-16n}\right) = C_6 \epsilon^{-8} \exp\left(C_2 \cdot \epsilon^{-16n}\right) \tag{45}$$

For Term 1, due to Lemma 8, if we set the step size of outer loop to be

$$\eta = C_1\left(d + m_2^2\right)^{-1}\epsilon^2,$$

the sample number of each iteration $k$ satisfies

$$n_k = C_3 \cdot \left(d + m_2^2\right)^4 \epsilon^{-18} \exp\left(C_2 \cdot \epsilon^{-16n}\right) \delta^{-1} = 64 \cdot C_0 C_1^{-3} C_6 \left(d + m_2^2\right)^4 \epsilon^{-18} \exp\left(C_2 \cdot \epsilon^{-16n}\right) \delta^{-1}$$

$$\geq 64 \cdot 2\ln\frac{C_0}{2\epsilon^2} \cdot d \cdot \left(C_1(d + m_2^2)^{-1}\epsilon^2\right)^{-3} \cdot \epsilon^{-2} \cdot C_6 \epsilon^{-8} \exp\left(C_2 \cdot \epsilon^{-16n}\right) \cdot \delta \geq 64Td\eta^{-3}\epsilon^{-2}\delta^{-1}C_{\mathrm{LSI},k}^{-1}$$

and the accuracy of the inner loop meets

$$
\begin{aligned}
&D_{\mathrm{KL}}\left(q'_{T-k\eta}(\cdot|\boldsymbol{x})\|q_{T-k\eta}(\cdot|\boldsymbol{x})\right) \le C_4\left(d+m_2^2\right)^{-10}\epsilon^{44}\exp\left(-2C_2\cdot\epsilon^{-16n}\right)\delta^4\\
=&2^{-13}\cdot C_0^{-4}\cdot C_1^8\left(d+m_2^2\right)^{-10}\epsilon^{28}\delta^4\cdot C_6^{-2}\epsilon^{16}\exp\left(-2C_2\cdot\epsilon^{-16n}\right)\\
\le&2^{-13}\cdot\left(2\ln\frac{C_0}{2\epsilon^2}\right)^{-4}\cdot d^{-2}\cdot\epsilon^4\cdot\delta^4\cdot C_1^8\left(d+m_2^2\right)^{-8}\epsilon^{16}\cdot C_{\mathrm{LSI},k}^2 \le 2^{-13}\cdot T^{-4}d^{-2}\epsilon^4\delta^4\eta^8 C_{\mathrm{LSI},k}^2
\end{aligned}
\tag{46}
$$

when $\epsilon^2 \le 1/2$. In this condition, we have

$$
\begin{aligned}
\mathbb{P}&\left[\left\|\mathbf{v}_k(\boldsymbol{x})-2\mathbb{E}_{\mathbf{x}_0'\sim q'_{T-k\eta}(\cdot|\boldsymbol{x})}\left[-\frac{\boldsymbol{x}-e^{-(T-k\eta)}\mathbf{x}_0}{\left(1-e^{-2(T-k\eta)}\right)}\right]\right\|^2 \le 4\epsilon^2\right]\\
&\ge 1-\exp\left(-\frac{1}{\delta/(2\lfloor T/\eta\rfloor)}\right)-\frac{\delta}{2\lfloor T/\eta\rfloor}\ge 1-\frac{\delta}{\lfloor T/\eta\rfloor},
\end{aligned}
\tag{47}
$$

where $q'_{T-k\eta}(\cdot|\boldsymbol{x})$ denotes the underlying distribution of output particles of the $k$-th inner loop. To achieve

$$
D_{\mathrm{KL}}\left(q'_{T-k\eta}(\cdot|\boldsymbol{x}_{k\eta})\|q_{T-k\eta}(\cdot|\boldsymbol{x}_{k\eta})\right) \le C_4\left(d+m_2^2\right)^{-10}\epsilon^{44}\exp\left(-2C_2\cdot\epsilon^{-16n}\right)\delta^4 \coloneqq \delta_{\mathrm{KL}},
$$

Lemma 19 requires the step size to satisfy

$$
\begin{aligned}
\tau_k &\le 2^{-4}\cdot 3^{-2}\cdot L^{-2}C_4 C_6^{-1}\left(d+m_2^2\right)^{-11}\epsilon^{52}\exp\left(-3C_2\epsilon^{-16n}\right)\delta^4\\
&\le C_6^{-1}\epsilon^8\exp\left(-C_2\epsilon^{-16n}\right)\cdot\frac{1}{4(3L)^2}\cdot\frac{1}{4d}\cdot C_4\left(d+m_2^2\right)^{-10}\epsilon^{44}\exp\left(-2C_2\cdot\epsilon^{-16n}\right)\delta^4\\
&\le\frac{C_{\mathrm{LSI},k}}{4\|\nabla^2 g_{T-k\eta}(\cdot|\boldsymbol{x})\|_2^2}\cdot\frac{1}{4d}\cdot\delta_{\mathrm{KL}}
\end{aligned}
$$

and the iteration number $Z_k$ meets

$$
Z_k \ge \frac{1}{C_{\mathrm{LSI},k}\tau_k}\cdot\ln\frac{2D_{\mathrm{KL}}\left(q'_{T-k\eta,0}(\cdot|\boldsymbol{x})\|q_{T-k\eta}(\cdot|\boldsymbol{x})\right)}{\delta_{\mathrm{KL}}}
$$

where $q'_{T-k\eta,0}(\cdot|\boldsymbol{x})$ denotes the initial distribution of the $k$-th inner loop. By choosing $\tau_k$ to be its upper bound and the initial distribution of $k$-th inner loop to be

$$
q'_{T-k\eta,0}(\boldsymbol{x}_0|\boldsymbol{x}) \propto \exp\left(-\frac{\left\|\boldsymbol{x}-e^{-(T-k\eta)}\boldsymbol{x}_0\right\|^2}{2\left(1-e^{-2(T-k\eta)}\right)}\right),
$$

we have

$$
\begin{aligned}
D_{\mathrm{KL}}\left(q'_{T-k\eta,0}(\cdot|\boldsymbol{x})\|q_{T-k\eta}(\cdot|\boldsymbol{x})\right) &\le\frac{L^2}{2C_{\mathrm{LSI},k}}\cdot e^{2(T-k\eta)}\left(d+\|\boldsymbol{x}\|^2\right)\\
&\le 2^{-1}L^2 C_6\epsilon^{-8}\exp\left(C_2\cdot\epsilon^{-16n}\right)\cdot e^{2(T-k\eta)}\left(d+\|\boldsymbol{x}\|^2\right)
\end{aligned}
$$

with Lemma 9 and Eq. 45. It implies the iteration number $Z_k$ of inner loops to be required as

$$
\begin{aligned}
Z_k \ge&\, C_5\cdot\left(d+m_2^2\right)^{12}\epsilon^{-16n-61}\exp\left(4C_2\epsilon^{-16n}\right)\cdot\left(d+\|\boldsymbol{x}\|^2\right)\delta^{-5}\\
=&\,2^8\cdot 3^4\cdot 5^2 L^2\cdot C_2 C_4^{-1}C_6^2\ln\left(2^{-4}L^2 C_0^4 C_4^{-1}C_6\right)\cdot\left(d+m_2^2\right)^{12}\epsilon^{-16n-61}\exp\left(4C_2\epsilon^{-16n}\right)\cdot\left(d+\|\boldsymbol{x}\|^2\right)\delta^{-5}\\
=&\,C_6\epsilon^{-8}\exp\left(C_2\cdot\epsilon^{-16n}\right)\cdot\left(2^4\cdot 3^2 L^2 C_4^{-1}C_6\cdot\left(d+m_2^2\right)^{11}\epsilon^{-52}\exp\left(3C_2\epsilon^{-16n}\right)\delta^{-4}\right)\\
&\cdot\left(\ln\left(2^{-4}L^2 C_0^4 C_4^{-1}C_6\right)+3C_2\epsilon^{-16n}+60\ln\frac{1}{\epsilon}+4\ln\frac{1}{\delta}+10\ln(d+m_2^2)+\ln(d+\|\boldsymbol{x}\|^2)\right)\\
\ge&\,\frac{1}{C_{\mathrm{LSI},k}\tau_k}\cdot\ln\frac{2D_{\mathrm{KL}}\left(q'_{T-k\eta,0}(\cdot|\boldsymbol{x})\|q_{T-k\eta}(\cdot|\boldsymbol{x})\right)}{\delta_{\mathrm{KL}}}.
\end{aligned}
$$

when $\ln(1/\epsilon) \ge 2$ and $\ln d \ge 2$ without loss of generality.

A sufficient condition to obtain Term $2 \le \epsilon^2$ is to make the following inequality establish

$$D_{\mathrm{KL}}\left(q'_{T-k\eta}(\cdot|\boldsymbol{x})\|q_{T-k\eta}(\cdot|\boldsymbol{x})\right) \le 2^{-3}\cdot\eta^2\epsilon^2\cdot C_6^{-1}\epsilon^8\exp\left(-C_2\epsilon^{-16n}\right)$$

which will be dominated by Eq. 46 in almost cases obviously.

Hence, combining Eq. 17, Eq. 47 and Eq. 18, there is

$$\sum_{k=0}^{N_1}\eta\cdot\mathbb{E}_{\hat{P}_T}\left[\|\mathbf{v}_k(\mathbf{x}_{k\eta})-2\nabla\ln p_{T-k\eta}(\mathbf{x}_{k\eta})\|^2\right] \le 10N_1\eta\cdot\epsilon^2 \tag{48}$$

with a probability at least $1 - N_1\cdot\delta/(\lfloor T/\eta\rfloor)$ which is obtained by uniformed bound. We require the gradient complexity in this stage will be

$$\begin{aligned}
\mathrm{cost} = \sum_{k=0}^{N_1}n_k\mathbb{E}_{\hat{P}_T}(Z_k) &= \sum_{k=0}^{N_1}C_3\cdot\left(d+m_2^2\right)^4\epsilon^{-18}\exp\left(C_2\cdot\epsilon^{-16n}\right)\delta^{-1} \\
&\quad\cdot\mathbb{E}_{\hat{P}_T}\left[C_5\cdot(d+m_2^2)^{12}\epsilon^{-16n-61}\exp\left(4C_2\epsilon^{-16n}\right)\cdot\left(d+\|\mathbf{x}_{k\eta}\|^2\right)\delta^{-5}\right] \\
&\le N_1\cdot C_3C_5(d+m_2^2)^{17}\epsilon^{-16n-79}\exp\left(5C_2\epsilon^{-16n}\right)\delta^{-6}
\end{aligned} \tag{49}$$

where the last inequality follows from Lemma 14.

**Stage 2: when $\frac{2L}{1+2L} \le e^{-2(T-k\eta)} \le 1$.** We have the LSI constant for this stage. It is a constant level LSI constant, which mean we should choose the sample and the iteration number similar to Stage 1. Therefore, for Term 1, by requiring

$$\begin{aligned}
n_k &= \frac{64}{L}\cdot C_0C_1^{-3}\cdot\left(d+m_2^2\right)^4\epsilon^{-10}\delta^{-1} \\
&\ge 64\cdot 2\ln\frac{C_0}{2\epsilon^2}\cdot d\cdot\left(C_1(d+m_2^2)^{-1}\epsilon^2\right)^{-3}\cdot\epsilon^{-2}\delta^{-1}\cdot L^{-1} \ge 64Td\eta^{-3}\epsilon^{-2}\delta^{-1}C_{\mathrm{LSI},k}^{-1}
\end{aligned}$$

and

$$D_{\mathrm{KL}}\left(q'_{T-k\eta}(\cdot|\boldsymbol{x})\|q_{T-k\eta}(\cdot|\boldsymbol{x})\right) \le 2^{-13}L^2\cdot C_0^{-4}C_1^8\cdot(d+m_2^2)^{-10}\epsilon^{28}\delta^4$$

$$\le 2^{-13}\cdot\left(2\ln\frac{C_0}{2\epsilon^2}\right)^{-4}\cdot d^{-2}\cdot\epsilon^4\delta^4\cdot C_1^8\left(d+m_2^2\right)^{-8}\epsilon^{16}\cdot C_{\mathrm{LSI},k}^2 \le 2^{-13}\cdot T^{-4}d^{-2}\epsilon^4\delta^4\eta^8C_{\mathrm{LSI},k}^2. \tag{50}$$

In this condition, we have

$$\mathbb{P}\left[\left\|\mathbf{v}_k(\boldsymbol{x})-2\mathbb{E}_{\mathbf{x}_0'\sim q'_{T-k\eta}(\cdot|\boldsymbol{x})}\left[-\frac{\boldsymbol{x}-e^{-(T-k\eta)}\mathbf{x}_0}{\left(1-e^{-2(T-k\eta)}\right)}\right]\right\|^2 \le 4\epsilon^2\right] \tag{51}$$

$$\ge 1-\exp\left(-\frac{1}{\delta/(2\lfloor T/\eta\rfloor)}\right)-\frac{\delta}{2\lfloor T/\eta\rfloor} \ge 1-\frac{\delta}{\lfloor T/\eta\rfloor},$$

where $q'_{T-k\eta}(\cdot|\boldsymbol{x})$ denotes the underlying distribution of output particles of the $k$-th inner loop. To achieve

$$D_{\mathrm{KL}}\left(q'_{T-k\eta}(\cdot|\boldsymbol{x}_{k\eta})\|q_{T-k\eta}(\cdot|\boldsymbol{x}_{k\eta})\right) \le 2^{-13}L^2\cdot C_0^{-4}C_1^8\cdot(d+m_2^2)^{-10}\epsilon^{28}\delta^4 := \delta_{\mathrm{KL}},$$

Lemma 19 requires the step size to satisfy

$$\begin{aligned}
\tau_k &\le 2^{-17}\cdot 3^{-1}\Sigma_{k,\max}^{-1}\cdot L^2C_0^{-4}C_1^8\left(d+m_2^2\right)^{-11}\epsilon^{28}\delta^4 \\
&\le \frac{\Sigma_{k,\min}}{\Sigma_{k,\max}}\cdot\frac{1}{4\Sigma_{k,\max}}\cdot\frac{1}{4d}\cdot 2^{-13}L^2\cdot C_0^{-4}C_1^8\cdot(d+m_2^2)^{-10}\epsilon^{28}\delta^4 \\
&\le \frac{C_{\mathrm{LSI},k}}{4\|\nabla^2 g_{T-k\eta}(\cdot|\boldsymbol{x})\|_2^2}\cdot\frac{1}{4d}\cdot\delta_{\mathrm{KL}}
\end{aligned}$$

and the iteration number $Z_k$ meets

$$Z_k \ge \frac{1}{C_{\mathrm{LSI},k}\tau_k}\cdot\ln\frac{2D_{\mathrm{KL}}\left(q'_{T-k\eta,0}(\cdot|\boldsymbol{x})\|q_{T-k\eta}(\cdot|\boldsymbol{x})\right)}{\delta_{\mathrm{KL}}}$$

where $q'_{T-k\eta,0}(\cdot|\boldsymbol{x})$ denotes the initial distribution of the $k$-th inner loop. By choosing $\tau_k$ to be its upper bound and the initial distribution of $k$-th inner loop to be

$$q'_{T-k\eta,0}(\boldsymbol{x}_0|\boldsymbol{x}) \propto \exp\left(-\frac{\left\|\boldsymbol{x} - e^{-(T-k\eta)}\boldsymbol{x}_0\right\|^2}{2\left(1 - e^{-2(T-k\eta)}\right)}\right),$$

we have

$$D_{\mathrm{KL}}\left(q'_{T-k\eta,0}(\cdot|\boldsymbol{x})\|q_{T-k\eta}(\cdot|\boldsymbol{x})\right) \leq \frac{L^2}{2C_{\mathrm{LSI},k}} \cdot e^{2(T-k\eta)}\left(d + \|\boldsymbol{x}\|^2\right) \leq \frac{L}{2} \cdot e^{2(T-k\eta)}\left(d + \|\boldsymbol{x}\|^2\right)$$

with Lemma 9 and Eq. 14. It implies the iteration number $Z_k$ of inner loops to be required as

$$\begin{aligned}
Z_k &\geq C'_5 \cdot (d + m_2^2)^{12} \epsilon^{-29} \cdot \left(d + \|\boldsymbol{x}\|^2\right) \delta^{-5} \\
&= 2^{22} \cdot 3^4 \cdot 5 \cdot L^{-2} C_0^4 C_1^{-8} \ln\left(\frac{2^8}{L} \cdot C_0^8 C_1^{-8}\right) \cdot (d + m_2^2)^{12} \epsilon^{-29} \delta^{-5} \left(d + \|\boldsymbol{x}\|^2\right) \\
&= \frac{1}{\Sigma_{k,\min}} \cdot \left(2^{-17} \cdot 3^{-1} \Sigma_{k,\max}^{-1} \cdot L^2 C_0^{-4} C_1^8 \left(d + m_2^2\right)^{-11} \epsilon^{28} \delta^4\right)^{-1} \\
&\quad \cdot \left(\ln\left(\frac{2^8}{L} \cdot C_0^8 C_1^{-8}\right) + 36\ln\frac{1}{\epsilon} + 4\ln\frac{1}{\delta} + 10\ln(d + m_2^2) + \ln(d + \|\boldsymbol{x}\|^2)\right) \\
&\geq \frac{1}{C_{\mathrm{LSI},k}\tau_k} \cdot \ln\frac{2D_{\mathrm{KL}}\left(q'_{T-k\eta,0}(\cdot|\boldsymbol{x})\|q_{T-k\eta}(\cdot|\boldsymbol{x})\right)}{\delta_{\mathrm{KL}}}.
\end{aligned}$$

when $\ln(1/\epsilon) \geq 2$ and $\ln d \geq 2$ without loss of generality.

For Term 2, we denote an optimal coupling between $q_{T-k\eta}(\cdot|\boldsymbol{x})$ and $q'_{T-k\eta}(\cdot|\boldsymbol{x})$ to be

$$\gamma \in \Gamma_{\mathrm{opt}}(q_{T-k\eta}(\cdot|\boldsymbol{x}), q'_{T-k\eta}(\cdot|\boldsymbol{x})).$$

Hence, we have

$$\begin{aligned}
&\left\|\frac{2e^{-(T-k\eta)}}{1 - e^{-2(T-k\eta)}}\left[\mathbb{E}_{\mathbf{x}'_0 \sim q'_{T-k\eta}(\cdot|\boldsymbol{x})}[\mathbf{x}'_0] - \mathbb{E}_{\mathbf{x}_0 \sim q_{T-k\eta}(\cdot|\boldsymbol{x})}[\mathbf{x}_0]\right]\right\|^2 \\
&= \left\|\mathbb{E}_{(\mathbf{x}_0, \mathbf{x}'_0) \sim \gamma}\left[\frac{2e^{-(T-k\eta)}}{1 - e^{-2(T-k\eta)}} \cdot (\mathbf{x}_0 - \mathbf{x}'_0)\right]\right\|^2 \leq \frac{4e^{-2(T-k\eta)}}{(1 - e^{-2(T-k\eta)})^2} \cdot \mathbb{E}_{(\mathbf{x}_0, \mathbf{x}'_0) \sim \gamma}\left[\|\mathbf{x}_0 - \mathbf{x}'_0\|^2\right] \\
&= \frac{4e^{-2(T-k\eta)}}{(1 - e^{-2(T-k\eta)})^2} W_2^2\left(q_{T-k\eta}(\cdot|\boldsymbol{x}), q'_{T-k\eta}(\cdot|\boldsymbol{x})\right) \\
&\leq \frac{4e^{-2(T-k\eta)}}{\left(1 - e^{-2(T-k\eta)}\right)^2} \cdot \frac{2}{C_{\mathrm{LSI},k}} D_{\mathrm{KL}}\left(q'_{T-k\eta}(\cdot|\boldsymbol{x})\|q_{T-k\eta}(\cdot|\boldsymbol{x})\right) \\
&\leq 8\eta^{-2} C_{\mathrm{LSI},k}^{-1} D_{\mathrm{KL}}\left(q'_{T-k\eta}(\cdot|\boldsymbol{x})\|q_{T-k\eta}(\cdot|\boldsymbol{x})\right),
\end{aligned}$$
(52)

where the first inequality follows from Jensen's inequality, the second inequality follows from the Talagrand inequality and the last inequality follows from

$$e^{-2(T-k\eta)} \leq e^{-2\eta} \leq 1 - \eta \implies \frac{e^{-2(T-k\eta)}}{(1 - e^{-2(T-k\eta)})^2} \leq \eta^{-2},$$

when $\eta \leq 1/2$. Therefore, a sufficient condition to obtain Term $2 \leq \epsilon^2$ is to make the following inequality establish

$$D_{\mathrm{KL}}\left(q'_{T-k\eta}(\cdot|\boldsymbol{x})\|q_{T-k\eta}(\cdot|\boldsymbol{x})\right) \leq 2^{-3} \cdot \eta^2 \epsilon^2 \cdot L^{-1}$$

which will be dominated by Eq. 50 in almost cases obviously.

Hence, combining Eq. 17, Eq. 51 and Eq. 52, there is

$$\sum_{k=N_1+1}^{\lfloor T/\eta\rfloor} \eta \cdot \mathbb{E}_{\hat{P}_T}\left[\|\mathbf{v}_k(\mathbf{x}_{k\eta}) - 2\nabla\ln p_{T-k\eta}(\mathbf{x}_{k\eta})\|^2\right] \leq 10(\lfloor T/\eta\rfloor - N_1)\eta \cdot \epsilon^2 \tag{53}$$

with a probability at least $1 - (\lfloor T/\eta \rfloor - N_1) \cdot \delta/(\lfloor T/\eta \rfloor)$ which is obtained by uniformed bound. We require the gradient complexity in this stage will be

$$
\begin{aligned}
\text{cost} = \sum_{k=N_1+1}^{\lfloor T/\eta \rfloor} n_k \mathbb{E}_{\hat{P}_T}(Z_k) &= \sum_{k=N_1+1}^{\lfloor T/\eta \rfloor} \frac{64}{L} \cdot C_0 C_1^{-3} \cdot \left(d + m_2^2\right)^4 \epsilon^{-10} \delta^{-1} \\
&\quad \cdot \mathbb{E}_{\hat{P}_T} \left[ C_5' \cdot (d + m_2^2)^{12} \epsilon^{-29} \cdot \left(d + \|\mathbf{x}_{k\eta}\|^2\right) \delta^{-5} \right] \\
&\leq (\lfloor T/\eta \rfloor - N_1) \cdot C_3' C_5' (d + m_2^2)^{17} \epsilon^{-39} \delta^{-6}
\end{aligned}
\tag{54}
$$

where the last inequality follows from Lemma 14. Combining Eq. 49 and Eq. 54, we know the total gradient complexity will be less than

$$
\max\left(C_3 C_5, C_3' C_5'\right) \cdot C_1^{-1} C_0 \cdot (d + m_2^2)^{18} \epsilon^{-16n-83} \exp\left(5 C_2 \epsilon^{-16n}\right) \delta^{-6}.
$$

Hence the proof is completed. $\qquad\square$

## E    AUXILIARY LEMMAS

**Lemma 11.** *(Lemma 11 of Vempala & Wibisono (2019)) Assume $\nu = \exp(-f)$ is $L$-smooth. Then $\mathbb{E}_\nu \|\nabla f\|^2 \leq dL$.*

**Lemma 12.** *(Girsanov's theorem, Theorem 5.22 in Le Gall (2016)) Let $P_T$ and $Q_T$ be two probability measures on path space $\mathcal{C}\left([0,T], \mathbb{R}^d\right)$. Suppose under $P_T$, the process $(\tilde{\mathbf{x}}_t)_{t \in [0,T]}$ follows*

$$
\mathrm{d}\tilde{\mathbf{x}}_t = \tilde{b}_t \mathrm{d}t + \sigma_t \mathrm{d}\tilde{B}_t.
$$

*Under $Q_T$, the process $(\hat{\mathbf{x}}_t)_{t \in [0,T]}$ follows*

$$
\mathrm{d}\hat{\mathbf{x}}_t = \hat{b}_t \mathrm{d}t + \sigma_t \mathrm{d}\hat{B}_t \quad \text{and} \quad \hat{\mathbf{x}}_0 = \tilde{\mathbf{x}}_0.
$$

*We assume that for each $t \geq 0$, $\sigma_t \in \mathbb{R}^{d \times d}$ is a non-singular diffusion matrix. Then, provided that Novikov's condition holds*

$$
\mathbb{E}_{Q_T}\left[\exp\left(\frac{1}{2}\int_0^T \left\|\sigma_t^{-1}\left(\tilde{b}_t - \hat{b}_t\right)\right\|^2\right)\right] < \infty,
$$

*we have*

$$
\frac{\mathrm{d}P_T}{\mathrm{d}Q_T} = \exp\left(\int_0^T \sigma_t^{-1}\left(\tilde{b}_t - \hat{b}_t\right)\mathrm{d}B_t - \frac{1}{2}\int_0^T \left\|\sigma_t^{-1}\left(\tilde{b}_t - \hat{b}_t\right)\right\|^2 \mathrm{d}t\right).
$$

**Corollary 2.** *Plugging following settings*

$$
P_T := \tilde{P}_T^{p_T}, \ Q_T := \hat{P}_T, \ \tilde{b}_t := \mathbf{x}_t + \mathbf{v}_k(\mathbf{x}_{k\eta}), \ \hat{b}_t := \mathbf{x}_t + 2\sigma^2 \nabla \ln p_{T-t}(\mathbf{x}_t), \sigma_t = \sqrt{2}\sigma, \text{ and } t \in [k\eta, (k+1)\eta],
$$

*into Lemma 12 and assuming Novikov's condition holds, then we have*

$$
D_{\mathrm{KL}}\left(\hat{P}_T \| \tilde{P}_T^{p_T}\right) = \mathbb{E}_{\hat{P}_T}\left[\ln \frac{\mathrm{d}\hat{P}_T}{\mathrm{d}\tilde{P}_T^{p_T}}\right] = \frac{1}{4}\sum_{k=0}^{N-1} \mathbb{E}_{\hat{P}_T}\left[\int_{k\eta}^{(k+1)\eta} \left\|\mathbf{v}_k(\mathbf{x}_{k\eta}) - 2\nabla \ln p_{T-t}(\mathbf{x}_t)\right\|^2 \mathrm{d}t\right].
$$

**Lemma 13.** *(Lemma C.11 in Lee et al. (2022)) Suppose that $p(\boldsymbol{x}) \propto e^{-f(\boldsymbol{x})}$ is a probability density function on $\mathbb{R}^d$, where $f(\boldsymbol{x})$ is $L$-smooth, and let $\varphi_{\sigma^2}(\boldsymbol{x})$ be the density function of $\mathcal{N}(\mathbf{0}, \sigma^2 \boldsymbol{I}_d)$. Then for $L \leq \frac{1}{2\sigma^2}$, it has*

$$
\left\|\nabla \ln \frac{p(\boldsymbol{x})}{(p * \varphi_{\sigma^2})(\boldsymbol{x})}\right\| \leq 6L\sigma d^{1/2} + 2L\sigma^2 \|\nabla f(\boldsymbol{x})\|.
$$

**Lemma 14.** *(Lemma 9 in Chen et al. (2022a)) Suppose that Assumption [$A_1$] and [$A_2$] hold. Let $(\mathbf{x})_{t \in [0,T]}$ denote the forward process 1.*

   *1. (moment bound) For all $t \geq 0$,*

$$
\mathbb{E}\left[\|\mathbf{x}_t\|^2\right] \leq d \vee m_2^2.
$$

2. *(score function bound) For all $t \geq 0$,*

$$\mathbb{E}\left[\|\nabla \ln p_t(\mathbf{x}_t)\|^2\right] \leq Ld.$$

**Lemma 15.** *(Variant of Lemma 10 in Chen et al. (2022a)) Suppose that Assumption [$A_2$] holds. Let $(\mathbf{x})_{t \in [0,T]}$ denote the forward process 1. For $0 \leq s < t$, if $t - s \leq 1$, then*

$$\mathbb{E}\left[\|\mathbf{x}_t - \mathbf{x}_s\|^2\right] \leq 2\left(m_2^2 + d\right) \cdot (t-s)^2 + 4d \cdot (t-s)$$

*Proof.* According to the forward process, we have

$$\mathbb{E}\left[\|\mathbf{x}_t - \mathbf{x}_s\|^2\right] = \mathbb{E}\left[\left\|\int_s^t -\mathbf{x}_r dr + \sqrt{2}\left(B_t - B_s\right)\right\|^2\right] \leq \mathbb{E}\left[2\left\|\int_s^t \mathbf{x}_r dr\right\|^2 + 4\|B_t - B_s\|^2\right]$$

$$\leq 2\mathbb{E}\left[\left(\int_s^t \|\mathbf{x}_r\| dr\right)^2\right] + 4d \cdot (t-s) \leq 2\int_s^t \mathbb{E}\left[\|\mathbf{x}_r\|^2\right] dr \cdot (t-s) + 4d \cdot (t-s)$$

$$\leq 2\left(m_2^2 + d\right) \cdot (t-s)^2 + 4d \cdot (t-s),$$

where the third inequality follows from Holder's inequality and the last one follows from Lemma 14. Hence, the proof is completed. □

**Lemma 16.** *(Corollary 3.1 in Chafaï (2004)) If $\nu, \tilde{\nu}$ satisfy LSI with constants $\alpha, \tilde{\alpha} > 0$, respectively, then $\nu * \tilde{\nu}$ satisfies LSI with constant $\left(\frac{1}{\alpha} + \frac{1}{\tilde{\alpha}}\right)^{-1}$.*

**Lemma 17.** *Let $\mathbf{x}$ be a real random variable. If there exist constants $C, A < \infty$ such that $\mathbb{E}\left[e^{\lambda \mathbf{x}}\right] \leq Ce^{A\lambda^2}$ for all $\lambda > 0$ then*

$$\mathbb{P}\{\mathbf{x} \geq t\} \leq C\exp\left(-\frac{t^2}{4A}\right)$$

*Proof.* According to the non-decreasing property of exponential function $e^{\lambda x}$, we have

$$\mathbb{P}\{\mathbf{x} \geq t\} = \mathbb{P}\left\{e^{\lambda \mathbf{x}} \geq e^{\lambda t}\right\} \leq \frac{\mathbb{E}\left[e^{\lambda \mathbf{x}}\right]}{e^{\lambda t}} \leq C\exp\left(A\lambda^2 - \lambda t\right),$$

The first inequality follows from Markov inequality and the second follows from the given conditions. By minimizing the RHS, i.e., choosing $\lambda = t/(2A)$, the proof is completed. □

**Lemma 18.** *If $\nu$ satisfies a log-Sobolev inequality with log-Sobolev constant $\mu$ then every 1-Lipschitz function $f$ is integrable with respect to $\nu$ and satisfies the concentration inequality*

$$\nu\{f \geq \mathbb{E}_\nu[f] + t\} \leq \exp\left(-\frac{\mu t^2}{2}\right).$$

*Proof.* According to Lemma 17, it suffices to prove that for any 1-Lipschitz function $f$ with expectation $\mathbb{E}_\nu[f] = 0$,

$$\mathbb{E}\left[e^{\lambda f}\right] \leq e^{\lambda^2/(2\mu)}.$$

To prove this, it suffices, by a routine truncation and smoothing argument, to prove it for bounded, smooth, compactly supported functions $f$ such that $\|\nabla f\| \leq 1$. Assume that $f$ is such a function. Then for every $\lambda \geq 0$ the log-Sobolev inequality implies

$$\text{Ent}_\nu\left(e^{\lambda f}\right) \leq \frac{2}{\mu}\mathbb{E}_\nu\left[\left\|\nabla e^{\lambda f/2}\right\|^2\right],$$

which is written as

$$\mathbb{E}_\nu\left[\lambda f e^{\lambda f}\right] - \mathbb{E}_\nu\left[e^{\lambda f}\right]\log \mathbb{E}\left[e^{\lambda f}\right] \leq \frac{\lambda^2}{2\mu}\mathbb{E}_\nu\left[\|\nabla f\|^2 e^{\lambda f}\right].$$

With the notation $\varphi(\lambda) = \mathbb{E}\left[e^{\lambda f}\right]$ and $\psi(\lambda) = \log \varphi(\lambda)$, the above inequality can be reformulated as

$$\lambda \varphi'(\lambda) \leq \varphi(\lambda) \log \varphi(\lambda) + \frac{\lambda^2}{2\mu} \mathbb{E}_\nu \left[\|\nabla f\|^2 e^{\lambda f}\right]$$

$$\leq \varphi(\lambda) \log \varphi(\lambda) + \frac{\lambda^2}{2\mu} \varphi(\lambda),$$

where the last step follows from the fact $\|\nabla f\| \leq 1$. Dividing both sides by $\lambda^2 \varphi(\lambda)$ gives

$$\left(\frac{\log(\varphi(\lambda))}{\lambda}\right)' \leq \frac{1}{2\mu}.$$

Denoting that the limiting value $\frac{\log(\varphi(\lambda))}{\lambda}\big|_{\lambda=0} = \lim_{\lambda \to 0^+} \frac{\log(\varphi(\lambda))}{\lambda} = \mathbb{E}_\nu[f] = 0$, we have

$$\frac{\log(\varphi(\lambda))}{\lambda} = \int_0^\lambda \left(\frac{\log(\varphi(t))}{t}\right)' dt \leq \frac{\lambda}{2\mu},$$

which implies that

$$\psi(\lambda) \leq \frac{\lambda^2}{2\mu} \implies \varphi(\lambda) \leq \exp\left(\frac{\lambda^2}{2\mu}\right)$$

Then the proof can be completed by a trivial argument of Lemma 17. $\qquad \square$

**Lemma 19.** *(Theorem 1 in Vempala & Wibisono (2019)) Suppose $p_*$ satisfies LSI with constant $\mu > 0$ and is L-smooth. For any $\mathbf{x}_0 \sim p_0$ with $D_{\mathrm{KL}}(p_0\|p_\infty) < \infty$, the iterates $\mathbf{x}_k \sim p_k$ of ULA with step size $0 < \tau \leq \frac{\mu}{4L^2}$ satisfy*

$$D_{\mathrm{KL}}(p_t\|p_\infty) \leq e^{-\mu\tau k} D_{\mathrm{KL}}(p_0\|p_\infty) + \frac{8\tau dL^2}{\mu}.$$

*Thus, for any $\delta > 0$, to achieve $D_{\mathrm{KL}}(p_t\|p_\infty) \leq \delta$, it suffices to run ULA with step size*

$$0 < \tau \leq \frac{\mu}{4L^2} \min\left\{1, \frac{\delta}{4d}\right\}$$

*for*

$$k \geq \frac{1}{\mu\tau} \log \frac{2D_{\mathrm{KL}}(p_0\|p_\infty)}{\delta}.$$

**Lemma 20.** *(Variant of Lemma 10 in Cheng & Bartlett (2018)) Suppose $-\ln p_\infty$ is m-strongly convex function, for any distribution with density function $p$, we have*

$$D_{\mathrm{KL}}(p\|p_\infty) \leq \frac{1}{2m} \int p(\boldsymbol{x}) \left\|\nabla \ln \frac{p(\boldsymbol{x})}{p_*(\boldsymbol{x})}\right\|^2 d\boldsymbol{x}.$$

*By choosing $p(\boldsymbol{x}) = g^2(\boldsymbol{x})p_*(\boldsymbol{x})/\mathbb{E}_{p_*}\left[g^2(\mathbf{x})\right]$ for the test function $g \colon \mathbb{R}^d \to \mathbb{R}$ and $\mathbb{E}_{p_*}\left[g^2(\mathbf{x})\right] < \infty$, we have*

$$\mathbb{E}_{p_*}\left[g^2 \ln g^2\right] - \mathbb{E}_{p_*}\left[g^2\right] \ln \mathbb{E}_{p_*}\left[g^2\right] \leq \frac{2}{m} \mathbb{E}_{p_*}\left[\|\nabla g\|^2\right],$$

*which implies $p_*$ satisfies m-log-Sobolev inequality.*

**Lemma 21.** *Using the notation in Table. 1, for each $t \in [0, \infty)$, the underlying distribution $p_t$ of the forward process satisfies*

$$D_{\mathrm{KL}}(p_t\|p_\infty) \leq 4(dL + m_2^2) \cdot \exp\left(-\frac{t}{2}\right)$$

*Proof.* Consider the Fokker–Planck equation of the forward process, i.e.,

$$d\mathbf{x}_t = -\mathbf{x}_t dt + \sqrt{2} dB_t, \quad \mathbf{x}_0 \sim p_0 \propto e^{-f_*},$$

we have

$$\partial_t p_t(\boldsymbol{x}) = \nabla \cdot (p_t(\boldsymbol{x})\boldsymbol{x}) + \Delta p_t(\boldsymbol{x}) = \nabla \cdot \left(p_t(\boldsymbol{x}) \nabla \ln \frac{p_t(\boldsymbol{x})}{e^{-\frac{1}{2}\|\boldsymbol{x}\|^2}}\right).$$

It implies that the stationary distribution is

$$p_\infty \propto \exp\left(-\frac{1}{2}\cdot\|\boldsymbol{x}\|^2\right).\tag{55}$$

Then, we consider the KL convergence of $(\mathbf{x}_t)_{t\geq 0}$, and have

$$
\begin{aligned}
\frac{\mathrm{d}D_{\mathrm{KL}}(p_t\|p_\infty)}{\mathrm{d}t} =& \frac{\mathrm{d}}{\mathrm{d}t}\int p_t(\boldsymbol{x})\ln\frac{p_t(\boldsymbol{x})}{p_\infty(\boldsymbol{x})}\mathrm{d}\boldsymbol{x} = \int \partial_t p_t(\boldsymbol{x})\ln\frac{p_t(\boldsymbol{x})}{p_\infty(\boldsymbol{x})}\mathrm{d}\boldsymbol{x} \\
=& \int \nabla\cdot\left(p_t(\boldsymbol{x})\nabla\ln\frac{p_t(\boldsymbol{x})}{p_\infty(\boldsymbol{x})}\right)\cdot\ln\frac{p_t(\boldsymbol{x})}{p_\infty(\boldsymbol{x})}\mathrm{d}\boldsymbol{x} \\
=& -\int p_t(\boldsymbol{x})\left\|\nabla\ln\frac{p_t(\boldsymbol{x})}{p_\infty(\boldsymbol{x})}\right\|^2\mathrm{d}\boldsymbol{x}.
\end{aligned}\tag{56}
$$

According to Proposition 5.5.1 of Bakry et al. (2014), if $p_\infty$ is a centered Gaussian measure on $\mathbb{R}^d$ with covariance matrix $\Sigma$, for every smooth function $f$ on $\mathbb{R}^d$, we have

$$\mathbb{E}_{p_\infty}\left[f^2\log f^2\right] - \mathbb{E}_{p_\infty}\left[f^2\right]\log\mathbb{E}_{p_\infty}\left[f^2\right] \leq 2\mathbb{E}_{p_\infty}\left[\Sigma\nabla f\cdot\nabla f\right]$$

For the forward stationary distribution Eq. 55, we have $\Sigma = \boldsymbol{I}$. Hence, by choosing $f^2(\boldsymbol{x}) = p_t(\boldsymbol{x})/p_\infty(\boldsymbol{x})$, we have

$$D_{\mathrm{KL}}\left(p_t\|p_\infty\right) \leq 2\int p_t(\boldsymbol{x})\left\|\nabla\ln\frac{p_t(\boldsymbol{x})}{p_\infty(\boldsymbol{x})}\right\|^2\mathrm{d}\boldsymbol{x}$$

Plugging this inequality into Eq. 56, we have

$$\frac{\mathrm{d}D_{\mathrm{KL}}(p_t\|p_\infty)}{\mathrm{d}t} = -\int p_t(\boldsymbol{x})\left\|\nabla\ln\frac{p_t(\boldsymbol{x})}{p_\infty(\boldsymbol{x})}\right\|^2\mathrm{d}\boldsymbol{x} \leq -\frac{1}{2}D_{\mathrm{KL}}(p_t\|p_\infty).$$

Integrating implies the desired bound,i.e.,

$$D_{\mathrm{KL}}(p_t\|p_\infty) \leq \exp\left(-\frac{t}{2}\right)D_{\mathrm{KL}}(p_0\|p_\infty) = C_0\exp\left(-\frac{t}{2}\right).$$

$\square$

**Lemma 22.** *Suppose $f_1\colon\mathbb{R}^d\to\mathbb{R}$ and $f_2\colon\mathbb{R}^d\to\mathbb{R}$ is $\mu$-strongly convex for $\|\boldsymbol{x}\|\geq R$. That means $v_1(\boldsymbol{x}) := f_1(\boldsymbol{x}) - \mu/2\cdot\|\boldsymbol{x}\|^2$ (and $v_2(\boldsymbol{x}) := f_2(\boldsymbol{x}) - \mu/2\cdot\|\boldsymbol{x}\|^2$) is convex on $\Omega = \mathbb{R}^d\setminus B(\boldsymbol{0},R)$. Specifically, we require that $\boldsymbol{x}\in\Omega$, any convex combination of $\boldsymbol{x} = \sum_{i=1}^k\lambda_i\boldsymbol{x}_i$ with $\boldsymbol{x}_1,\ldots,\boldsymbol{x}_k\in\Omega$ satisfies*

$$v_1(\boldsymbol{x}) \leq \sum_{i=1}^k\lambda_k v_1(\boldsymbol{x}_i).$$

*Then, we have $f_1 + f_2$ is $2\mu$-strongly convex for $\|\boldsymbol{x}\|\geq R$.*

*Proof.* We define $v(\boldsymbol{x}) = f(\boldsymbol{x}) - \mu\|\boldsymbol{x}\|^2$. Hence, by considering $\boldsymbol{x}\in\Omega$ and its convex combination of $\boldsymbol{x} = \sum_{i=1}^k\lambda_i\boldsymbol{x}_i$ with $\boldsymbol{x}_1,\ldots,\boldsymbol{x}_k\in\Omega$, we have

$$v(\boldsymbol{x}) = v_1(\boldsymbol{x}) + v_2(\boldsymbol{x}) \leq \sum_{i=1}^k\lambda_i v_1(\boldsymbol{x}_i) + \sum_{i=1}^k\lambda_i v_2(\boldsymbol{x}_i) = \sum_{i=1}^k\lambda_i v(\boldsymbol{x}_i).$$

Hence, the proof is completed. $\square$

**Lemma 23.** *Suppose $q$ is a distribution which satisfies LSI with constant $\mu$, then its variance satisfies*

$$\int q(\boldsymbol{x})\|\boldsymbol{x} - \mathbb{E}_{\tilde{q}}[\mathbf{x}]\|^2\mathrm{d}\boldsymbol{x} \leq \frac{d}{\mu}.$$

*Proof.* It is known that LSI implies Poincaré inequality with the same constant (Rothaus, 1981; Villani, 2021; Vempala & Wibisono, 2019), which can be derived by taking $\rho \to (1 + \eta g)\nu$ in Eq. (10). Thus, for $\mu$-LSI distribution $q$, we have

$$\mathrm{var}_q\left(g(\mathbf{x})\right) \leq \frac{1}{\mu}\mathbb{E}_q\left[\|\nabla g(\mathbf{x})\|^2\right].$$

for all smooth function $g\colon \mathbb{R}^d \to \mathbb{R}$.

In this condition, we suppose $\boldsymbol{b} = \mathbb{E}_q[\mathbf{x}]$, and have the following equation

$$\int q(\boldsymbol{x})\left\|\boldsymbol{x} - \mathbb{E}_q\left[\mathbf{x}\right]\right\|^2 \mathrm{d}\boldsymbol{x} = \int q(\boldsymbol{x})\left\|\boldsymbol{x} - \boldsymbol{b}\right\|^2 \mathrm{d}\boldsymbol{x}$$

$$= \int \sum_{i=1}^{d} q(\boldsymbol{x})\left(\boldsymbol{x}_i - \boldsymbol{b}_i\right)^2 \mathrm{d}\boldsymbol{x} = \sum_{i=1}^{d} \int q(\boldsymbol{x})\left(\langle\boldsymbol{x}, \boldsymbol{e}_i\rangle - \langle\boldsymbol{b}, \boldsymbol{e}_i\rangle\right)^2 \mathrm{d}\boldsymbol{x}$$

$$= \sum_{i=1}^{d} \int q(\boldsymbol{x})\left(\langle\boldsymbol{x}, \boldsymbol{e}_i\rangle - \mathbb{E}_q\left[\langle\mathbf{x}, \boldsymbol{e}_i\rangle\right]\right)^2 \mathrm{d}\boldsymbol{x} = \sum_{i=1}^{d} \mathrm{var}_q\left(g_i(\mathbf{x})\right)$$

where $g_i(\boldsymbol{x})$ is defined as $g_i(\boldsymbol{x}) := \langle\boldsymbol{x}, \boldsymbol{e}_i\rangle$ and $\boldsymbol{e}_i$ is a one-hot vector ( the $i$-th element of $\boldsymbol{e}_i$ is 1 others are 0).

Combining this equation and Poincaré inequality, for each $i$, we have

$$\mathrm{var}_q\left(g_i(\mathbf{x})\right) \leq \frac{1}{\mu}\mathbb{E}_q\left[\|\boldsymbol{e}_i\|^2\right] = \frac{1}{\mu}.$$

By combining the equation and inequality above, we have

$$\int q(\boldsymbol{x})\left\|\boldsymbol{x} - \mathbb{E}_q\left[\mathbf{x}\right]\right\|^2 \mathrm{d}\boldsymbol{x} = \sum_{i=1}^{d} \mathrm{var}_q\left(g_i(\mathbf{x})\right) \leq \frac{d}{\mu}$$

Hence, the proof is completed. $\qquad\square$

# F  EMPIRICAL RESULTS

## F.1  EXPERIMENT SETTINGS AND MORE EMPIRICAL RESULTS

We choose $1,000$ particles in the experiments and use MMD (with RBF kernel) as the metric. We choose $T \in \{-\ln 0.99, -\ln 0.95, -\ln 0.9, -\ln 0.8, -\ln 0.7\}$. We use 10, 50, or 100 iterations to approximate $\hat{p}$ chosen by the corresponding problem. The inner loop is initialized with importance sampling mean estimator by 100 particles. The inner iteration and inner loop sample-size are chosen from $\{1, 5, 10, 100\}$. The outer learning rate is chosen from $\{T/20, T/10, T/5\}$. When the algorithm converges, we further perform LMC until the limit of gradient complexity. Note that the gradient complexity is evaluated by the product of outer loop and inner loop.

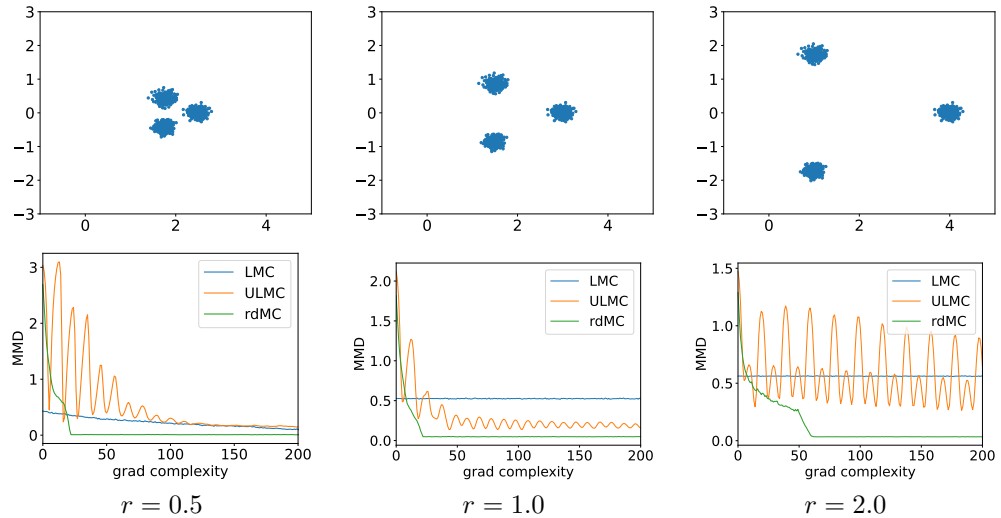

Figure 4: Maximum Mean Discrepancy (MMD) convergence of LMC, ULMC, RDMC.

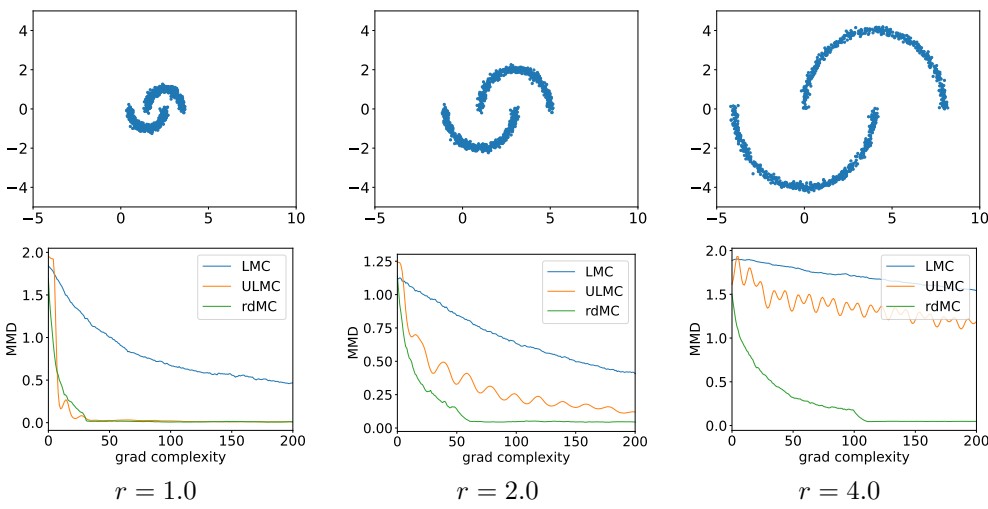

Figure 5: Maximum Mean Discrepancy (MMD) convergence of LMC, ULMC, RDMC.

## F.2  MORE INVESTIGATION ON ILL-BEHAVED GAUSSIAN CASE

Figure 6 demonstrate the differences between Langevin dynamics and the OU process in terms of their trajectories. The former utilizes the gradient information of the target distribution $\nabla \ln p_*$,

to facilitate optimization. However, it converges slowly in directions with small gradients. On the other hand, the OU process constructs paths with equal velocities in each direction, thereby avoiding the influence of gradient vanishing directions. Consequently, leveraging the reverse process of the OU process is advantageous for addressing the issue of uneven gradient sampling across different directions.

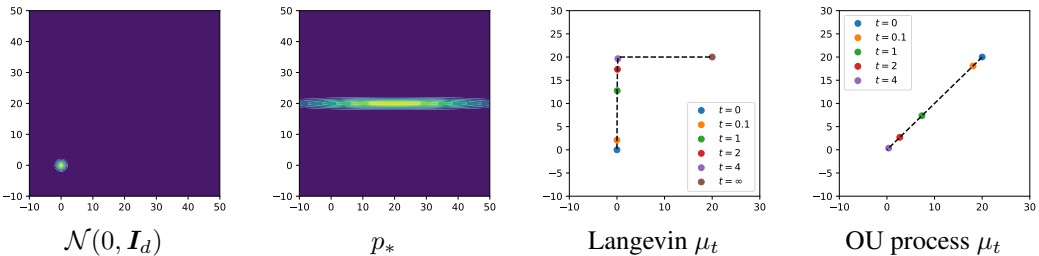

$$\mathcal{N}(0, \boldsymbol{I}_d) \qquad p_* \qquad \text{Langevin } \mu_t \qquad \text{OU process } \mu_t$$

Figure 6: Langevin dynamics vs (reverse) OU process. We consider the sampling path between the 2-dimensional normal distributions $\mathcal{N}(0, \boldsymbol{I}_2)$ and $\mathcal{N}((20, 20), \mathrm{diag}(400, 1))$. The mean $\mu_t$ of Langevin dynamics show varying convergence speeds in different directions, while the OU process demonstrates more uniform changes.

In order to further demonstrate the effectiveness of our algorithm, we conducted additional experiments comparing the Langevin dynamics with our proposed method in our sample scenarios. To better highlight the impacts of different components, we chose the 2-dimensional ill-conditioned Gaussian distribution $\mathcal{N}\left((20, 20), \begin{pmatrix} 400 & 0 \\ 0 & 1 \end{pmatrix}\right)$ (shown in Figure 7) as the target distribution to showcase this aspect. In this setting, we obtained an oracle sampler for $q_t$, enabling us to conduct a more precise experimental analysis of the sample complexity.

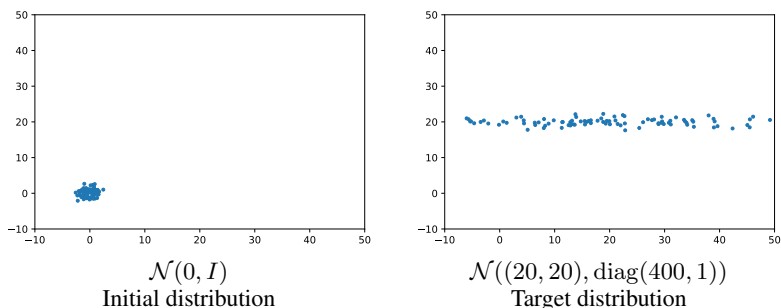

$$\mathcal{N}(0, I) \qquad\qquad \mathcal{N}((20, 20), \mathrm{diag}(400, 1))$$
Initial distribution $\qquad\qquad$ Target distribution

Figure 7: Initial and Target distribution sample illustration.

Figure 8 illustrates the distributions of samples generated by the LMC algorithm at iterations 10, 100, 1000, and 10000. It can be observed that these distributions deviate from the target distribution.

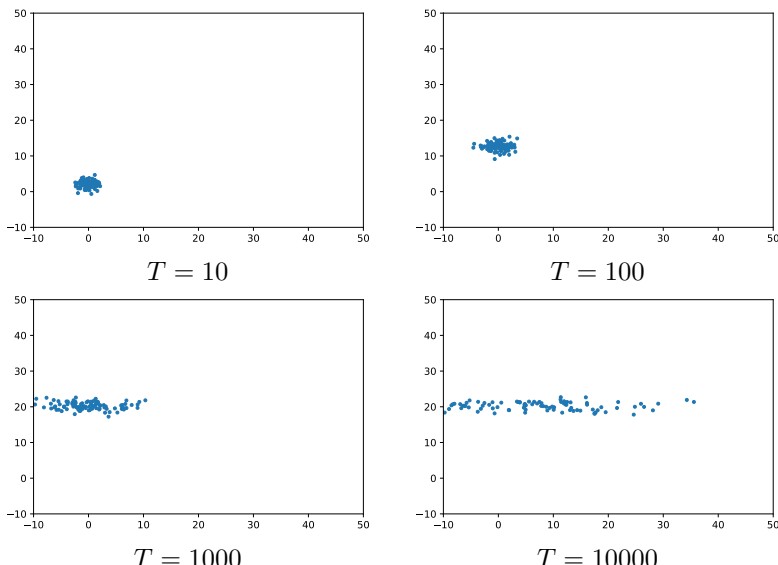

Figure 8: Illustration of LMC with different iterations.

Figure 9 demonstrates the performance of the RDMC algorithm in the scenario of infinite samples, revealing its significantly superior convergence compared to the LMC algorithm.

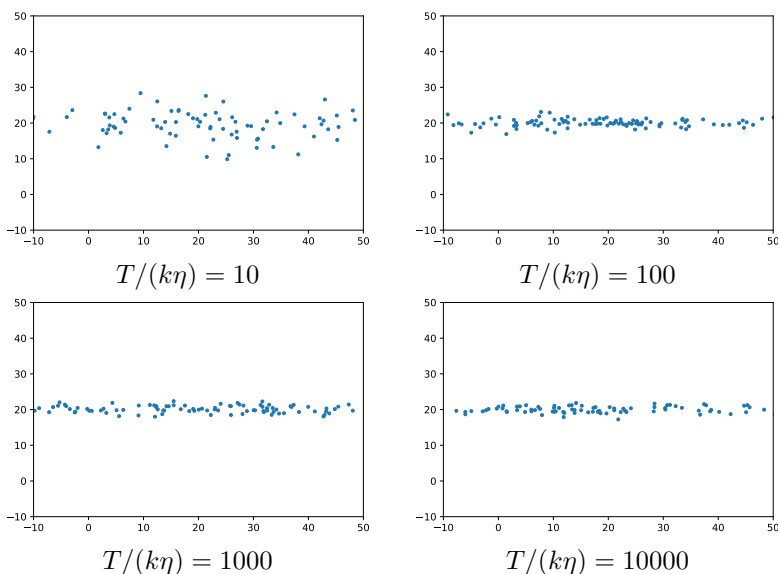

Figure 9: Illustration of RDMC (oracle sampler, infinite samples) with different iterations.

Figure 10 showcases the convergence of the RDMC algorithm in estimating the score under different sample sizes. It can be observed that our algorithm is not sensitive to the sample quantity, demonstrating its robustness in practical applications.

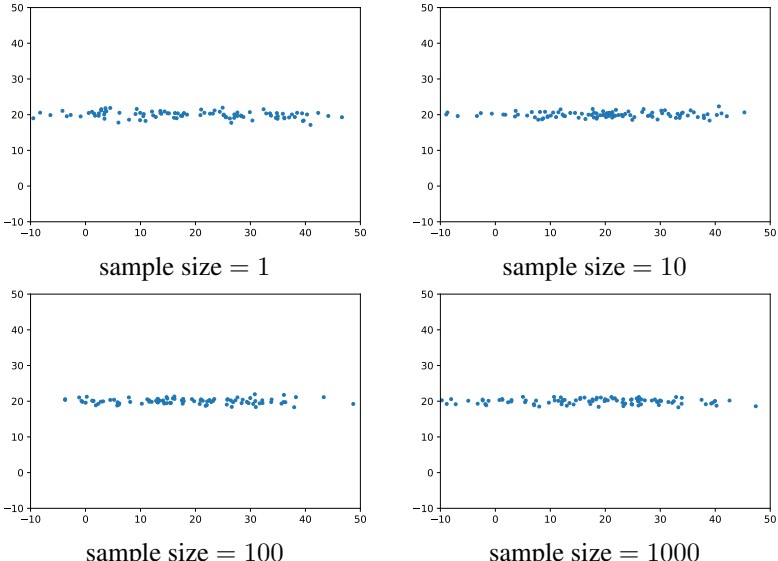

Figure 10: Illustration of RDMC (oracle sampler, finite samples, $\frac{T}{k\eta} = 100$).

Figure 11 illustrates the convergence behavior of the RDMC algorithm with an inexact solver. It can be observed that even when employing LMC as the solver for the inner loop, the final convergence of our algorithm surpasses that of the original LMC. This is attributed to the insensitivity of the algorithm to the precision of the inner loop when $t$ is large. Additionally, when $t$ is small, the log-Sobolev constant of the inner problem is relatively large, simplifying the problem as a whole and guiding the samples towards the target distribution through the diffusion path.

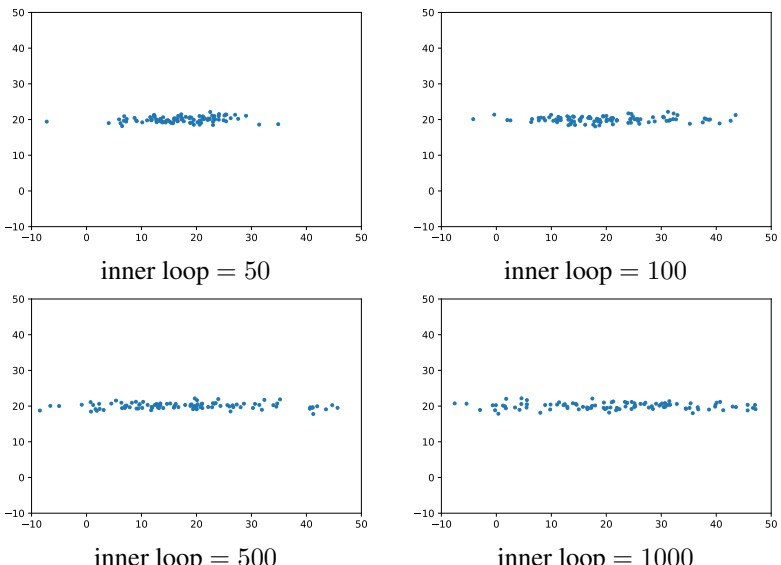

Figure 11: Illustration of RDMC (inexact sampler, finite samples, $\frac{T}{k\eta} = 100$).

### F.3 SAMPLING FROM HIGH-DIMENSIONAL DISTRIBUTIONS

To validate the dimensional dependency of RDMC algorithm, we consider to extend our Gaussian mixture model experiments.

In the log-Sobolev distribution context (see Section 4.2.1), both the Langevin-based algorithm and our proposed method exhibit polynomial dependence on dimensionality. However, the log-Sobolev

constant grows exponentially with radius $R$, which is the primary influencing factor. This leads to behavior akin to the 2-D example presented earlier. A significant limitation of the Langevin-based algorithm is its inability to converge within finite time for large $R$ values, in contrast to the robustness of our algorithm. In higher-dimensional scenarios (e.g., $r = 2$ for $d = 50$ and $d = 100$), we observe a notable decrease in rdMC performance after approximately 100 computations. This decline may stem from the kernel-based computation of MMD, which tends to be less sensitive in higher dimensions. For these large $r$ cases, LMC and ULMC fails to converge in finite time. Overall, Figure 12 exhibits trends consistent with Figure 3, corroborating our theoretical findings.

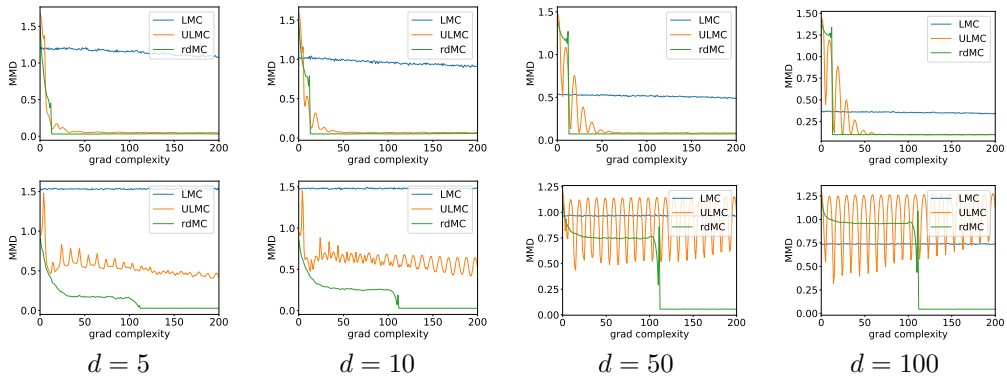

Figure 12: Illustration of RDMC and MCMC for high-dimensional Gaussian Mixture model (6 modes). The first row shows the case of $r = 1$. The second row shows the case of $r = 2$.

For heavy-tail distribution (refer to Section 4.2.2), the complexity of both RDMC and MCMC are quite high, so it is not feasible to sample from them in limited time. Nonetheless, as our algorithm only has the polynomial dependency of $d$, but the Langevin-based algorithms have exponential dependency with respect to $d$, we may have more advantages. For example, we can use the $n_{\text{th}}$ moment demonstrate the similarity of different distributions. To illustrate the phenomenon, we consider the extreme case – Cauchy distribution[2]. According to Figure 13, RDMC has better approximation for the true distribution and the approximation gap increases with respect to dimension.

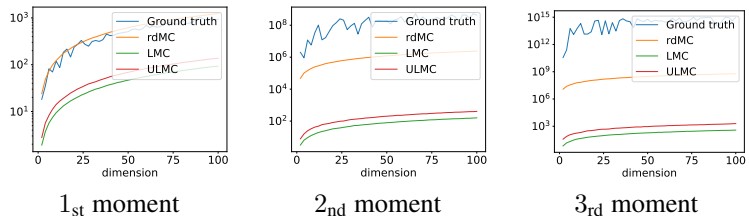

Figure 13: Moment estimators for Cauchy distribution when $n = 1000$. It reflects the closeness between different distributions. For high dimensional moments, we computes the sum across all dimensions (trace) for proper plotting. The algorithm are evaluated with 1K gradient complexity.

### F.4 Discussion on the choice of $\tilde{p}_0$

Note that different $\tilde{p}_0$ may have impacts on the algorithm. In this subsection, we discuss the impact of choice of $\tilde{p}_0$ and try to make a clear demonstration for the influence in real practice.

Practically, selecting $T$ determines the initial distribution for reverse diffusion. While any $T$ choice can achieve convergence, the computational complexity varies with different $T$ values. According to Figure 14, we can notice that with the increase of $T$, the modes of $p_T$ tend to merge to a single mode, and the speed is exponentially fast (Lemma 21). Even with $T = -\ln 0.95$, the dis-connectivity of modes can be alleviated significantly. Thus, the choice of $T$ is not sensitive when choosing $T$ from $-\ln 0.95$ to $-\ln 0.7$.

---

[2]The mean of Cauchy distribution does not exist. We use the case for better heavy-tail demonstration.

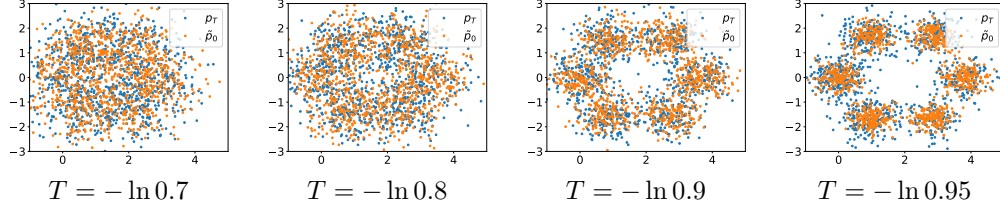

| $T = -\ln 0.7$ | $T = -\ln 0.8$ | $T = -\ln 0.9$ | $T = -\ln 0.95$ |

Figure 14: Choice of $p_T$ and the approximation by $\tilde{p}_0$. For $T$ from $-\ln 0.95$ to $-\ln 0.7$, $\tilde{p}_0$ can approximate $p_T$ properly, where each modes are connected to make the approximated distribution well-mixed. Then the RDMC can be performed properly.

However, when choosing too small or too large $T$, there would be some consequence:

- If $T$ is too small, the approximation of $\tilde{p}_0$ would be extremely hard. For example, if $T \to 0$, the algorithm would be similar to Langevin algorithm;

- If $T$ is too large, it would be wasteful to transport from $T$ to $-\ln 0.7$ since the distribution in this interval is highly homogeneous[3].

In summary, aside from the $T \to 0$ scenario, our algorithm exhibits insensitivity to the choice of $T$ (or $\tilde{p}_0$). Selecting an appropriate $T$ can reduce computational demands when using a constant step size schedule.

## F.5 NEAL'S FUNNEL

Neal's Funnel is a classic demonstration of how traditional MCMC methods struggle with convergence unless specific parameterization strategies are employed. We further investigate the performance of our algorithm for this scenario. As indicated in Figure 15, our method demonstrates more rapid convergence compared to Langevin-based MCMC approaches. Additionally, Figure 16 reveals that while LMC lacks efficient exploration capabilities, ULMC fails to accurately represent density. This discrepancy stems from incorporating momentum/velocity. Our algorithm strikes an improved balance between exploration and precision, primarily due to the efficacy of the reverse diffusion path, thereby enhancing convergence speed.

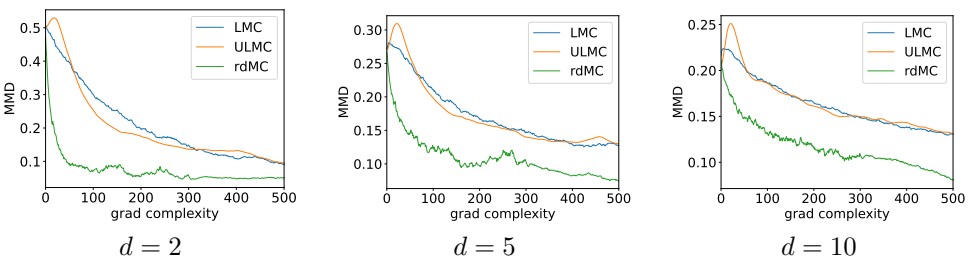

| $d = 2$ | $d = 5$ | $d = 10$ |

Figure 15: Convergence of Neal's Funnel for RDMC, LMC, and ULMC.

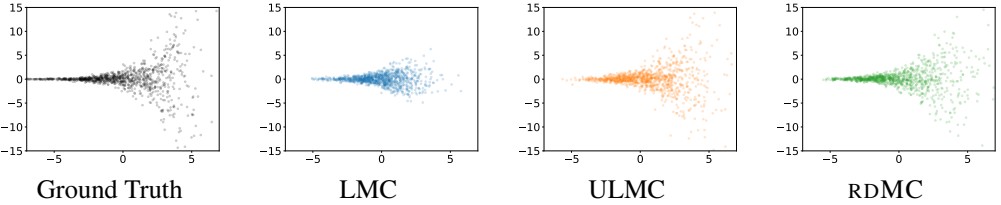

| Ground Truth | LMC | ULMC | RDMC |

Figure 16: Samples from Neal's Funnel.

---

[3]It is possible to consider varied step size scheduling, which can be interesting future work.

