# OpenReview forum: "Reverse Diffusion Monte Carlo"
_ICLR.cc/2024/Conference — ICLR 2024 poster_

### Official Review · Reviewer_NhM6 · 2023-10-27

**Soundness:** 3 good
**Presentation:** 3 good
**Contribution:** 3 good
**Rating:** 8
**Confidence:** 3

**Summary:**

This work takes a very well-known reformulation of the score in VP-SDEs (see [1,2,3]) that is admissible to the sampling setting where one only has access to the density of the target distribution up to a constant unlike in standard diffusion models where one has access to samples. The authors propose estimating the score via MC, 100% akin to the heat-semigroup (Schroedinger Foellmer Sampler) approach in [4], which rather than time reversing a VP-SDE it can be seen as time reversing the h-transform of a pinned Brownian motion [9].

There is quite a bit of missing literature that has been already explored empirically over the last 2 years in the works of [1,2,3,4]  and also [5, 6, 7, 8].

The ULA-based estimators proposed in this work are the main novelty focus, furthermore, complexity guarantees are provided for these estimators and are shown to outperform vanilla ULA approaches, which is quite promising combined with the experiments added during the rebuttal.

Notice that in contrast to parametric VI approaches the inner loop scheme proposed has theoretical guarantees for estimating the score that is practically feasible unlike [4,8] which is hindered nonpractical due to the. nonconvex objective required to train the NN estimators of the score.

[1] Vargas, F., Grathwohl, W.S. and Doucet, A., 2022, September. Denoising Diffusion Samplers. In The Eleventh International Conference on Learning Representations. https://openreview.net/forum?id=8pvnfTAbu1f

[2] Berner, J., Richter, L. and Ullrich, K., 2022. An optimal control perspective on diffusion-based generative modeling. In NeurIPS 2022 Workshop on Score-Based Methods.

[3] Zhang, D., Chen, R.T.Q., Liu, C.H., Courville, A. and Bengio, Y., 2023. Diffusion Generative Flow Samplers: Improving learning signals through partial trajectory optimization. arXiv preprint arXiv:2310.02679.

[4] Vargas, F., Reu, T. and Kerekes, A., 2023, July. Expressiveness Remarks for Denoising Diffusion Based Sampling. In Fifth Symposium on Advances in Approximate Bayesian Inference.

[5] Huang, J., Jiao, Y., Kang, L., Liao, X., Liu, J. and Liu, Y., 2021. Schrödinger-Föllmer sampler: sampling without ergodicity. arXiv preprint arXiv:2106.10880.

[6] Vargas, F., Ovsianas, A., Fernandes, D., Girolami, M., Lawrence, N.D. and Nüsken, N., 2023. Bayesian learning via neural Schrödinger–Föllmer flows. Statistics and Computing, 33(1), p.3.

[8]  Tzen, B. and Raginsky, M., 2019, June. Theoretical guarantees for sampling and inference in generative models with latent diffusions. In Conference on Learning Theory (pp. 3084-3114). PMLR.

**Strengths:**

The paper is well-written and formatted, and has the potential to become a theory-oriented paper with some extra work if the contributions are restructured,  and proper acknowledgments are made, this would require significant re-writing and further development and discussion of lemma 3 and proposition 2, and also theoretical comparison to other methods beyond ULA, simply the statement of these 2 results alone is not a strong enough contribution for ICLR.

Note that the ULA-based estimators proposed in this work can be seen as a novel (e.g. Algorithm 2), however, without proper experimentation/numerics, this is still not a complete contribution either, currently, the paper is a lot of floating ideas without a concrete exploration of any particular, in part, this is due to the authors not being aware of the current state of the field. However, having to run a couple of inner ULA iterations every time one has to evaluate the drift is highly nonpractical, and claims made in the paper such as :

"
Via this combination, we are
able to efficiently obtain accurate score estimation by virtue of the ULA algorithm when t is close
to T. When t is close to 0, we are able to quickly obtain rough score estimates via the importance
sampling approach.
"

Are highly unvalidated as 2 dimensional toy examples without proper comparison to other diffusion sampling-based approaches.

**Weaknesses:**

I will break this down into 2 subgroups

1. Lack of Novelty (+ failure to acknowledge prior work)

    *  Lemma 1 is very straight-forward and non-practical as you cannot sample from q, instead in practice the authors do IS and sample from the OU-process' transition kernel which is available analytically, which in turn results in expressing the score as done in [1] ( see equation 84 in [1] and the line that follows it connecting it to the score, or the equation above Equation 24 of the same paper ... or equations 10-13 in [4] ).

2. The paper falls short as a sampling paper empirically

    * Evaluations in 2d simply do not meet the bar for a conference paper. Methodologies such as SMC among many other modern ML variants are able to do quite well in multimodal 2d examples.
    * This MC-IS method for estimating the score will NEVER work well in high dimensions due to variance and thus why works such as [1,2,3,4] which are clearly aware of this formulation (as they either state it in their appendices or use it for subsequent calculation) pursue an optimization alternative to estimating the drift.  See [5] Figure 2 for how poor the performance of these estimators is compared to LMC and alternative NN approaches for learning the score as dimension scales.  The authors briefly allude to this issue and propose combining with an inner ULA loop, however as before the authors missed that previous and more practical approaches (based on score matching) have already been developed to address this same issue, thus without any empirical validation and careful comparison it is very unclear why one would select the proposed approach.

Note prior results establishing the exact same expressions for the OU-drift and similar expressions for score-matching SDEs applied to sampling have been public since late 2021.

**Questions:**

Here are some suggestions:

1. For the MC estimator of the score of the forward SDE please cite [5] as the original work to propose this class of estimators for sampling with SDEs. This is not the score of an OU process it is instead the score of a "pinned Brownian motion" (A Brownian bridge like SDE which starts at the target distribution and maps it to a point mass). The authors might be tempted to argue that this is not the score however note that this quantity  (-logarithmic derivative of the value function) is related to the score by an additive term of $\nabla \ln p_t^{\mathrm{ref}}$ where $p_t^{\mathrm{ref}}$ is the marginal density of the associated reference process (in the case of an OU forward process this is just a linear term), see Remark 4 and its proof in [4].
2. The authors should cite and acknowledge that their Lemma 1 (or exactly equivalent versions thereof)  have already been discussed and derived in the works [1,4] in the context of sampling, and mathematical optimization objectives which aim to learn an estimator for this exact same score the context of sampling have been explored empirically and theoretically in [1,2,3] (also 4 but this is concurrent work).
3. Something that could strengthen the theory contribution is comparing to the Log Sobolev constant of the score in Schrodinger follmer samplers [5] (The score of a pinned brownian motion) and using this to quantify algorithmic design insights for the forward process.
4. Overall these theoretical contributions can have a stronger impact if written with the state of the current field on diffusion-based samplers in mind, and contextualizing how these results apply to practically successful methodologies.
7. Note that since the estimator for the score is a ratio of expectations computed via MC the resulting estimator is itself biased, there is no discussion of this in the paper.

**Update:** in the revised versions authors do a careful literature review of both prior and parametric methods which have already explored score-based time reversals for sampling, furthermore with the updated numerics found in the appendix the proposed schemes offer for a much more compelling story.

To justify the notable increase in score I will list a couple of points:

* The log Sobolev constants derived for the OU process and the overall context of the analysis could be of further use and impact to works that focus on time reversal and gen modeling or time reversal and sampling. So there is a potential for impact beyond the proposed algorithm.
* The assumptions in the work have been refined and the sketches are clearer to read in the revised versions.
* The authors have been responsive and very helpful in clarifying points throughout the discussion (albeit a bit combative ... which is not ideal but to a tolerable level).

---

> ### Author Response · Authors · 2023-11-11
> **Quick clarification**
>
> Thank you for your review.
> For better discussion, we would like to clarify certain misunderstandings before presenting our formal rebuttal. Firstly, we acknowledge the concerns raised regarding the omission of parameterized sampling algorithms from our discussion.
> This was based on the fact that these algorithms primarily fall under the Variational Inference (VI) regime, while our algorithm is an improved version of MCMC.
> Nevertheless, we are grateful for your references and suggestions and will incorporate a discussion of these elements in our updated manuscript.
>
> It is crucial to emphasize that our algorithm is non-parametric, focusing on the theoretical analysis and understanding of both asymptotic and non-asymptotic behaviors of discretized dynamics. Here, our rdMC demonstrates superior performance compared to Langevin-based algorithms, particularly for ill-behaved distributions.
>
> A key aspect of our work is the exploration of log-Sobolev constant dependency of the gradient complexity, where we have found that rdMC outperform conventional methods. This distinction makes a direct comparison with parameterized models infeasible.
>
> Our work does indeed share similarities with parameterized diffusion-based sampling algorithms.
> Some ideas might be embedded in your references, which will be discussed in our next version.
>  However, the main algorithm and concentrations still differ. Once the parametric model is chosen, there is always an asymptotic error. In contrast, our method turns score-matching into a non-parametric mean estimation problem and is asymptotically exact, just like the traditional Monte Carlo methods.
> Our work represents a pioneering effort in examining non-parametric algorithms that leverage diffusion dynamics, backed by robust theoretical guarantees. We believe that our theoretical foundations distinctly support these algorithms.
>
> We acknowledge your proposed works could be viewed as a parametric VI version of rdMC, analogous to the relationship between Black-Box Variational Inference and Langevin Dynamics in the VI-MCMC pair [1]. This perspective offers an intriguing angle, viewing diffusion models as a Parametric Approximation to rdMC. However, it is important to note that it's not entirely appropriate to directly compare a newly established, theoretically guaranteed MCMC work with VI models. Our algorithm can approximate the score with arbitrarily small error.
> Yet, the potential connections between these methodologies present an interesting avenue for discussion.
>
> In light of your comments, we plan to have more discussion about connections between our algorithm and the ones you proposed. Additionally, we will extend our empirical results to more high-dimensional scenarios to further substantiate our claims.
>
> We genuinely appreciate your constructive suggestions and look forward to incorporating these insights into our revised work.
>
>
> [1] Black-Box Variational Inference as Distilled Langevin Dynamics

---

> > ### Comment · Reviewer_NhM6 · 2023-11-11
> > **Thank you for the prompt response**
> >
> > Dear Authors,
> >
> > I am indeed aware that some of these approaches I have refferenced are parametric VI-based methods as opposed to your nonparametric approach.
> >
> > However, that does not warrant completely missing out on acknowledging these methods in particular when many of these papers have already explored/proposed some of the main propositions/remarks in your work.
> >
> > My concern which I feel is being dodged in this review is that estimating the score of a VP-SDE for the task sampling has already been proposed by at least 3 prior works before yours, and this is one of the central remarks/observations you pose as novel, which it is not, whether these methods are VI based or not is not relevant, the point is that they are estimating the same score as your work and they should be acknowledged to be the first to do so. Also, note the work in [2] has many remarks pertaining to the score beyond parametric estimation.
> >
> > What should be acknowledged more carefully is that re-expressing the score/drift of a time reversal for sampling is something that has already been studied quite extensively, you will see expressions in [2] for example that are akin to your IS estimator. Your work is not the first to study estimators for these quantities, you are the first to propose a very specific estimator using ULA.
> >
> > Furthermore note that if you parse through the references I have also provided a non-parametric approach for estimating the score of a Pinned Brownian motion for sampling [1], (heat semi-group rather than OU semi-group as you do). To my knowledge this work from 2021 is the first work to estimate the drifts of time reversals via virtually the same expectations you have in your work only that they take the IS route, the main differentiating factors in your work are the use of a ULA sampler to estimate the drift.
> >
> > Rather than circumventing these and arguing how different they are, I suggest the authors carefully and honestly acknowledge prior work and use it to more precisely quantify their contribution. For example, you make a good point here:
> >
> > > It is crucial to emphasize that our algorithm is non-parametric, focusing on the theoretical analysis and understanding of both asymptotic and non-asymptotic behaviors of discretized dynamics. Here, our rdMC demonstrates superior performance compared to Langevin-based algorithms, particularly for ill-behaved distributions.
> >
> > Thus I am sorry to say I find this clarification to obfuscate the issue at hand rather than clarify. That said I appreciate the time put in your response and look forward to the revised version of this script.
> >
> > [1] Huang, J., Jiao, Y., Kang, L., Liao, X., Liu, J. and Liu, Y., 2021. Schrödinger-Föllmer sampler: sampling without ergodicity. arXiv preprint arXiv:2106.10880.
> > [2] Vargas, F., Reu, T. and Kerekes, A., 2023, July. Expressiveness Remarks for Denoising Diffusion Based Sampling. In Fifth Symposium on Advances in Approximate Bayesian Inference.

---

> ### Author Response · Authors · 2023-11-12
> **Thank you for the prompt response**
>
> Dear reviewer,
>
> Thank you for your prompt response. Let me be straightforward about this. You have missed the main contribution of this paper.
>
> Our method is not an approximate Bayesian inference method, like the ones you have cited. Our method is asymptotically exact, like the traditional MCMC methods. It leverages the reverse diffusion process, but does not really build a diffusion model. That is why we can obtain an overall convergence rate that decreases to zero as the computational complexity increases. The existing works, after choosing the parametric (neural network) model, all suffer from an irremovable error.
>
> For [Huang et al.] you mentioned, the assumptions are much stronger than log-Sobolev assumptions, as they assume the Lipschitz continuity of the drift term, so they are unable to deal with general distributions and make a direct comparison with MCMC. Also, the drift term is estimated by importance sampling, which limits the usage as you mentioned.
>
> We will have a separate paragraph citing these works and discussing the distinction.

---

> ### Comment · Reviewer_NhM6 · 2023-11-12
> **Dismissive language**
>
> Dear Author ,
>
> It is not in your interest to be dismissive towards a reviewer. I am taking the time and effort to engage with you promptly and I am willing to reassess, however I do implore you to be respectful and not dismissive.
>
> Firstly I am 100% aware that your method is asymptocally exact and more akin to mcmc I have not missed this focus and aspect of your contribution. My remarks are not taking this away so please stop trying to demerit my review and address the following individual points I have already made.
>
> 1 . I have citied a paper in my review which you have neither acknowledged nor responded too [1]  this work is also asymptotically exact it is more MCMC alike and it is NOT approximate inference.
> 2. Both MCMC methods and approximate inference methods often target the same task . That is a sampling from unnormalized densities that can be evaluated pointwise .  The point I am raising is the  following
>
> “ There are prior works that write down the exact same expectation formulations of the score to tackle the sampling task.”
>
> Most of these prior works (not [1]) decide to learn an estimator for this score via a variational approach rather than an asymptotically exact approach.  Please carefully read these other papers I agree the VI aspect is different but that is not my point be more thoughtful and read my feedback carefully.
>
>
> In short please carefully address these points and stop dismissing them as non-relevant, this is very poor etiquette I am trying to work with you here and you are making my job much harder, let’s remain professional and on topic, please.
>
> Please be mindful of my time and effort.
>
> [1] Huang, J., Jiao, Y., Kang, L., Liao, X., Liu, J. and Liu, Y., 2021. Schrödinger-Föllmer sampler: sampling without ergodicity. arXiv preprint arXiv:2106.10880.

---

> > ### Author Response · Authors · 2023-11-12
> >
> > Thank you for your feedback. I must express my concern regarding the rating of "1 - strong reject." It seems somewhat dismissive, particularly as the evaluation did not address main contributions of the paper.
> > In response to your comments, we have included a concise comparison with Huang et al. We are expanding our scope to consider a more general distribution class.
> >
> > **Anyway, let’s move forward and stop those emotional evaluations.**
> >
> > We are trying our best to give the credit for prior work and please stay tuned.
> >
> > We are eager to resolve any misunderstandings and collaborate to enhance the quality of our paper. We will diligently work on addressing your points in the coming days and look forward to constructive engagement with you.

---

> ### Comment · Reviewer_NhM6 · 2023-11-12
> **Thank you for your response and mediation**
>
> Dear reviewers,
>
> The score 1 was not intended as dismissive of the work, but mostly to alert the strong concern of lacking evaluations and the very incomplete literature, in the current version of the manuscript. This score will very likely change as the revised version is uploaded, especially since I believe you understand some of the concerns I raised and at least part of them are not too difficult to address.
>
> Notice that the open review platform allows for a score of 1, its part of the accepted scores / formal review process and is not intended to dismiss the nature of the contribution but in this case at least point to the current state of the paper, e.g. its presentation, literature review, numerics, claims, etc.
>
> > Anyway, let’s move forward and stop those emotional evaluations.
>
> Agreed, let's stay focused. In that spirit, I want to revist some of your points in the rebuttal that I possibly missed / misinterpreted:
>
> > 1.   It is crucial to emphasize that our algorithm is non-parametric, focusing on the theoretical analysis and understanding of both asymptotic and non-asymptotic behaviors of discretized dynamics. ...
>
> I think we have gone back and forth on this point a bit already. To summarise my views here:
>
> * I agree the proposed schemes in this work is line with traditional MCMC works, thus comparisons to algorithms of this flavour such as ULA,  SFS [1], and such seem more than reasonable.
> * There is a "diffusion models for sampling" aspect to your algorithm, and whilst prior work is mostly (not entirely e.g. [1]) focused on VI approaches to estimate the score, the overal should be discussed and prior relevant formulations of the score should be credited appropiately.
>
> > 2. A key aspect of our work is the exploration of log-Sobolev constant dependency of the gradient complexity, where we have found that rdMC outperform conventional methods. This distinction makes a direct comparison with parameterized models infeasible.
>
> I agree mostly here and this is something that I missed in my initial review. I do think you are justified in not needing to compare empirically to these approaches, the comparison to ULA  / MCMC schems here is sufficient.
>
> However, as I mentioned before there is a significant overlap in that the discussed parametric approaches are trying to estimate the same score.
>
> > The existing works, after choosing the parametric (neural network) model, all suffer from an irremovable error.
>
> This is true but notice that prior work [2] establishes quite detailed bounds on the error that these parametric estimators of the score can potentially achieve, and it may be interesting (**yet not necessary**) to discuss at a high level how the complexity of these bounds compares to the results on the inner loop you have provided.
>
> Of course, in practice, these bounds are not realistic (whilst yours are) as the networks need to be trained via a non-convex objective that is not guaranteed to reach these errors.
>
> Something I mentioned in my initial review, was that I found your log-Sobolev explorations quite exciting as they might also have something more to say in regards to expressiveness bounds when using NN parametrized estimators [2]. This is not directly relevant and you do not need to address this in the manuscript.
>
> > We believe that our theoretical foundations distinctly support these algorithms.
>
> I do agree with this point, although as mentioned before this is not the first work that is entirely of this flavour (e.g. [1]) an important differentiating factor you offer however is a much more refined inner loop with finer guarantees compared to [1] ([1] simply does IS with no inner-loop and is thus very high variance).
>
> I think the presentation of the work would be improved and made much more clear if this differentiating factor (in particular the complexity of the inner loop scheme) is emphasized as the main contribution / differentiating factor, which also renders the method more practical compared to [1].
>
> This said more high dimensional numerics (and complex dists) are needed to truly verify the method empirically. In particular, such verifications must be done such that ULA and the new proposed algorithm have roughly the same computation budget (e.g. inner_loop_steps * outer_loo_steps  = ula_steps).
>
> P.S. I have updated my review a bit to reflect on some of this discussion.
>
> [1] Huang, J., Jiao, et al Schrödinger-Föllmer sampler: sampling without ergodicity
>
> [2] Vargas F. Et al Expressiveness Remarks for Denoising Diffusion Based Sampling.

---

> > ### Author Response · Authors · 2023-11-15
> >
> > > There is a "diffusion models for sampling" aspect to your algorithm, and whilst prior work is mostly (not entirely e.g. [1]) focused on VI approaches to estimate the score, the overal should be discussed and prior relevant formulations of the score should be credited appropiately.
> >
> >
> > A key aspect of our work is the exploration of log-Sobolev constant dependency of the gradient complexity, where we have found that rdMC outperform conventional methods. This distinction makes a direct comparison with parameterized models infeasible. We have added discussion in Related work and Appendix A4.
> >
> >
> > > This is true but notice that prior work [2] establishes quite detailed bounds on the error that these parametric estimators of the score can potentially achieve, and it may be interesting (yet not necessary) to discuss at a high level how the complexity of these bounds compares to the results on the inner loop you have provided.
> >
> > It is interesting to see the parametric error analysis of diffusion-based sampling algorithms. From a high-level perspective, [2] focus on the expressiveness of neural networks to learn the score function and we focus on the feasiblity and complexity to learn the score function for different time step $t$. Given our algorithm and analysis, if the neural network is well-specified (contains the real score function), the score learning of neural networks can be reformulated as a a statistical problem, where samples are obtained with computation efforts as we mentioned.
> >
> >
> >
> >
> > > I do agree with this point, although as mentioned before this is not the first work that is entirely of this flavour (e.g. [1]) an important differentiating factor you offer however is a much more refined inner loop with finer guarantees compared to [1] ([1] simply does IS with no inner-loop and is thus very high variance). I think the presentation of the work would be improved and made much more clear if this differentiating factor (in particular the complexity of the inner loop scheme) is emphasized as the main contribution / differentiating factor, which also renders the method more practical compared to [1].
> >
> > Thank you for your suggestion. We have modified our contribution part and focus on the differentiating factor. Moreover, we also try to address the differences in our related work part and Appendix.
> >
> >
> >
> > > This said more high dimensional numerics (and complex dists) are needed to truly verify the method empirically. In particular, such verifications must be done such that ULA and the new proposed algorithm have roughly the same computation budget (e.g. inner_loop_steps * outer_loo_steps = ula_steps).
> >
> > Thank you for your comments. We have added the high-dimensional empirical results in appendix F.3, which is similar to the low-dim case. The key reason is that dimension is not the leading factor, but the exponential log-Sobolev constant. So the behavior should be similar. And our computation complexity has already take the product (inner_loop_steps * outer_loo_steps) into account in both theory and experiments.

---

> > > ### Comment · Reviewer_NhM6 · 2023-11-15
> > > **Thank you for your revision**
> > >
> > > with the current numerical ablations, the overall paper and method present a more compelling story I have updated my score to reflect this, for the camera-ready version I would suggest to the authors to (I believe you get an extra page which should allow you for this):
> > >
> > > 1. Move the high-dimensional results to the main rather than the appendix
> > > 2. Add one or two more realistic/challenging classical targets for MCMC (e.g. Neals Funnel,  LGCP, etc) you can see this found in quite standard/seminal ULA-MCMC papers e.g.  [9]
> > >
> > >
> > > [9] Girolami, M. and Calderhead, B., 2011. Riemann manifold Langevin and hamiltonian monte carlo methods. Journal of the Royal Statistical Society Series B: Statistical Methodology, 73(2), pp.123-214.

---

> ### Author Response · Authors · 2023-11-16
>
> Thank you for your swift feedback.
>
> We have incorporated experiments of Neal's Funnel into our Appendix F.5. It is shown that our algorithm can make a better balance between exploration and precision for Neal's Funnel sample.
>
> More comprehensive empirical studies and bags of tricks could be a promising direction for future exploration.
> In this paper, our focus is on strengthening the theoretical underpinnings and enhancing our understanding of the rdMC algorithm  (as well as the benefits of the diffusion-like dynamics) compared with MCMC.
>
> If you have any further inquiries or suggestions, please do not hesitate to reach out to us.

---

> > ### Comment · Reviewer_NhM6 · 2023-11-17
> > **SFS Disucssion in appendix**
> >
> > Dear Authors,
> >
> > Thank you for the funnel results the story is indeed more compelling. I do appreciate the focus of your contributions to be mostly theoretical although it's worth noting ICLR will have a largely applied audience so adding the high dim experiments in the camera-ready version were you get an extra page could heavily increase the audience, and impact (follow-up works).
> >
> > Another thing, I keep missing is that the comparison to ULA on the theoretical side is really not quite remarked clearly enough. there are a couple of paragraphs (e.g. section. 4.1 paragraph following Lemma 2 ) where its slightly mentioned, the shrinking effect of the OU process on $q_t$ made this clear to me, and its somewhat clear how a less complex distribution is being targeted at each point.  However I think it would benefit a lot if you could have either a targeted remark or a subsubsection that emphasizes this much more clearly "The algorithm exhibits lower isoperimetric dependency compared to conventional MCMC techniques", because at the moment I have to navigate back and forth looking for the complexity remarks on ULA, it would help to have something a bit more side by side.
> >
> > I'm currently going through the appendix one minor comment is you use Pinsker's inequality maybe  ~3 times, and on 2 occasions it seems to be without reference e.g. page 17 and 14.
> >
> > Finally in your comparison to SFS there is a Lipchitz constant remark which Im not sure I fully follow:
> >
> > > However, due to the strong ¨assumptions of the Schrodinger-F ¨ ollmer process, it remains open to compare SFS with conventional ¨ MCMC under general conditions. Specifically, SFS requires the Lipschitz continuity of the drift term, which makes the tail of initial and target distribution highly dependen ...
> >
> > Firstly the drift in SFS (with some further derivation using the h-transform / playing around with the provided expectations) can be decomposed into two terms roughly speaking (assuming volatility of 1):
> >
> > $$b(x,t) = \frac{x}{t} +  \nabla \ln p_{T-t} (x) $$
> >
> > Where $ \nabla \ln p_{t} (x) $ is the score of the SDE:
> >
> > $$ x_0 \sim p^*$$
> > $$dx_t = -\frac{x}{T-t} \mathrm{d}t + \mathrm{d}W_t$$
> >
> > This SDE transports the target data distribution $p^*$ to a delta centered at 0 $\mathrm{Law}x_T = \delta_{0}$. Either way, their framework holds if one assumes $ \nabla \ln p_{T-t} (x)$ is Lip, my understanding is that your work also makes this assumption **[A1]**? in which case I'm not sure this **Specifically, SFS requires the Lipschitz continuity of the drift term,** can be highlighted as a differentiating factor.

---

> ### Author Response · Authors · 2023-11-18
>
> Thank you for your helpful comments. We have highlighted the “lower isoperimetric dependency” in Section 4.1.  Moreover, the parts related to Pinsker's inequality were rearranged for better reading experience.
>
> For the Lipschitz continuity of the drift term, we add more explanations in our Appendix. We summarize it as below
>
> The main reason is that the delta distribution side makes $p_t$ ill-behaved (when the variance is approaching 0, which makes $-\log p_t \approx \Vert x\Vert^2/(2\sigma^2)$ is exploding). They also provided a sufficient condition when both $p_*\cdot\exp(\Vert x\Vert^2)$ and its gradient are Lipschitz continuous, and the former is bounded below by a positive value. However, this condition may not be met when the variance of $p_*$ exceeds 1, limiting its general applicability of this condition.
> Thus, the generality of SFS remains an open question.
> On the other hand, due to the smoothness of $p_\infty$ in OU process, the smoothness of $\log p_t$ is widely accepted in diffusion analysis [1,2]. [3] also suggest that the smoothness of concave $\log p_0$ can imply the smoothness of $\log p_t$. In summary, the smoothness of $\log p_t$ in OU process is much weaker than a process towards to a delta distribution, which generalize the analysis to more distributions.
>
>
>
>
> [1] Sitan Chen, Sinho Chewi, Jerry Li, Yuanzhi Li, Adil Salim, and Anru R Zhang. Sampling is as easy as learning the score: theory for diffusion models with minimal data assumptions. arXiv preprint, arXiv:2209.11215, 2022a.
>
> [2] Hongrui Chen, Holden Lee, and Jianfeng Lu. Improved analysis of score-based generative modeling: User-friendly bounds under minimal smoothness assumptions. In International Conference on Machine Learning, pp. 4735–4763. PMLR, 2023.
>
> [3] Holden Lee, Chirag Pabbaraju, Anish Sevekari, and Andrej Risteski. Universal approximation for log-concave distributions using well-conditioned normalizing flows. arXiv preprint arXiv:2107.02951, 2021a.

---

> > ### Comment · Reviewer_NhM6 · 2023-11-18
> >
> > Thank you for the revision and the very insightful discussion these are indeed much more convincing/thorough arguments , I have raised my score again to reflect the recent changes, I think now there’s a much more compelling story in terms of comparing to competing / related approaches as well as the introduction / general flow of the paper has drastically improved.
> >
> > Thank you for putting the time to address my concerns despite the initial bumpy discussion.

---

> > > ### Author Response · Authors · 2023-11-18
> > >
> > > Thanks a lot for your engagement and helpful comments!
> > >
> > > Our conversations have been instrumental in clearing up initial confusions, enhancing the persuasiveness and clarity of our paper. This improved justifications and accessibility for readers perfectly embodies the spirit of the ICLR openreview process.
> > >
> > > Such interactions underscore the immense value of our community. The feedbacks from reviewers and other peers have been exceptionally valuable, demonstrating the power of collaborative improvement inherent in this process.
> > >
> > > Really appreciate your time and effort.

---

### Official Review · Reviewer_qHDx · 2023-10-31

**Soundness:** 3 good
**Presentation:** 3 good
**Contribution:** 3 good
**Rating:** 6
**Confidence:** 5

**Summary:**

This paper explores reverse diffusion in Monte Carlo sampling, transforming score estimation into mean estimation. The algorithm claims to approximate the target distribution accurately, especially for Gaussian mixture models, outperforming Langevin-style MCMC methods. They claim that this algorithm offers a fresh solution for complex distributions.

**Strengths:**

- It is an extremely interesting problem to investigate.
- It is easy to follow.
- Theoretical results seem to support most of their claims.

**Weaknesses:**

- Lack of practical and experimental results for complex distributions e.g. high-dimensional multimodal distributions. The method's performance is based on empirical observations and may not generalize well across diverse datasets or problem domains. Its effectiveness might be limited to specific scenarios and may not be universally applicable.
- Lack of complexity analysis. The combination of different sampling techniques (importance sampling, ULA) adds algorithmic complexity. Managing the interactions between these techniques and ensuring their proper integration can be challenging.  Also, The method might demand significant computational resources, especially when dealing with large sample sizes and high-dimensional spaces. This could limit its practicality for resource-constrained applications.  The sample size required for accurate estimation scales exponentially with the dimension due to the KL divergence between distributions. This exponential growth can make the method computationally infeasible for high-dimensional spaces. Could the authors please elaborate on these issues?
- The accuracy of the estimation relies heavily on the dimensionality of the problem. High-dimensional spaces exacerbate the sample size requirement, making it challenging to apply the method effectively in real-world applications.
- Creating n Monte Carlo samples at each iteration can be computationally expensive, especially if n is large. This might limit the method's scalability and efficiency for high-dimensional or complex distributions.
- The method uses random samples $\xi$ at each iteration. The quality of these random samples is crucial; if they are not truly random or are biased in some way, it can introduce errors in the sampled results. Furthermore, the way the samples are generated and combined in the update equation (step 6) could introduce bias if not done correctly. Biased estimators can lead to incorrect conclusions about the target distribution.
-  The method's performance might degrade in high-dimensional spaces. Monte Carlo methods often face challenges in high-dimensional settings due to the curse of dimensionality, where the sampling space becomes sparse, making it harder to obtain representative samples.
- How sensitive is this framework to initial distribution $p_0$?

**Questions:**

Please refer to the weakness section.

---

> ### Author Response · Authors · 2023-11-15
>
> **Thanks for your suggestions. In summary, we have added more experiments (Appendix F3 F4 F5) and highlight our complexity analysis. Our current one includes both inner, outer loop, and sample-size, which is still much more effient than Langevin-based ones.
>  Both inner iterations and samples do not bring too much computation overhead compared with exponential log-Sobolev constant. We also verify that our algorithm is not sensitive to $\tilde{p}_0$.**
>
> **If you have any further inquiries or suggestions, please do not hesitate to reach out to us. Detailed response are as below.**
>
> >  *Lack of practical and experimental results for complex distributions*
>
> Thanks. We have added the experiments for high-dimensional setting in our Appendix F3. In general, as the leading factor of isolated multi-mode distribution is the log-Sobolev constant (exponential dependency), the dimension dependency (polynomial) is not the main challenge. Thus, our algorithm still has significant improvement.
>
> >  *Lack of complexity analysis. The combination of different sampling techniques (importance sampling, ULA) adds algorithmic complexity. Managing the interactions between these techniques and ensuring their proper integration can be challenging. Also, The method might demand significant computational resources, especially when dealing with large sample sizes and high-dimensional spaces. This could limit its practicality for resource-constrained applications. The sample size required for accurate estimation scales exponentially with the dimension due to the KL divergence between distributions. This exponential growth can make the method computationally infeasible for high-dimensional spaces. Could the authors please elaborate on these issues? The accuracy of the estimation relies heavily on the dimensionality of the problem. High-dimensional spaces exacerbate the sample size requirement, making it challenging to apply the method effectively in real-world applications.*
>
>  Our paper provides a detailed complexity analysis in both Sections 4.1 and 4.2 under different assumptions. We incorporate both gradient complexity and sample complexity within our propositions, directly stemming from the log-Sobolev constant estimation and Theorem 1.
>
> **We highlight that our theory indicate that considering inner * outer loop, our algorithm is still much more efficient than Langevin-based algorithms. Our experimental results also verify this point.**
>
> Additionally, it is noteworthy that our algorithm's dependency on the dimension $ d $ is polynomial, not exponential, further underscoring its efficiency. And our algorithm can reduce the exponential term induced by the log-Sobolev constant
>
> > *Creating n Monte Carlo samples at each iteration can be computationally expensive, especially if n is large. This might limit the method's scalability and efficiency for high-dimensional or complex distributions.*
>
> Thanks for your suggestion. In our analysis, we have already considered the complexity of $n$ samples. The main point of our algorithm is that even with $n$ samples, the total complexity is still less than Langevin-based algorithm. Note that the complexity with respect to $n$ is linear and the needed accuracy is polynomial, but the complexity of the original problem is exponential.
> In addition, the computation of MC samples can be paralled, while exponetial iterations cannot be reduced.
>
>
> > *The method uses random samples at each iteration. The quality of these random samples is crucial; if they are not truly random or are biased in some way, it can introduce errors in the sampled results. Furthermore, the way the samples are generated and combined in the update equation (step 6) could introduce bias if not done correctly. Biased estimators can lead to incorrect conclusions about the target distribution.*
>
> Thank you for you question. Indeed, the quality of these random samples is crucial, so we analysis the necessary accuracy of the inner loop to obtain the proper samples in our Theorem 1. When the error is controlled, we can guarantee the convergence of our algorithm. In real practice, the algorithm is even less sensitive to the inner loop error, as the theoretical bound considers the worst case.
>
> > *The method's performance might degrade in high-dimensional spaces. Monte Carlo methods often face challenges in high-dimensional settings due to the curse of dimensionality, where the sampling space becomes sparse, making it harder to obtain representative samples.*
>
> We appreciate your feedback. In response, we have included high-dimensional experiments in Appendix F3. It is crucial to note that our approach maintains a polynomial dependency on dimensionality, ensuring the identification of representative samples within a finite timeframe.
>
> > *How sensitive is this framework to initial distribution $p_0$*
>
> Thank you for your question. Are you referring $\tilde{p}_0$? If so, we have added a discussion in the current Appendix F4.

---

> ### Author Response · Authors · 2023-11-20
>
> We are grateful once again for your invaluable review. In our revised manuscript and response, we have endeavored to thoroughly address all your concerns and questions.
>
> We've enriched our work with new empirical results and an ablation study to demonstrate the effectiveness and robustness of our algorithm in high-dimensional scenarios. We also highlight our an in-depth complexity analysis, suggesting that our algorithm maintains higher efficiency than Langevin-based methods, considering inner, outer loops, and sample complexity.
> Furthermore, we have refined our writing to ensure clarity and prevent misunderstandings.
>
> Could you please spare some time to review our updated response and verify if it satisfactorily answers your questions? We are eager to engage in further discussion and offer additional clarifications for any new inquiries you may have.

---

> > ### Comment · Reviewer_qHDx · 2023-11-22
> >
> > I'd like to thank the authors for incorporating the revisions specially on the complexity and conducting additional experiments. While it's acknowledged that certain concerns raised by other reviewers are fully valid and the fact there are similarities between this paper and others, I maintain that this paper remains interesting to me. The recent additions made by the authors have addressed some of the main issues I initially had and thus I'd like to keep my score.

---

> > > ### Author Response · Authors · 2023-11-22
> > >
> > > Thank you for your response and suggestions. We'd like to highlight that, despite some overarching similarities between the VI-based diffusion research and our paper, our core contribution and focal points differ significantly, as also noted by Reviewer NhM6.
> > > VI and MCMC are also high related from a high-level perspective.
> > > We are really grateful that Reviewer NhM6 appreciates our contribution after necessary communication and clarifications.
> > > We have diligently addressed all raised concerns and strongly believe that these enhancements are crucial to the value of our paper.
> > >
> > > Should you require any further information or wish for more detailed explanations, please do not hesitate to contact us. We are eager to engage in additional discussions and collaborate closely with you to refine our paper.

---

> ### Author Response · Authors · 2023-11-21
>
> Dear Reviewer qHDx,
>
> Thank you for your insightful feedback once again. We hope that our response addresses your concerns and questions.
>
> As our author-reviewer discussion nears its end, we'd appreciate knowing if your concerns are resolved. We are open for any further discussion and would appreciate a reassessment of the rating in light of the manuscript's improvements.

---

### Official Review · Reviewer_dWe5 · 2023-10-31

**Soundness:** 3 good
**Presentation:** 2 fair
**Contribution:** 3 good
**Rating:** 8
**Confidence:** 2

**Summary:**

The paper considers the problem of reverse diffusion Monte Carlo and shows that the score estimation can be viewed as a mean estimation problem by exploiting a decomposition of the transition kernel. The theoretical properties of the proposed method are extensively analysed. The performance of the proposed method is assessed on the Gaussian mixture models; it is illustrated that the proposed approach performed better than existing Langevin-type MCMC methods.

**Strengths:**

-	Extensive theoretical analysis of the proposed method.

-	Good performance on (a single) toy example.

**Weaknesses:**

-	It is quite hard work to verify the theory. I would expect nothing less from such a paper, so by itself, it is, of course, not a problem. However, I believe there is room for improving the clarity and flow of the proofs.

-	With the current presentation of the results, it is hard to verify the reproducibility of the results; in the experiments, the robustness of the algorithms to the input hyperparameters (for example, the choice of step size \eta in Algorithms 1/2).

**Questions:**

It’s ok that for a non-expert in the specific topic, it may be hard to go over the proofs. However, I believe that a good and precise presentation of the theoretical results can lead to even a non-expert with enough theoretical background to follow and verify the results. Some representative things which could be improved:
-	 In D2 (proof of lemma 2 and 3): the paper refers to Proposition 2 in Ma et al. (2019). However, I cannot find the Proposition 2 in Ma et al. (2019).
-	Lemma 9: not an obvious mismatch between the statement of the lemma and the final line in the proof (RHS of inequality is d/mu for the former, 1/mu for the latter);
-	The proof of lemma 9 starts with “It is known that LSI implies Poincare inequality with the same constant,…”. Perhaps a reference would help.

---

> ### Author Response · Authors · 2023-11-15
>
> **Thank you for your helpful suggestions. We have significantly modified the proof for better reading experience and add more explanation for clarity. We also add a proof sketch for non-experts (Appendix B). We are trying our best to make our analysis user-friendly.**
>
> **If you have any further inquiries or suggestions, please do not hesitate to reach out to us. Detailed response are as below.**
>
>
> > *Q1. Clarity and flow of the proofs.*
>
> Thank you for your suggestion. We have significantly rearranged the proof and lemmas for better reading experience. Moreover, we have added a proof sketch in Appendix B. Please take a look and feel free to tell us if you have any question.
>
> > *Q2. Hyper-parameters.*
>
> We have included the hyperparameters in our appendix and added the hyperlink in the main context.
>
> > *Q3. In D2 (proof of lemma 2 and 3): the paper refers to Proposition 2 in Ma et al. (2019). However, I cannot find the Proposition 2 in Ma et al. (2019). - Lemma 9: not an obvious mismatch between the statement of the lemma and the final line in the proof (RHS of inequality is d/mu for the former, 1/mu for the latter); - The proof of lemma 9 starts with “It is known that LSI implies Poincare inequality with the same constant,...”. Perhaps a reference would help.*
>
> For the Proposition 2 in Ma et al. (2019), Please refer to page 11 of the arXiv version (Appendix B.1), which is a formal version of Proposition 1. We also revised our content to reiterate the PNAS version proposition to make our paper self-contained.
>
> For Lemma 9 (previous notation, current Lemma 22), sorry for the confusion. The previous version of last line is for a single dimension. We have also added the reference for LSI and PI.

---

> ### Author Response · Authors · 2023-11-20
>
> We sincerely appreciate your insightful feedback once again. In response, we have diligently addressed each of your concerns and queries. Notably, we've incorporated a proof sketch and have significantly enhanced the clarity of our proof through thoughtful reorganization. Additionally, we've refined our writing to minimize any possible misunderstandings.
> Would you be able to spare some time to review our revised response and confirm if it adequately addresses your questions? We are eager to engage in further discussions and provide additional clarifications on any new queries you may have.

---

> > ### Comment · Reviewer_dWe5 · 2023-11-20
> > **Score raised**
> >
> > I appreciate the revised version and the changes made by the authors. I believe the manuscript has significantly improved, and therefore, I raised my score.

---

> > > ### Author Response · Authors · 2023-11-20
> > >
> > > Thank you so much for recognition and for your constructive feedback; your support and acknowledgment are greatly appreciated.

---

### Official Review · Reviewer_UPK6 · 2023-11-06

**Soundness:** 2 fair
**Presentation:** 2 fair
**Contribution:** 2 fair
**Rating:** 6
**Confidence:** 4

**Summary:**

This paper considers a diffusion modelling approach to the classical problem of sampling from an unnormalised density.

**Strengths:**

Sampling with reverse diffusions seem to have multiple advantages (compared to usual Langevin dynamics) in terms of how the algorithms behave. This paper builds on this observation and tries to bring the diffusion modelling methodology into regular Monte Carlo sampling.

**Weaknesses:**

Unclear writing and claims not supported rigorously. See more below in the questions part.

**Questions:**

My questions are as follows.

1) at multiple points, the paper claims that the SDE resulting from the diffusion approach has better behaviour, e.g., the last sentence of Section 2.1 claims that the isoperimetric constant of this SDE is better. Right after Lemma 1, another claim is made "It is important to point out that the property of $q_{T-t}(\cdot | x)$ is better than $p_*$". Here as well, the sentence is badly written (what property?) But in any case, these claims, as far as I am able to see are not rigorously proven.

2) Theorem 1 *assumes* that $q_{T-{k\eta}}$ is log-Sobolev, instead of proving something about it. As such I think the whole motivation is unclear as authors didn't show how ill-posedness is tackled by reverse diffusion approach.  Can authors show, if $p_*$ has a log-Sobolev constant, then $q$ does actually have a better behaviour in terms of this constant?

Small comments:

- In Lemma 1, point out where the proof is in Appendix

---

> ### Author Response · Authors · 2023-11-15
>
> **Thank you for your suggestion. We followed your points and try to make our claims more clear and rigorous. We also highlight our analysis of $q$'s  log-Sobolev are in Sec 4.1 and 4.2. We try our best to revise our manuscript to address your concerns.**
>
> **If you have any further inquiries or suggestions, please do not hesitate to reach out to us. Detailed response are as below.**
>
> > *Q1. The rigorousness of the claims.*
>
> Thank you for your suggestion.
> We have revised the claims as below:
>
>
> 1. Thus, we analyze the complexity to estimate the score by drawing samples from the posterior distribution and found that our proposed algorithm is much more efficient in certain cases when the log-Sobolev constant of the target distribution is small.
>
>
> 2. For any $ t > 0 $, we observe that $ -\log q_{T-t} $ incorporates an additional quadratic term. In scenarios where $ p_* $ adheres to a log-Sobolev condition, this term enhances $ q_{T-t} $'s log-Sobolev constant, thereby accelerating convergence. Conversely, with heavy-tailed $ p_* $ (where $ f_* $'s growth is slower than a quadratic function), the extra term retains quadratic growth, yielding sub-Gaussian tails and log-Sobolev properties. Notably, as $ t $ approaches $ T $, the quadratic component becomes predominant, rendering $ q_{T-t} $ strongly log-concave and facilitating sampling.
> In summary, every $ q_{T-t} $ exhibits a larger log-Sobolev constant than $ p_* $. As $ t $ increases, this constant grows, ultimately leading $ q_{T-t} $ towards strong convexity. Consequently, this provides a sequence of distributions with log-Sobolev constants surpassing those of $ p_* $, enabling efficient score estimation for $ \nabla\ln p_{T-t} $.
>
>
> > *Q2. log-Sobolev constant of $q$.*
>
> Thank you for pointing out this. In fact, we are unable to whether $q_{T-k\eta}$ is log-Sobolev with A1 and A2.
> That is the reason why we add Section 4.1 and 4.2 to discuss the assumption of $p_*$ and log-Sobolev constant of $q_{T-k\eta}$. We have added more explanation before 4.1 to highlight this point:
>
>  *Note that the log-Sobolev constants of $q_{T-k\eta}$ depend on the properties of $p_ \ast$, so we should add more assumptions to estimate the log-Sobolev constants for $q_{T-k\eta}$. Next, we will demonstrate the benefits of using $q_{T-k\eta}$ for ill-conditioned LSI or non-LSI  $p_ \ast $.*
>
> > *Q3. Point out where the proof is in Appendix*
>
> Thank you for your suggestion. We have revised our manuscript.

---

> ### Author Response · Authors · 2023-11-20
>
> Thank you sincerely for the time and effort you've invested in reviewing our work; your feedback is invaluable. We've aimed to address your questions clearly and concisely, minimizing misunderstandings for improved readability. Should you have any further concerns or need more information, we're more than willing to assist. If our responses meet your expectations, we would greatly appreciate your acknowledgment of our progress. Thank you once again for your crucial role in enhancing our work. We eagerly await your feedback.

---

> ### Author Response · Authors · 2023-11-21
>
> Dear Reviewer UPK6,
>
> Thank you for your insightful feedback once again. We hope that our response addresses your concerns and questions.
>
> As our author-reviewer discussion nears its end, we'd appreciate knowing if your concerns are resolved. We are open for any further discussion and would appreciate a reassessment of the rating in light of the manuscript's improvements.

---

> > ### Comment · Reviewer_UPK6 · 2023-11-21
> >
> > Thank you for your replies. I decided to keep my score as it is (it is already weak accept) -- as I'd want to see some more precise support for the claims I pointed out in my Q2. Thank you

---

> > > ### Author Response · Authors · 2023-11-21
> > >
> > > We hope that the summary provided is clear and helpful. Should you have any further questions or need additional clarification, please feel free to reach out. We're always ready to engage in further discussion and working with you to improve our paper.

---

> ### Author Response · Authors · 2023-11-21
>
> Thank you for your swift feedback.
>
> We consider two cases as in Section 4.1 and 4.2 for more intuitive demonstrations.
>
> (1) log-Sobolev constant of $p_*$ is exponential to radius $R$ as in [$A_3$], which is common for mixture models. Then due to the OU process, we have that the log-Sobolev constant of any $p_T$ is improved for any $T>0$. Once, the log-Sobolev constant for $q$ is NOT exponential wrt $R$, the reverse diffusion can save the computation:
>
> As Lemma 2 suggest
>
>  *Under [$A_1$], the log-Sobolev constant for $q_t$ in the  forward OU process is $\frac{e^{-2t}}{2 \left(1-e^{-2t}\right)}$
> when
> $0\leq t\leq\frac{1}{2} \ln\left(1+\frac{1}{2L}\right)$.
> This estimation indicate that when quadratic term dominate the log-density of $q_t$, the log-Sobolev property is well-guaranteed.*
>
> It means that the log-Sobolev constant within this interval is independent from $R$, and the endpoint of $p_T$ is smoother and more concentrated than $p_0$ that leads to a better log-Sobolev constant.
>
>
>
> (2) $p_*$ is not log-Sobolev due to the heavy-tail property.
> We can assume the tail behavior of $p_*$, as [$A_4$],
>
> *For any $r>0$, we can find some $R(r)$ satisfying $f _ * (x)+r\left\|x\right\|^2$ is convex for $\left\|x\right\|\ge R(r)$. Without loss of generality, we suppose $R(r) = c _ R/r^n$ for some $n>0,c_R>0$*
> where $n$ defines the growth rate of the tail.
>
> However, due to the additional quadratic of $q$, we can still keep $q$ is sub-Gaussian and log-Sobolev.
>
> *Under [$A_1$] and [$A_4$] the log-Sobolev constant for $q_t$ in the forward OU process is $\frac{e^{-2t}}{6(1-e^{-2t})}\cdot e^{-16\cdot 3L\cdot R^2\left(\frac{e^{-2t}}{6(1-e^{-2t})}\right)}$
> for any
> $ t\geq 0$.*
>
> It means for any $t>0$, we can get a log-Sobolev constant for $q$, but the original problem is non-log-Sobolev. Reverse diffusion provides a procedure that decompose the non-log-Sobolev sampling problem into log-Sobolev sampling subproblems.

---

### Public Comment · ~Mingtian_Zhang1 · 2023-11-13
**Elegent method**

The challenge of sampling from unnormalized densities is significant and holds great relevance in various applications like ML for science or Bayesian methods. A burgeoning area of interest is leveraging diffusion processes to enhance this sampling. However, a key difficulty is that the score of the Gaussian convolved energy distribution is not tractable.

The proposed method is very neat. Although the paper has its limitations, such as its computational intensity for executing multiple LDMCMCs for each score estimation within an LGMCMC, the experiment is somewhat simplistic and the writing can be improved. Despite these factors, the elegance, ingenuity and potential impact of the proposed method cannot be overlooked, suggesting it could play a vital role in advancing the field.

---

> ### Author Response · Authors · 2023-11-15
>
> Thank you for your insightful comments and encouragement. Your support fuels our motivation to further refine the manuscript.
>
> A key aspect of our work involves providing a comprehensive theoretical analysis, particularly addressing questions like the complexity of score estimation at various time steps and the potential advantages of diffusion-based algorithms. We're eager to contribute ideas that may inspire other researchers and spark future innovations.
>
> Recognizing the limitations of our current manuscript, we've dedicated ourselves to enhancing its quality. This includes incorporating additional empirical results and refining our writing for a better reading experience. We're hopeful that these updates will not only address your concerns but also provide valuable insights for the broader research community.

---

### Meta-Review · Area_Chair_fqSk · 2023-12-06

**Metareview:**

For the problem of sampling from a high-dimensional density, the authors derive a new Monte Carlo sampling algorithm using the reverse process in a diffusion model. This involves sampling from a certain posterior distribution, which they illustrate can have improved log-Sobolev constant. They show improvements for Gaussian mixture models both theoretically and empirically. All reviewers gave a positive evaluation of the paper.

**Justification For Why Not Higher Score:**

The methodology of the paper seems very natural in light of previous works (e.g., [El Alaoui, Montanari, Sellke 2022]). Of course it is valuable to work out the details and implement the algorithm.

**Justification For Why Not Lower Score:**

All reviewers gave a positive evaluation. Both theory and experiments are sound.

---

### Decision · Program_Chairs · 2024-01-16

Accept (poster)